# The molecular basis of force selectivity by PIEZO2

Eric M. Mulhall[1✉], Oleg Yarishkin[1], Rose Z. Hill[1,2,3], Anna K. Koster[1] & Ardem Patapoutian[1✉]

PIEZOs are mechanically gated ion channels that transduce force into electrochemical signals[1]. PIEZO1 responds to diverse stimuli including membrane stretch[2] and shear stress[3], whereas PIEZO2 is generally tuned to detect cellular indentation[4,5]. The functional specialization of PIEZO2 is proposed to underlie its distinct physiological roles, including mediating the sense of touch[6,7]. How PIEZO2 achieves this selectivity despite its close structural similarity to PIEZO1 is unclear. Here we combine single-molecule MINFLUX fluorescence nanoscopy with electrophysiology to link the conformational states of PIEZO2 to channel gating in intact cells. We find that PIEZO2 is intrinsically more rigid than PIEZO1, and that disparate mechanical stimuli paradoxically evoke opposite conformational and gating responses in each channel. These unique gating properties arise in part from a connection to the actin cytoskeleton, and we identify filamin-B (FLNB) as a molecular tether that is required for this interaction. This complex alters how force is transmitted to PIEZO2 and confers heightened sensitivity to and selectivity for cellular indentation. PIEZO2 and FLNB are co-expressed in somatosensory neurons and colocalize within tens of nanometres at the end organs of cutaneous mechanosensory afferents. These findings help to explain why PIEZO2 is a specialized mechanosensor and provide a molecular blueprint for understanding how cells decode diverse mechanical stimuli across tissues and organ systems.

PIEZO2 is a mechanotransduction channel that mediates an array of physiological processes in vertebrates, including touch[6], proprioception[8] and respiration[9]. Pathogenic loss- or gain-of-function mutations in this channel can cause extensive mechanosensory defects or debilitating neurological diseases such as Gordon syndrome, Marden–Walker syndrome or distal arthrogryposis type 5 (ref. 10). Despite recent advances in understanding the physiological roles of PIEZO2, the underlying structural correlates of function remain poorly understood.

PIEZO1 and PIEZO2 are homotrimeric membrane proteins with three identical subunits that assemble to form a triskelion[11] (Fig. 1a). Each subunit has a blade of 36 transmembrane domains extending outward and upward from a central C-terminal pore domain, forming a concave shape around 24 nm in diameter in membrane-free structural models[12–15]. The blades are proposed to function as the primary sensors of mechanical force because they directly connect to the ion-conducting pore and, in the case of PIEZO1, blade expansion by membrane tension correlates with channel activation[16–18]. Mechanosensitive ion channels are generally gated through either a 'force from lipid' mechanism driven by membrane tension, a 'force from filament' mechanism involving force transmission through cytoskeletal tethers, or a hybrid of the two[19]. While membrane tension has been established as the primary gating stimulus for PIEZO1[2,12,20,21], comparatively little is known about how PIEZO2 is gated by physiologically relevant forces.

PIEZO1 and PIEZO2 have distinct roles in physiology. While PIEZO1 is predominantly expressed in non-neuronal cells, such as erythrocytes[22] and chondrocytes[23], PIEZO2 is primarily expressed in somatosensory neurons[6], such as those that mediate proprioception and touch. These divergent physiological roles are consistent with their distinct functional properties. PIEZO1 is primarily activated by membrane tension, but PIEZO2 is preferentially activated by cellular indentation[4], the mode of stimulus that many cutaneous mechanosensory neurons are tuned to detect[24]. Indentation stimuli, for example by poking a cell with a blunt probe, apply a localized force that both deforms the membrane and strains the underlying cytoskeleton[25]. By contrast, techniques like cell-attached pressure clamp recordings expand the membrane like a balloon and produce relatively uniform lateral membrane tension[25]. Comparing the responses of mechanotransduction channels to each of these stimulus modalities can provide clues about their gating mechanism. Notably, atomic-force-microscopy experiments show that PIEZO2 is approximately threefold more sensitive to indentation compared with PIEZO1 in heterologous cells[5] yet, paradoxically, PIEZO2 is equally less sensitive to membrane tension in cell-attached or excised patch recordings[26]. Moreover, only a small minority of PIEZO2-transfected cells exhibits mechanosensitive currents in cell-attached recordings, and these currents are consistently small[4,27]. These observations suggest that PIEZO2 might not be gated purely through a force-from-lipid mechanism. Despite these clear differences in function, the cryo-electron

[1]Howard Hughes Medical Institute, Department of Neuroscience, Dorris Neuroscience Center, Scripps Research, La Jolla, CA, USA. [2]Vollum Institute, Oregon Health & Science University, Portland, OR, USA. [3]Present address: Department of Chemical Physiology and Biochemistry, Oregon Health & Science University, Portland, OR, USA. ✉e-mail: emulhall@scripps.edu; ardem@scripps.edu

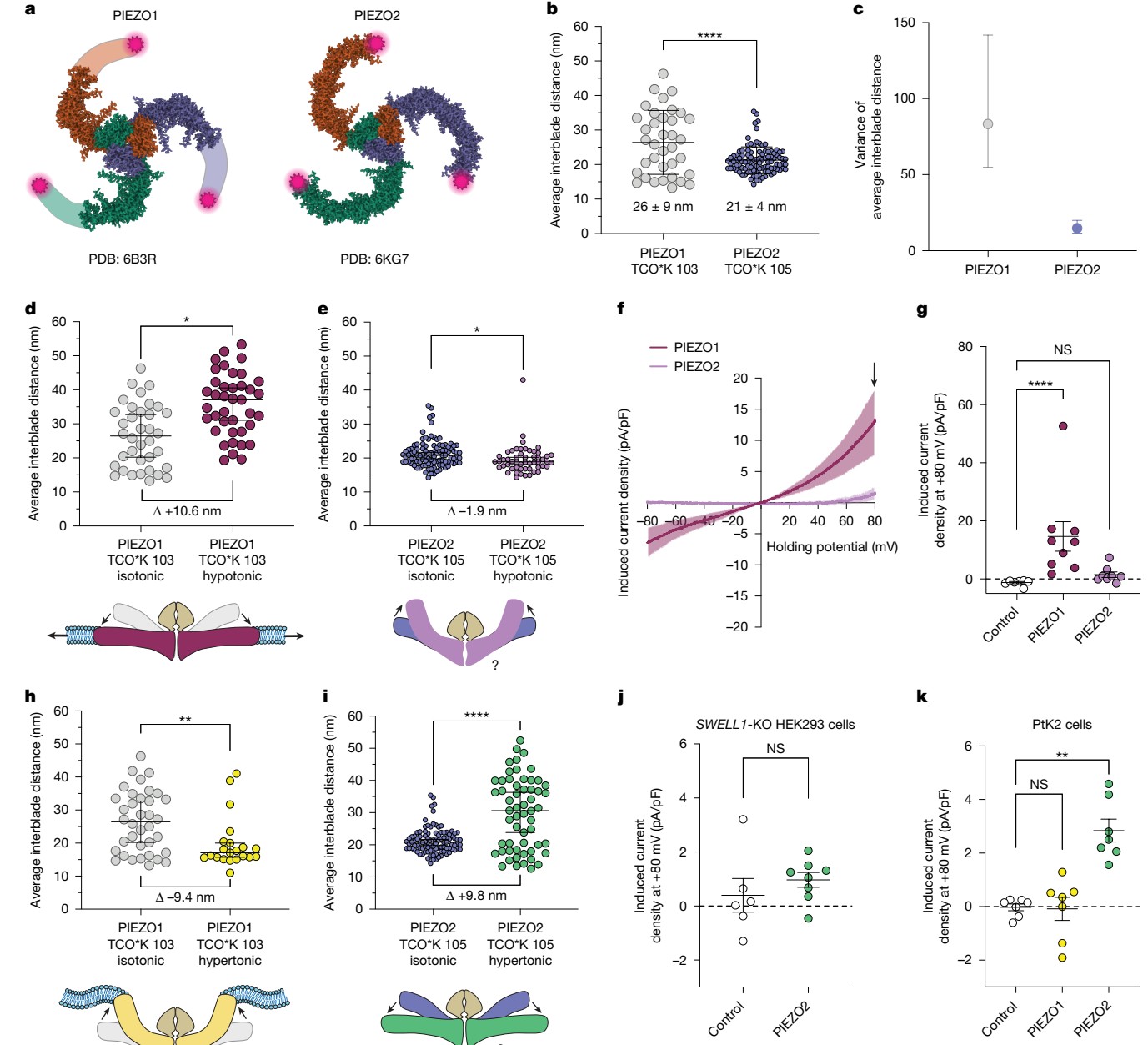

**Fig. 1 | The divergent structural mechanics of PIEZO1 and PIEZO2 in a cell membrane. a**, Structural models of PIEZO1 and PIEZO2, showing the tag positions (magenta stars). **b**, Interblade distances for mouse PIEZO1 tagged at residue 103 (grey; $n = 36$ molecules, $n = 4$ cells) and PIEZO2 tagged at residue 105 (blue; $n = 52$ molecules, $n = 4$ cells) in unstimulated PtK2 cells. Kolmogorov–Smirnov test, ****$P = 6.3 \times 10^{-5}$, $D = 0.44$. **c**, The variance and 95% confidence intervals (CI) of the mean interblade distances in **b**. $F$ test of equality of variances, $P = 8.5 \times 10^{-8}$, $F = 5.6$. **d**, PIEZO1 interblade distances from expansive membrane stretch (red; median = 34.7 nm; $n = 41$ molecules, $n = 3$ cells) versus unstimulated (grey; median = 26.5 nm). Kolmogorov–Smirnov test, *$P = 0.01$, $D = 0.38$. **e**, PIEZO2 during expansive membrane stretch (magenta; median: 18.9 nm; $n = 53$ molecules, $n = 3$ cells) versus unstimulated (blue; median = 19.9 nm). Kolmogorov–Smirnov test, *$P = 0.01$, $D = 0.26$. **f**, Current–voltage curves of hypo-osmotic-swelling-evoked current density for PIEZO1 ($n = 9$ cells) and PIEZO2 ($n = 8$ cells) in *SWELL1*-KO HEK293 cells. The arrow

denotes the current at +80 mV, used for quantification. **g**, Peak stretch-induced current density at +80 mV for **f**, including untransfected controls ($n = 7$ cells). Kruskal–Wallis test with Dunn's post hoc test, ****$P = 6.2 \times 10^{-5}$. **h**, PIEZO1 interblade distances from hyperosmotic stimulus (yellow; median = 19.8 nm; $n = 21$ molecules, $n = 3$ cells) versus unstimulated (grey). Kolmogorov–Smirnov test, **$P = 0.008$, $D = 0.46$. **i**, PIEZO2 during hyperosmotic stimulus (green; median = 30.6 nm; $n = 57$ molecules, $n = 3$ cells). Kolmogorov–Smirnov test; ****$P = 1.3 \times 10^{-9}$, $D = 0.54$. **j**, Peak stretch-induced current from hyperosmotic stimulus in *SWELL1*-KO HEK293 cells expressing PIEZO2 ($n = 8$ cells) and untransfected controls ($n = 6$ cells). Mann–Whitney $U$-test, $P = 0.181$. **k**, The peak induced current density from hyperosmotic stimulus in PtK2 control cells ($n = 6$ cells) versus cells expressing PIEZO1 or PIEZO2 ($n = 7$ cells each). Kruskal–Wallis test, **$P = 0.001$. Data are mean ± s.d. (**b**), median ± 95% CI (**d**, **e**, **h** and **i**) and mean ± s.e.m. (**f**, **g**, **j** and **k**). All statistical tests were two-sided.

microscopy (cryo-EM) structures of each channel are highly similar (Fig. 1a) and provide few clues about their functional specialization, highlighting the need to investigate their structural mechanics in a cellular context.

## Distinct nanomechanics of PIEZO channels

To compare the structural mechanics of PIEZO1 and PIEZO2, we combined MINFLUX fluorescence nanoscopy with fluorogenic DNA PAINT

to measure the conformation of individual molecules in a cell membrane. We previously used MINFLUX to measure the conformational states of PIEZO1 using conjugated spontaneously photoblinking dyes[16]; however, these dyes irreversibly bleach on excitation and produce variable photon counts. To increase the photon yield per molecular position, we used DNA PAINT, which exploits transient binding of fluorophore-labelled oligonucleotides[28]. Programmable binding kinetics through sequence design and the use of more stable dyes can yield far more localization events per molecule, which in turn can resolve a molecular position with higher confidence. We chose *Potorous tridactylus* kidney (PtK2) cells for imaging because they have an exceptionally flat morphology[29] that allows imaging probes to easily reach molecules on the apical membrane adjacent to the coverslip. This minimizes optical aberrations, eliminates the complication of restricted diffusion of probes between the basal membrane and the coverslip and removes the need for permeabilization, which could otherwise expose intracellular channels that are not functionally active.

Using genetic code expansion, the click chemistry substrate *trans*-cyclooctene-lysine (TCO*K) was incorporated into equivalent extracellular loops of the distal blade in PIEZO1 and PIEZO2 (Methods). These sites were labelled with a tetrazine-conjugated DNA oligonucleotide that served as a transient docking site for a complementary oligonucleotide conjugated to the fluorescent dye ATTO 643 freely diffusing in the imaging buffer (Extended Data Fig. 1a). The cells were fixed in an isosmotic cross-linking solution to preserve morphology and imaged without permeabilization to keep the plasma membrane intact. Conventional DNA PAINT is hindered by background from unbound fluorophores, so we used self-quenching fluorogenic imager probes with ATTO 643 and an IowaBlack fluorescence quencher on opposite termini that become unquenched when bound to a docking strand[30]. Under identical imaging conditions, fluorogenic DNA PAINT yielded lower background, higher localization precision and more detections per binding event (Extended Data Fig. 1b–d). We verified that fluorogenic DNA-PAINT did not introduce measurable bias in MINFLUX localizations due to quencher–fluorophore interactions, and that we maintained nanometre-scale stability over extended acquisitions (Extended Data Fig. 2). Raw localizations were clustered and fit with three-dimensional (3D) Gaussian mixture models to obtain precise fluorophore positions, enabling us to directly measure 3D distances between the distal blade domains of individual PIEZO channels (Methods). The resulting enhanced localization density, especially compared with traditional photoblinking dyes[16], enabled us to apply more stringent parameters in our 3D clustering algorithm (Methods and Extended Data Fig. 3), increasing confidence that each cluster corresponds to a single PIEZO protomer.

Compared with PIEZO1, the blades of PIEZO2 are significantly less expanded in the absence of stimulation (Fig. 1b). Notably, while thermal fluctuations alone are thought to cause large deformations in the shape of PIEZO1[16,17], the markedly narrower range of conformational states in PIEZO2 also reveals that the blade domains are much more conformationally rigid than PIEZO1 (Fig. 1c). If blade expansion generally correlates with activation, the larger force required to gate PIEZO2 in excised membrane patches[26] might reflect the extra energy required to bend the more rigid blades by the same distance. These differences in flexibility might also explain why the distal blades of PIEZO2 were resolved by cryo-EM but were not for PIEZO1[12,14].

The primary gating stimulus for PIEZO1 is membrane tension[2,12,20,21]. Expansive membrane stretch from osmotic swelling increases tension[31], similarly to cell-attached pressure clamp recordings[25]. To isolate PIEZO-mediated currents, we used *SWELL1*-knockout (KO) HEK293 cells, which lack the ubiquitous SWELL1-dependent chloride current evoked by hypotonic stimuli that otherwise obscures the measurements of mechanosensitive cation channels[16]. We replicated our previous findings[16], and show that membrane tension from osmotic swelling expands the blades of PIEZO1 (Fig. 1d) and gates the channel

(Fig. 1f). This behaviour is well explained by a force-from-lipids model, whereby mechanical energy from the bilayer is transmitted directly to the channel without requiring intermediary components[19]. Notably, osmotic swelling significantly compacted the blade domains of PIEZO2 by 1.9 nm on average (Fig. 1e) and did not significantly activate the channel (Fig. 1f,g). This lack of activation appears consistent with cell-attached pressure-clamp recordings in heterologous cells, in which only a small minority of cells expressing PIEZO2 displays mechanosensitive currents, and those with responses are invariably small[4,27]. These distinct gating properties indicate that PIEZO2 does not follow a strict force-from-lipids mechanism and may instead recruit additional cellular components that alter how force is transmitted to the channel.

The mismatch in curvature between the concave PIEZO1 structure and planar lipid bilayer results in a bending stress that can substantially expand the blades without externally applied tension[16,17]. A hypertonic extracellular environment reduces cell volume and causes plasma membrane slackening and folding[32]. Consequently, the blades should compact as residual tension is relieved and as the channel partitions into membrane folds matching its intrinsic curvature. While we observed this effect for PIEZO1 (Fig. 1h), we found that the blade domains of PIEZO2 are significantly expanded by the same stimulus (Fig. 1i). This raises the question of how a stimulus that reduces membrane tension can expand the blades of PIEZO2. In addition to slackening, hyperosmotic shrinkage is thought to fold slack membrane into the gaps of the comparably rigid cortical actin meshwork[32]. If PIEZO2 is tightly coupled to actin, we hypothesize that this might recruit a membrane bending force that expands the blade domains. Consistent with this idea, a hypertonic stimulus does not significantly activate PIEZO2 in *SWELL1*-KO HEK293 cells (Fig. 1j), but it does activate the channel in PtK2 cells (Fig. 1k), of which the actin cortex is roughly one order of magnitude stiffer[33,34] and more tightly anchored to the plasma membrane[35]. Together, these data favour a tether-coupled membrane gating model for PIEZO2, whereby deformation of the membrane relative to a connection to actin can apply a gating force to the channel (Fig. 2a).

## Cytoskeletal modulation of PIEZO2 gating

If actin tethering controls how force is transmitted to PIEZO2, then destabilizing the actin cytoskeleton should revert the channel to a force-from-lipids mechanism of gating, consistent with its close structural similarity to PIEZO1 (Fig. 2d). Indeed, when cells were treated with the actin-destabilizing drug cytochalasin D, expansive membrane stretch from osmotic swelling significantly expanded the blades of PIEZO2 and activated the channel (Fig. 2b,c). This could explain why pressure-evoked PIEZO2 currents recorded in cell-attached recordings become larger in magnitude and more frequent after treatment with actin-disrupting compounds[4]. Notably, the unstimulated conformation of PIEZO2 significantly decreased under these conditions but had no measurable effect on the conformation PIEZO1, suggesting that specific tethering of PIEZO2 to actin is responsible for a resting state of blade expansion (Fig. 2b and Extended Data Fig. 4). The significantly smaller distribution of conformational states compared with PIEZO1 in the same conditions also implies that the relative rigidity of PIEZO2 is an intrinsic property of the channel rather than a result of mechanical coupling to actin.

Previous work has implicated the intrinsically disordered intracellular domain between transmembrane helices 12 and 13 (IDR5) in PIEZO2 in activation by indentation forces, and this domain was hypothesized to be involved with a connection to the actin cortex[4] (Fig. 2g). We found that, like actin disruption, deletion of the IDR5 domain in PIEZO2 (PIEZO2(ΔIDR5)) decreased the resting state of expansion of the channel, and expansive membrane stretch from osmotic swelling significantly expanded the blades and activated the channel (Fig. 2e,f). If PIEZO2 does scaffold to actin, we also reasoned that the channel should be immobilized in the cell membrane and that disruption of

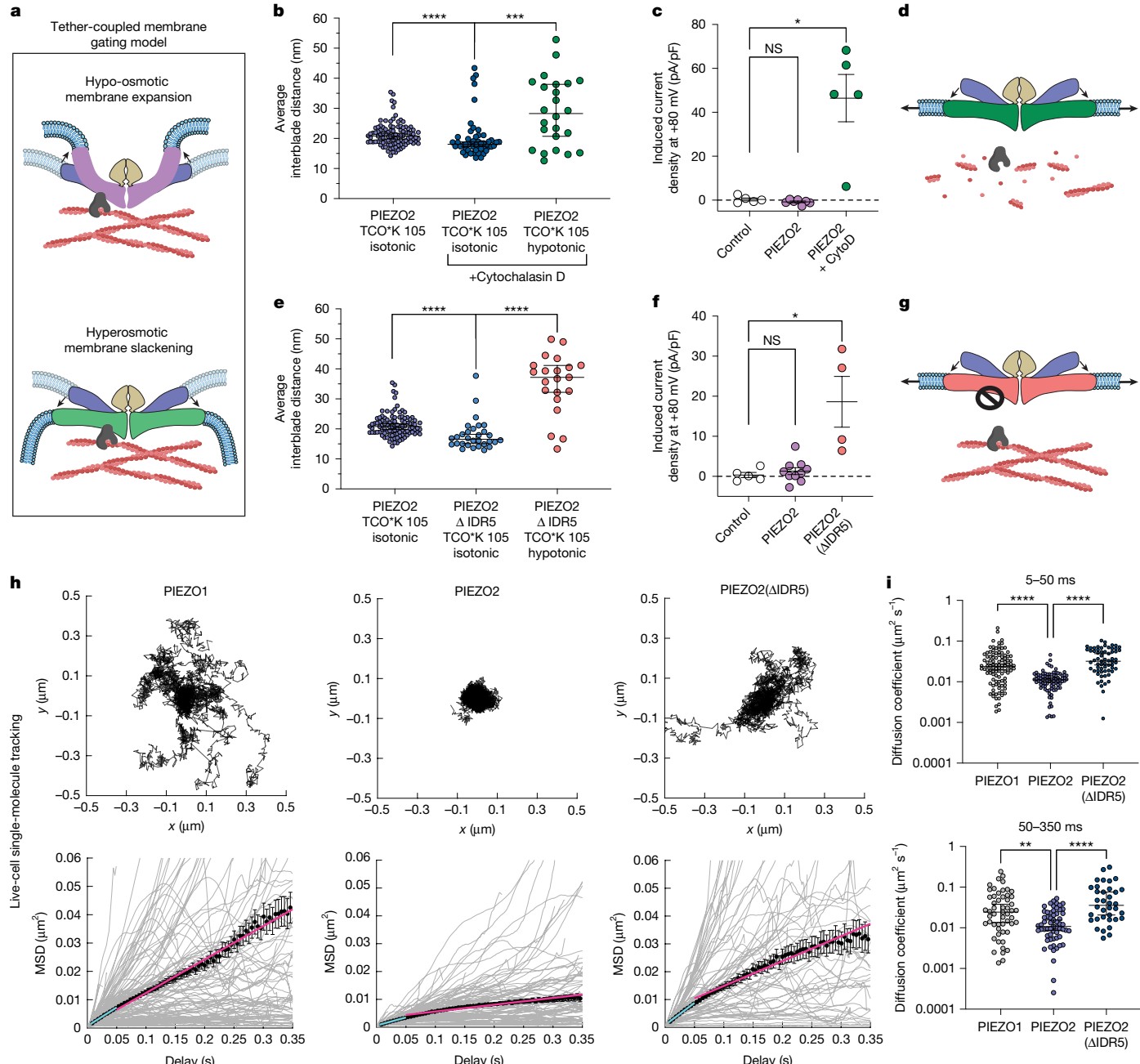

**Fig. 2 | A connection to the actin cytoskeleton regulates the structural mechanics of PIEZO2. a**, Tether-coupled gating model: changes in membrane properties (for example, curvature) relative to a cytoskeletal tether exert force on the blade domains. **b**, Cytochalasin D (CytoD) decreases the resting PIEZO2 interblade distance (dark blue; median = 20.1 nm; $n$ = 51 molecules, $n$ = 3 cells; Kolmogorov–Smirnov test, ****$P$ = $3.4 \times 10^{-5}$, $D$ = 0.40) and permits blade expansion from expansive membrane stretch (green; median = 28.3 nm; $n$ = 24 molecules, $n$ = 5 cells; Kolmogorov–Smirnov test, ***$P$ = 0.0002, $D$ = 0.53). **c**, CytoD enables stretch-evoked PIEZO2 activation in *SWELL1*-KO HEK293 cells. $n$ = 5 (control), $n$ = 7 (PIEZO2) and $n$ = 5 (PIEZO2 + CytoD) cells. Kruskal–Wallis test, *$P$ = 0.048. **d**, Schematic of PIEZO2 blade expansion after actin disruption. **e**, PIEZO2(ΔIDR5) shows reduced interblade distance at rest (light blue; median = 16.6 nm; $n$ = 30 molecules, $n$ = 4 cells; Kolmogorov–Smirnov test, ****$P$ = $9.7 \times 10^{-6}$, $D$ = 0.57) and robust expansion from hypo-osmotic swelling (orange; median = 37.2 nm; $n$ = 21 molecules, $n$ = 6 cells; Kolmogorov–Smirnov test, ****$P$ = $3.9 \times 10^{-7}$, $D$ = 0.79). **f**, Hypo-osmotic swelling gates PIEZO2(ΔIDR5) in *SWELL1*-KO cells. $n$ = 5 (control), $n$ = 10 (PIEZO2) and $n$ = 4 (PIEZO2(ΔIDR5)). Kruskal–Wallis test, *$P$ = 0.011. **g**, Schematic of PIEZO2(ΔIDR5) blade expansion from expansive membrane stretch. **h**, Overlaid MINFLUX single-molecule

trajectories aligned by centre of mass (≥1 s) (top). Bottom, MSD analysis for trajectories with ≥200 localizations. Trajectories (grey) were averaged (50 bins in 350 ms; black circles) and fit at 5–50 ms (cyan, microscopic $D$: PIEZO1 = 0.0218, PIEZO2 = 0.0096, PIEZO2(ΔIDR5) = 0.0264 $\mu m^2 s^{-1}$) and 50–350 ms (magenta, macroscopic $D$: PIEZO1 = 0.0197, PIEZO2 = 0.0041, PIEZO2(ΔIDR5) = 0.0149 $\mu m^2 s^{-1}$). $n$ = 55 (PIEZO1), $n$ = 61 (PIEZO2) and $n$ = 36 (PIEZO2(ΔIDR5)) trajectories. **i**, Diffusion coefficients from individual trajectories in **h**. Top, microscopic $D$ for PIEZO2 (blue; median = 0.011 $\mu m^2 s^{-1}$), PIEZO1 (grey; median = 0.024 $\mu m^2 s^{-1}$) and PIEZO2 ΔIDR5 (dark blue; median = 0.032 $\mu m^2 s^{-1}$). Kruskal–Wallis test, ****$P$ = $3.8 \times 10^{-9}$ (PIEZO1 versus PIEZO2), ****$P$ = $1.3 \times 10^{-14}$ (PIEZO2 versus PIEZO2(ΔIDR5)). $n$ = 104 (PIEZO1), $n$ = 87 (PIEZO2) and $n$ = 65 (PIEZO2(ΔIDR5)) trajectories. Bottom, macroscopic $D$ for PIEZO2 (blue circles; median = 0.011 $\mu m^2 s^{-1}$), PIEZO1 (grey; median = 0.024 $\mu m^2 s^{-1}$) and PIEZO2(ΔIDR5) (dark blue; median = 0.036 $\mu m^2 s^{-1}$). Kruskal–Wallis test, **$P$ = 0.0023, ****$P$ = $1.2 \times 10^{-9}$. $n$ = 55 (PIEZO1), $n$ = 61 (PIEZO2) and $n$ = 36 (PIEZO2(ΔIDR5)) trajectories. Data are mean ± s.e.m. (**c**, **f** and **h**) and median ± 95% CI (**b**, **e** and **i**). All statistical tests are two-sided. The diagrams in **a**, **d** and **g** were created using BioRender; Mulhall, E. M. https://BioRender.com/5k5114d (2026).

this connection should increase its mobility. To test this, we performed single-molecule 3D MINFLUX tracking in live cells expressing PIEZO proteins tagged at the C terminus with a HaloTag and labelled with a Janelia Fluor 635 HaloTag ligand positioned near the channel's centre of mass. We ensured that only individual molecules were tracked by exploiting the roughly quantal photon emission intensity pattern to isolate single emitters resident in the plasma membrane (Methods and Extended Data Fig. 5a,b). Single-molecule trajectories were reconstructed from 3D MINFLUX localizations and diffusion coefficients were obtained by weighted fits of mean-squared displacement (MSD) versus time (Methods). We found that PIEZO1 freely diffuses with a median diffusion coefficient of $D = 0.02$ $\mu m^2$ $s^{-1}$ (Fig. 2h), consistent with independent measurements of PIEZO1 diffusion in red blood cells[36], neural stem cells[37] and fibroblasts[38]. By contrast, PIEZO2 is significantly more immobilized than PIEZO1 (median $D = 0.004$ $\mu m^2$ $s^{-1}$) (Fig. 2h). Analysis of microscopic diffusion during the first 50 ms of the trajectories, before the channel will tend to interact with diffusional barriers created by membrane–cytoskeletal interactions[36], shows that diffusion is restricted essentially from the outset of measuring its motion (Fig. 2i (top)), indicative of a strong tethering interaction.

Compared with the wild-type channel, PIEZO2(ΔIDR5) showed markedly increased mobility (Fig. 2h,i), suggesting that the interaction responsible for immobilization is abolished when IDR5 is removed. Notably, these data differ from an earlier study reporting no change in diffusion after deletion of PIEZO2 IDR5[4]. This discrepancy probably arises from the use of 2D TIRF single-particle tracking on densely labelled channels, which cannot reliably resolve individual molecules nor discriminate those in the plasma membrane from intracellular pools in the secretory pathway. Substitution of PIEZO2 IDR5 into the corresponding intracellular loop of PIEZO1 did not alter the diffusion properties relative to wild-type PIEZO1 (Extended Data Fig. 5c). This implies that the IDR5 domain alone is not sufficient to confer tethering, and that additional PIEZO2-specific elements are required for immobilization. Overall, these findings demonstrate that actin-dependent immobilization of PIEZO2 modifies how force is conveyed to the channel, and that IDR5 is necessary but not sufficient by itself for this interaction.

## FLNB tethering shapes PIEZO2 function

As IDR5 does not possess a canonical actin-binding motif, and most anchored membrane proteins such as integrins, cadherins or ion channels rely on intermediate adaptors to couple to actin[39], we hypothesized that PIEZO2 tethers to actin through an intermediate protein that binds at least partially to IDR5. To identify candidate tethers, we used protein cross-linking and mass spectrometry (MS)-based proteomics (Fig. 3a). We compared the abundance ratio of proteins that cross-linked to either PIEZO2 or PIEZO2(ΔIDR5) and identified candidates that are known to mediate interactions between actin and membrane proteins (Fig. 3b). As disruption of the connection to actin is expected to render PIEZO2 sensitive to expansive membrane stretch, we next performed a small interfering RNA (siRNA) knockdown screen targeting mRNAs of candidate tethering proteins and measured PIEZO2 responses to osmotic swelling. We found that only knockdown of *FLNB*, an actin-binding scaffold protein that binds to cortical actin filaments and anchors membrane proteins to the cytoskeleton[40], elicited significant stretch responses from PIEZO2 (Fig. 3c). Notably, knockdown of the closely related family member *FLNA* had no effect on stretch responses. We next made CRISPR-mediated clonal *FLNB*-knockout cells on the *SWELL1*-KO HEK293 background to confirm that the effects seen with siRNA knockdown are indeed due to loss of FLNB (Extended Data Fig. 6). PIEZO2 in these cells showed significant responses to expansive membrane stretch in all cells tested (Fig. 3d), suggesting that FLNB is a component of the molecular tether that confers force selectivity to PIEZO2.

We next assessed conformational changes of PIEZO2 in the absence of FLNB using MINFLUX. As the *P. tridactylus* genome remains

unsequenced and 21-nucleotide siRNAs produced only partial knockdown in *SWELL1*-KO HEK293 cells after 48 h, we instead used 27-nucleotide double-stranded Dicer-substrate siRNAs (DsiRNAs), which can be up to 100-fold more potent than the corresponding 21-nucleotide single-stranded siRNAs[41]. We transfected four arrayed DsiRNAs against the mRNA encoding Potoroo *Flnb*—designed using the published PtK2 cell transcriptome as a ref. 42—twice over a five-day period to obtain robust knockdown before imaging. Like actin disruption and deletion of PIEZO2 IDR5, *Flnb* knockdown significantly decreased the resting blade conformation in the absence of a stimulus and allowed the blades to expand with expansive membrane stretch (Fig. 3e,f). We also performed live-cell single-molecule MINFLUX tracking and observed that *Flnb* knockdown significantly increased the diffusion rate of PIEZO2 (Fig. 3g–i and Extended Data Fig. 7). These data suggest that FLNB is a component of the molecular tether that links PIEZO2 to actin and is required for conferring selectivity to indentation forces.

PIEZO2 is more sensitive to cellular indentation than PIEZO1[5]. To determine whether actin tethering through FLNB contributes to this heightened sensitivity, we measured mechanically evoked responses from cellular indentation with a blunt glass probe. In *SWELL1*-KO/*FLNB*-KO HEK293 cells, PIEZO2 showed reduced responses at all of the tested indentation depths (Fig. 3j), significantly lower peak current densities (Extended Data Fig. 8a) and a significantly higher gating threshold (Fig. 3k) compared with the control cells. These results demonstrate that FLNB confers heightened sensitivity to cellular indentation to PIEZO2. By contrast, PIEZO1 responses to both indentation and membrane stretch did not differ significantly between *FLNB*-KO and control cells (Extended Data Fig. 8b–d), indicating that PIEZO1 gating is independent of FLNB under these conditions. PIEZO2(ΔIDR5) also exhibited significantly reduced indentation responses and increased indentation threshold[4], although to a larger degree than those observed for *FLNB* deletion (Fig. 3j,k). We suspect that deletion of IDR5 might disrupt the intrinsic structural mechanics of the channel or that FLNB functions as part of a larger protein complex that interacts with this domain.

Compared with the conformation of the membrane-free cryo-EM structure, PIEZO2 is significantly expanded in the cell membrane of PtK2 cells by 2.76 nm, on average, without stimulation (Fig. 3l). Notably, while actin disruption with cytochalasin D compacts the blades to almost exactly that of the cryo-EM structure on average, the more potent drug latrunculin A, along with IDR5 deletion and *Flnb* knockdown, decreased the blade conformation even further (Fig. 3l). This is surprising because the blades presumably experience the same membrane bending forces that expand the blades of PIEZO1 by ~7 nm on average[16], yet we measure conformations even more compact on average than the membrane-free cryo-EM structure. This discrepancy could arise from the focused-refinement steps of the single-particle cryo-EM reconstruction of the blades of PIEZO2[14], which tends to overly weight conformations that align most readily rather than those that might reflect the true average conformation[43]. It could also arise, for example, from the presence of a separate unidentified binding protein that modulates the conformation of the blades. Overall, we propose that, if blade expansion correlates with channel open probability, a more expanded resting state resulting from FLNB tethering might be responsible for lowering the force threshold for gating by decreasing the distance the blades need to be deformed, thereby keeping it primed in a more sensitive state.

As FLNB is responsible for conferring selectivity and sensitivity to indentation forces in heterologous cells, we wondered whether it is present alongside PIEZO2 in peripheral mechanosensory neurons tuned to detect innocuous touch in the skin. We first used single-molecule fluorescence in situ hybridization (smFISH) to quantify *Piezo2* and *Flnb* co-expression in mouse dorsal root ganglion (DRG) neuronal cell bodies positive for *Ntrk2* or *Ntrk3*—markers that collectively cover most of the cutaneous low-threshold mechanoreceptors (LTMRs)[44]—and

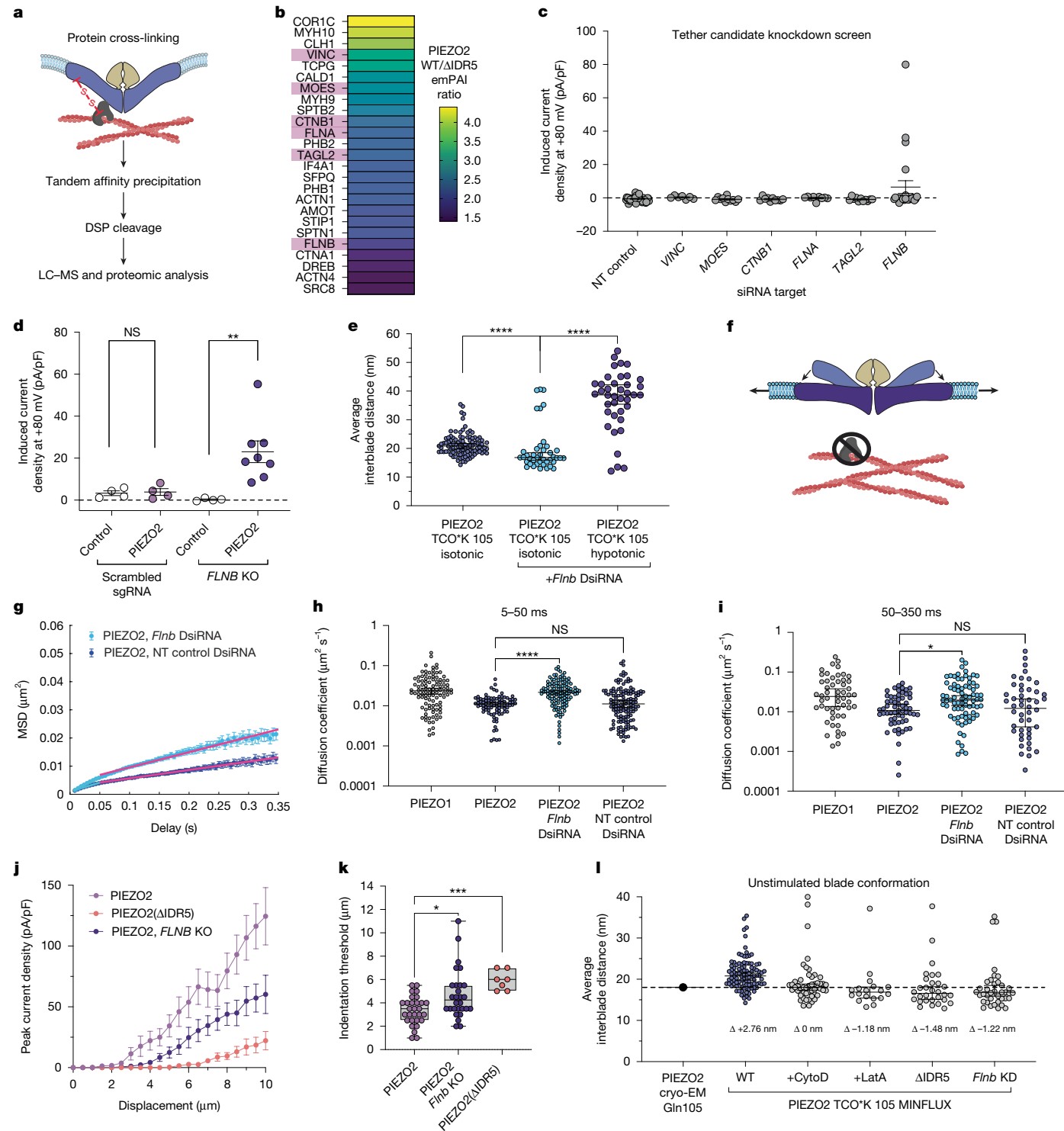

**Fig. 3** | See next page for caption.

found 95% and 96% co-expression, respectively (Extended Data Fig. 9a–d). We next co-stained PIEZO2 and FLNB protein in fixed skin sections from *Piezo2smFP-Flag* knock-in mice using antibodies against endogenous FLNB and the smFlag epitope, focusing on two specialized mechanoreceptor terminals defined by these LTMR markers: Meissner corpuscles, which detect touch and low-frequency vibration in glabrous skin; and lanceolate endings, which encircle hair follicles in hairy skin to sense their deflection[24]. We found that FLNB and PIEZO2 co-localize prominently within both types of mechanoreceptor endings (Fig. 4a). To determine whether FLNB and PIEZO2 lie within a

reasonable distance needed for direct interaction, we performed STED super-resolution imaging of lanceolate endings stained for PIEZO2 and FLNB with a full width at half maximum (FWHM) resolution of around 60–80 nm and quantified puncta overlap using intensity-profile correlations (Fig. 4b–e, Methods and Extended Data Fig. 9e–g). Analysis of FLNB and PIEZO2 intensity profiles in lanceolate endings yielded a mean Spearman's rank correlation of $\rho = 0.72$ (Fig. 4e), indicating that the two proteins lie within just tens of nanometres of each other. This nanoscale proximity implies that FLNB is ideally positioned to tether PIEZO2, supporting a model in which they form a physically

**Fig. 3 | FLNB is a molecular tether that confers force selectivity and sensitivity to PIEZO2. a**, The cross-linking MS workflow. LC, liquid chromatography. **b**, The exponentially modified protein abundance index (emPAI) ratio of proteins cross-linked to PIEZO2 versus PIEZO2(ΔIDR5). Candidate cytoskeletal–membrane tethers highlighted (magenta). **c**, Stretch-induced current in *SWELL1*-KO HEK293 cells expressing PIEZO2 with non-targeting siRNA ($n = 46$), *FLNB* siRNA ($n = 24$) or other candidate siRNAs (*VINC* ($n = 7$), *MOES* ($n = 11$), *CTNB1* ($n = 10$), *FLNA* ($n = 9$) and *TAGL2* ($n = 11$)). **d**, Stretch-induced current in *SWELL1*-KO HEK293 cells electroporated with scrambled CRISPR sgRNA (untransfected or expressing PIEZO2; $n = 4$ cells each) compared with clonal *SWELL1*-KO/*FLNB*-KO cells (untransfected ($n = 4$) or expressing PIEZO2 ($n = 8$)). Mann–Whitney *U*-test, **$P = 0.004$. **e**, The PIEZO2 interblade distance in WT PtK2 cells (blue; median = 18.0 nm; $n = 30$ molecules, $n = 4$ cells) versus PtK2 cells with Potoroo *Flnb* DsiRNA (cyan; median = 16.8 nm; $n = 41$ molecules, $n = 3$ cells; Kolmogorov–Smirnov test, ****$P = 4.5 \times 10^{-7}$, $D = 0.79$) versus PtK2 cells with *Flnb* DsiRNA + hypo-osmotic swelling (purple; median = 37.0 nm; $n = 39$ molecules, $n = 3$ cells; Kolmogorov–Smirnov test, ****$P = 3.2 \times 10^{-10}$, $D = 0.51$). **f**, Schematic of expansive stretch without FLNB. **g**, MINFLUX tracking MSD fits: 5–50 ms (cyan, PIEZO2 + non-targeting DsiRNA: $D = 0.0106$ μm$^2$ s$^{-1}$; PIEZO2 + *Flnb* DsiRNA:

$D = 0.0181$ μm$^2$ s$^{-1}$) and 50–350 ms (magenta, PIEZO2 + non-targeting DsiRNA: $D = 0.00495$ μm$^2$ s$^{-1}$; PIEZO2 + *Flnb* DsiRNA: $D = 0.0090$ μm$^2$ s$^{-1}$). $n = 55$ (PIEZO1), $n = 61$ (PIEZO2), $n = 81$ (PIEZO2 *Flnb* DsiRNA) and $n = 50$ (PIEZO2 NT DsiRNA) trajectories. **h**, Microscopic $D$ from individual trajectories. $n = 104$ (PIEZO1), $n = 87$ (PIEZO2), $n = 126$ (PIEZO2 *Flnb* DsiRNA) and $n = 138$ (PIEZO2 + non-targeting DsiRNA) trajectories. Kruskal–Wallis test, ****$P = 2.1 \times 10^{-10}$. **i**, Macroscopic $D$ from individual trajectories. $n = 55$ (PIEZO1), $n = 61$ (PIEZO2), $n = 81$ (PIEZO2 *Flnb* DsiRNA) and $n = 50$ (PIEZO2 non-targeting DsiRNA) trajectories. Kruskal–Wallis test, *$P = 0.0183$. **j**, Indentation-evoked current in *SWELL1*-KO cells. $n = 26$ (PIEZO2), $n = 7$ (PIEZO2(ΔIDR5)) and $n = 20$ (PIEZO2 FLNB KO) cells. **k**, The indentation threshold for macroscopic current from **j**. Kruskal–Wallis test, *$P = 0.0214$, ***$P = 0.0002$. The box plots show the median (centre line), interquartile range (box limits, 25th–75th percentiles) and the minimum and maximum values (whiskers). **l**, Measured PIEZO2 blade conformation relative to cryo-EM (18.04 nm). For PIEZO2 + latrunculin A, median = 16.9 nm; $n = 18$ molecules, $n = 4$ cells; other values and $n$ are described in Figs. 1b, 2b,e and 3e. Data are mean ± s.e.m. (**c**, **d**, **g**, **j** and **k**) and median ± 95% CI (**e**, **h**, **i** and **l**). All statistical tests were two-sided. The diagrams in **a** and **f** were created using BioRender; Mulhall, E. M. https://BioRender.com/5k5114d (2026).

contiguous mechanotransduction complex that facilitates gating by low-threshold indentation forces.

Finally, to test whether FLNB modulates the force selectivity endogenous PIEZO2 in somatosensory neurons, we recorded mechanically evoked currents from dissociated mouse DRG neurons. We restricted our analysis to large-diameter (≥70 μm) neurons, of which the mechanical responses are predominantly mediated by PIEZO2[45]. We elicited expansive membrane stretch with osmotic swelling in the presence of the SWELL1 inhibitor DCPIB, but did not observe significant currents above the baseline (Fig. 4f). This indicates both that SWELL1 is sufficiently blocked by DCPIB, and that endogenous PIEZO2 does not respond to expansive membrane stretch. By contrast, DsiRNA-mediated *Flnb* knockdown elicited significant responses to expansive membrane stretch compared with control cells (Fig. 4f), indicating that FLNB is required for conferring force selectivity to PIEZO2 in somatosensory neurons.

## Discussion

Here we provide evidence that PIEZO2 acquires its exquisite sensitivity to and selectivity for cellular indentation from a physical tether to the cortical actin network through FLNB. This scaffold fundamentally alters how force is transmitted to the channel. Whereas the primary gating force for PIEZO1 is lateral membrane tension, PIEZO2 appears to instead sense deformation of the bilayer against its FLNB–actin linkage. This difference yields distinct mechanisms of activation, highlighted by the fact that identical mechanical stimuli drive opposite conformational and gating responses (Fig. 1d–k).

Even without FLNB, the structural mechanics of PIEZO1 and PIEZO2 are intrinsically different (Figs. 1b and 3e). The blades of PIEZO2 have far higher apparent rigidity compared with those of PIEZO1 because they explore a smaller distribution of conformational states and are bent significantly less by the curvature mismatch of the planar plasma membrane and bowl-shaped structure of the channel (Fig. 1b,c). Notably, PIEZO2 blades in cells are even more compacted than in the membrane-free cryo-EM structure (Fig. 3l), despite ostensibly experiencing the same bending forces that expand PIEZO1 blades by around 7 nm on average[16]. Future studies could use targeted mutagenesis of predicted high-affinity interfaces to pinpoint the regions that confer this rigidity and reveal their impact on gating mechanics.

FLNB tethering biases the blades of PIEZO2 into a pre-expanded state, which correlates with a lower activation threshold by cellular indentation (Fig. 3). This indicates that tethering makes the channel more sensitive to indentation forces by reducing the distance the blades need to be deflected to open the ion-conducting pore. In cells

or tissues that naturally do not express FLNB, we therefore predict PIEZO2 to act as a higher-threshold sensor of membrane tension, consistent with its higher pressure activation threshold in cell-attached recordings[26]. Differentially expressing PIEZO1 and PIEZO2 along with FLNB or other modulators might be a basic mechanism by which cells can calibrate mechanical sensitivity across a wide dynamic range.

Filamins are approximately 280 kDa actin-scaffolding proteins that consist of three family members—FLNA, FLNB and FLNC[40]. Each protein contains an N-terminal actin-binding domain, two rod regions of 24 immunoglobulin-like repeats separated by a flexible hinge and a C-terminal self-association domain (Extended Data Fig. 6c). Filamins are broadly implicated in cytoskeletal organization and are proposed to function as actin filament cross-linkers that influence cell shape and mechanical properties[40]. Both strong cortical coupling to the membrane and a stiff, highly cross-linked cortex tends to inhibit mechanosensitive ion channels by dampening the propagation of membrane tension[46–48]. For example, *FLNA* deletion increases PIEZO1 activity in smooth muscle cells, probably through changes in cytoskeletal stiffness rather than direct tethering[49]. However, *FLNB* KO did not significantly affect PIEZO1 responses to indentation or membrane stretch (Extended Data Fig. 8b–d), suggesting that FLNB does not substantially impact cell stiffness in our conditions. Filamins also prominently act as a scaffold for membrane proteins such as integrins, β-spectrins, G-protein-coupled receptors and ion channels[40]. Consistent with this role, our results identify FLNB as a component of the scaffold required for a direct channel–actin connection.

FLNB is expressed in many cell types throughout the body and in a variety of immortalized cells lines[50], which might explain why PIEZO2 appears to broadly retain its mechanosensitive specialization. Filamins are also regulated by various processes including calcium-dependent proteolysis, phosphorylation and mechanical unfolding[51], which may in turn impact PIEZO2 function. For example, protein kinase A (PKA) modulates the activity of both PIEZO2 and filamins[52,53]. Notably, PKA modulation of PIEZO2 requires the cytoskeleton-binding IDR5 segment—despite the absence of PKA phosphorylation sites within this region—suggesting that FLNB could be the phosphorylation target[52]. This type of modulation might tune the PIEZO2–actin linkage dynamically and could be involved in switching PIEZO2 from a lower-threshold sensor of indentation to a higher-threshold sensor of membrane tension depending on the needs of the cell. Clinically, loss of FLNB function in humans and mice can cause perinatal respiratory distress, musculoskeletal defects and distal arthrogryposis[44]. These phenotypes significantly overlap with those caused by deleterious mutations in *PIEZO2*[10], suggesting shared physiological roles. Although it is beyond

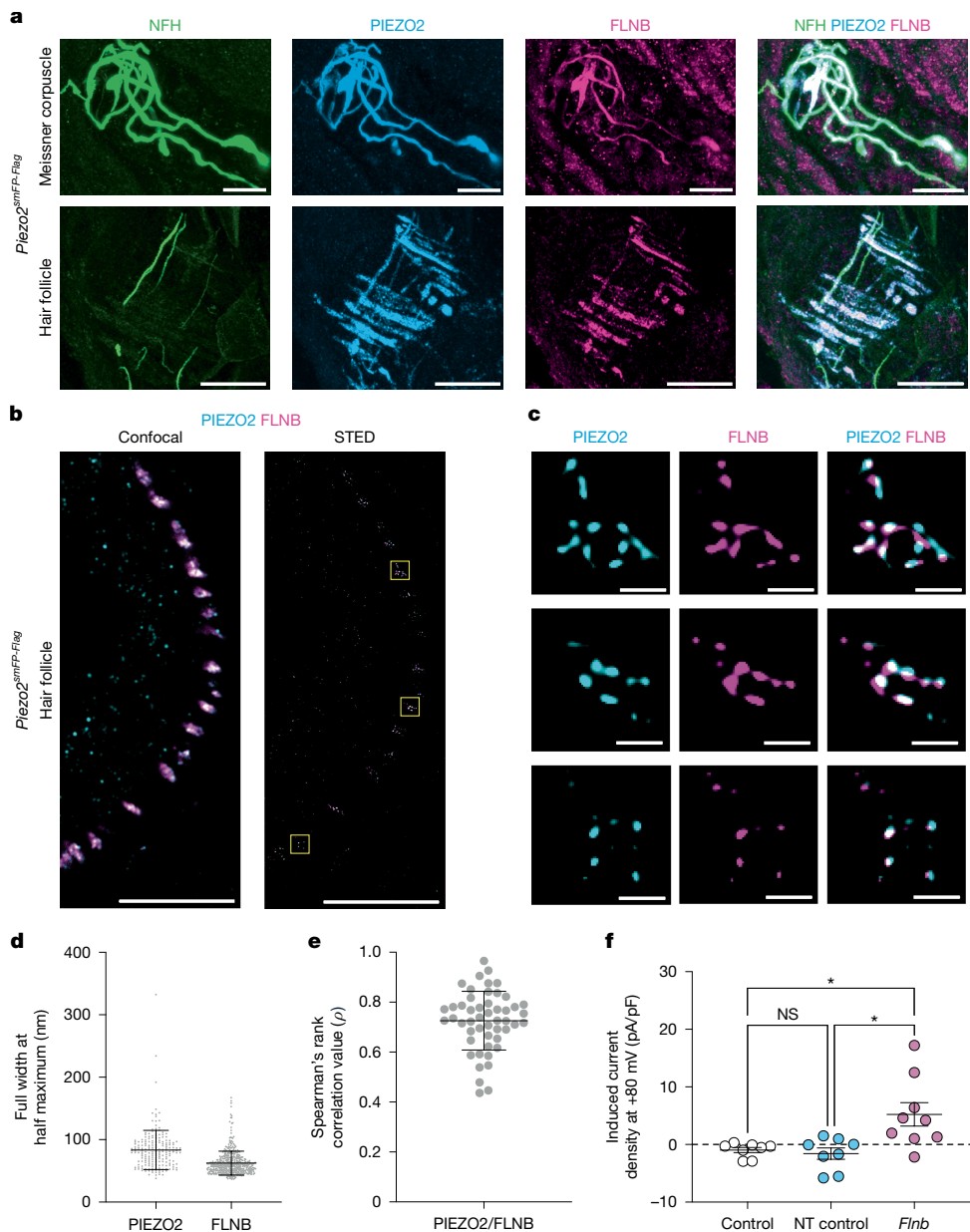

**Fig. 4 | FLNB and PIEZO2 in somatosensory neurons. a**, Representative images of sectioned skin from *Piezo2^smFP-Flag* mice that contain end-organs formed by *Ntrk2*[+] and/or *Ntrk3*[+] LTMRs: lanceolate endings around hair follicles from back skin (top, five sections from two mice) and Meissner corpuscles from glabrous skin of paw digits (bottom, six sections from one mouse). Each section was co-stained with antibodies against NFH, PIEZO2–Flag and FLNB. Scale bars, 10 μm. **b**, Representative super-resolution STED microscopy image of PIEZO2–Flag and FLNB immunostaining in lanceolate endings showing co-localization of single puncta (six sections from two mice). Scale bars, 10 μm. **c**, Isolated single lanceolates from the yellow boxes in **b** (representative images of six sections from two mice). Scale bars, 500 nm. **d**, FWHM resolution of all identified peaks.

Mean ± s.d. FWHM: anti-PIEZO2–Flag = 83 ± 31 nm, anti-FLNB = 62 ± 19 nm. *n* = 201 (PIEZO2) and *n* = 469 (FLNB) puncta. **e**, Quantification of colocalization using Spearman's rank correlation value *ρ* = 0.72 ± 0.12 (mean ± s.d.). *n* = 50 lanceolates from 5 follicles and 2 mice. **f**, The amplitude of stretch-induced currents in dissociated large diameter (≥70 μm) DRG neurons without any treatment (control, *n* = 8 cells), nucleofected with non-targeting DsiRNA (*n* = 8 cells) or nucleofected with DsiRNAs targeting *Flnb* (*n* = 9 cells). *n* = 3 mice for each condition. Two-sided Kruskal–Wallis test with Dunn's multiple-comparison test; *P* = 0.9999 (control versus non-targeting DsiRNA), *P* = 0.0105 (control versus *Flnb* DsiRNA), *P* = 0.0201 (non-targeting DsiRNA versus *Flnb* DsiRNA).

the scope of this current study, future studies should carefully examine how disruption of the PIEZO2–FLNB connection alters touch discrimination, proprioception and related behaviours.

In summary, we directly measured the force-dependent conformational changes of PIEZO2 during channel gating and established a direct link between the structural state of the channel and its function. Compared with PIEZO1, differences in intrinsic structural mechanics and actin tethering through FLNB fundamentally alter the way PIEZO2

is gated by mechanical force. These data lay the foundation for understanding how PIEZOs are functionally specialized to detect diverse mechanical stimuli. Peripheral mechanosensory neurons distinguish indentation, membrane tension, phasic displacement and osmotic pressure to encode touch, proprioception, vibration and bladder fullness. This functional diversity, despite a common reliance on PIEZO2, suggests that sensory end organs contain additional specialized ultrastructural features beyond FLNB, such as extracellular tethers

or caveolae[54,55], that modulate how PIEZO2 responds to mechanical stimuli, enabling them to generate specific sensory outputs. Identifying and characterizing these specialized ultrastructural features will be essential for defining the molecular basis of PIEZO2-mediated mechanotransduction.

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

## Methods

### Study design

No statistical methods were used to predetermine sample size and the sample size was based on a previous study[16]. All attempts at replication were successful, and all experiments were repeated more than once, as indicated in the figure legends. The experiments were not randomized, and investigators were not blinded to allocation during experiments and outcome assessment. For MINFLUX, all experiments were repeated at least three times with separate biological and technical replicates, imaged over at least two separate days. Label-free controls were included to ensure signal specificity. For electrophysiology, experiments were conducted as previously described without blinding[1,6,16]. Untransfected control groups were measured on each day of experimentation, and each separate manipulation was paired with wild-type PIEZO1 and PIEZO2 control measurements.

### Expression constructs

The coding sequence of mouse PIEZO2 (UniProtKB: Q8CD54) was codon optimized, synthesized and cloned into the pcDNA3.1 plasmid. For imaging and MS experiments, an amber stop codon was inserted after amino acid 104 (TCO*K 105) through site-directed mutagenesis using the Q5 Site-Directed Mutagenesis Kit (New England Biolabs), and a C-terminal HaloTag and Strep-Tag II. Labelled PIEZO1 plasmids with a tag after position 102 (TCO*K-103) were prepared the same way from mouse PIEZO1 (UniProtKB: E2JF22), as described previously[16]. For genetic code expansion, the tRNA and tRNA synthetase expression plasmid of pNEU-hMbPylRS-4xU6M15 (Addgene, 105830) was modified to have a mNeonGreen sequence upstream of the tRNA synthetase sequence separated by a T2A self-cleaving peptide to mark transfected cells. For electrophysiology experiments, codon-optimized mouse *Piezo2* was cloned into the pcDNA3.1 vector upstream of an IRES mNeonGreen sequence, and mouse *Piezo1* was cloned into the pcDNA3.1 vector upstream of an IRES eGFP sequence. PIEZO2(ΔIDR5) was created by deleting the coding sequence of amino acids 621–673 as described previously[4]. The PIEZO1 + IDR5 chimera was created by replacing the fifth IDR of PIEZO1 (amino acids 551–575) with the corresponding IDR5 region from PIEZO2 (amino acids 620–672). The sequences of each plasmid were verified using whole-plasmid sequencing (Plasmidsaurus). All DNA sequences were viewed and designed in SnapGene software (Dotmatics).

### Coverslips with embedded gold fiducials

#1.5 D263 borosilicate coverglass (Warner Instruments) were cleaned by boiling in 1% Hellmanex III detergent (Hellma) in MilliQ water and sonicating for 15 min in a water bath sonicator. The coverslips were washed five times with MilliQ water, exchanged into 100% ethanol and then dried and placed onto a sheet of 4 in × 4 in × 1/8 in 304 stainless steel. Then, 150 nm gold nanoparticles with Nanopartz Surface Polymer (Nanopartz) were prepared by diluting to 5 μg ml⁻¹ in 100% ethanol and sonicated in a bath sonicator to break up aggregates. This solution was applied to the coverslips at 0.079 μl mm⁻² of surface area and allowed to dry. The coverslips were then covered with a borosilicate glass Petri dish and placed into a muffle furnace. The furnace was heated to 600 °C at a rate of 30 °C min⁻¹, held at this temperature for 5 h and then allowed to cool overnight. The coverslips were stored at room temperature until use. Immediately before plating cells, the coverslips were coated with a 1:100 dilution of LDEV-free Matrigel (Corning) in DMEM (Thermo Fisher Scientific).

### Cell preparation for structural MINFLUX imaging

PtK2 cells (ATCC, CCL-56) were maintained at 37 °C with 5% $CO_2$ in minimum essential medium (MEM) supplemented with 2 mM GlutaMAX, 25 mM HEPES (Thermo Fisher Scientific), 10% FBS, 1 mM sodium pyruvate (Gibco), 1× MEM non-essential amino acids solution (Gibco) and 100 μg ml⁻¹ penicillin and streptomycin. Cells were authenticated by the supplier by morphological analysis, species verification by isoenzymology, short-tandem-repeat profiling and mycoplasma testing. Cells were further verified to be free of mycoplasma using the using the MycoAlert Mycoplasma Detection Kit (Lonza). The cells were plated onto Matrigel-coated coverslips with embedded gold fiducials, exchanged into a medium containing 250 μM of the click amino acid *trans*-cyclooct-2-en-L-lysine (axial isomer) (SiChem), and transfected with 1 μg of an equimolar ratio of PIEZO expression plasmid and of the tRNA/tRNA synthetase expression plasmids using TransfeX transfection reagent (ATCC). After 48 h, the cells were washed four times with prewarmed medium in 15 min intervals to remove excess click amino acid. The cells were washed in prewarmed 1× PBS and then fixed in prewarmed 1× PBS containing 0.8% PFA and 0.1% glutaraldehyde for 15 min.

For osmotic stimulation experiments, the cells were washed in 1× PBS and then exposed to a modified Ringer's solution at 120 mOsm (48.8 mM NaCl, 5 mM KCl, 10 mM HEPES (pH 7.40) and 10 mM D-glucose) or at 480 mOsm (140 mM NaCl, 5 mM KCl, 10 mM HEPES (pH 7.40), 10 mM D-glucose and 190.3 mM mannitol) for 2.5 min at room temperature. The cells were then fixed in the same osmotic solution containing 0.8% PFA and 0.1% glutaraldehyde for 15 min and quenched in 1× PBS containing 25 mM Tris, pH 8.0. The osmolality of all solutions was determined to be ±5 mOsm using a vapour pressure osmometer.

For experiments with cytochalasin D or latrunculin A, a stock solution of 10 mM in DMSO was diluted to 10 μM in prewarmed medium and added to the cells. After incubation at 37 °C for 30 min, the cells were washed and fixed as described above, except each solution, including fixatives, contained 10 μM cytochalasin D or latrunculin A.

After fixation, all of the coverslips were washed in 1× PBS and quenched with 1× PBS + 25 mM Tris pH 8.0 for 10 min. The coverslips were then blocked in 1× PBS + 1% BSA for 10 min, PIEZOs were labelled with a custom DNA PAINT docking strand modified with a 3′ tetrazine (Supplementary Table 1) at 1 μM in blocking solution for 15 min and washed with blocking solution and 1× PBS. After labelling, cells were washed with and mounted in DNA PAINT imaging buffer (1× PBS + 500 mM NaCl + 0.5 mM EDTA) containing an enzymatic oxygen-scavenging system of 3,4-dihydroxybenzoic acid and protocatechuate 3,4-dioxygenase (from *Pseudomonas*) in addition to the triplet-state quencher Trolox ((+/−)−6-hydroxy-2,5,7,8-tetramethylchromane-2-carboxylic acid) as described previously[56]. This imaging solution containing 1–10 nM of a custom, complementary fluorogenic DNA PAINT imaging strand modified with a 5′ ATTO 643 dye and a 3′ IowaBlack fluorescence quencher (Supplementary Table 1). The coverslip was placed onto a glass slide containing a cavity well (Globe Scientific) filled with imaging buffer and was then sealed onto the slide using Elite Double 22 dental epoxy (Zhermack).

### Cell preparation for MINFLUX tracking

For tracking single PIEZO ion channels, the cells were transfected and prepared as for structural MINFLUX imaging. After washing away excess click amino acid, the cells were labelled with 0.5–2 nM Janelia Fluor 635-HaloTag ligand (Janelia Materials) in culture medium for 15 min at 37 °C, and then washed several times with fresh medium before incubating the cells for 30 min at 37 °C. This labelling concentration was chosen to obtain sparsely labelled channels conjugated to single dyes with non-overlapping fluorescent puncta. After another wash step, the cells were maintained at 37 °C for up to 4 h before imaging. Immediately before imaging, the cells were exchanged into supplemented culture medium without Phenol Red and mounted and sealed onto a glass slide with a cavity well as described above. The coverslip was imaged for a maximum of 1 h before discarding.

### 3D MINFLUX imaging

MINFLUX data were acquired on the commercial MINFLUX 3D microscope an Olympus IX83 microscope body (Abberior Instruments)

using Imspector software (v.16.3.15645-m2205) with MINFLUX drivers. A ×100 oil-immersion objective lens (UPL SAPO100XO/1.4, Olympus) and a 642-nm excitation laser was used for imaging. Transfected cells were identified by expression of the mNeonGreen fluorescent marker driven by the tRNA/tRNA synthetase plasmid. A field of view around the cell was chosen containing at least three separate embedded gold nanoparticles for active sample stabilization through back-scattering from a 980 nm laser source through a closed control loop, typically resulting in less than 1 nm mean s.d. in the $x,y,z$ axes. A 5–25 $\mu m^2$ region of interest (ROI) was chosen at the top face of the flat extension of a PtK2 cell. For structural MINFLUX imaging, at least three isolated gold fiducial nanoparticles were chosen for use by the active beamline stabilization system, and the localization error was verified to be less than 5 nm for each fiducial. The sample was imaged using a 5–12% 642-nm laser power, measured to be around 4.30 µW per percent set power at the sample plane. The pinhole diameter was set to be 0.47–0.6 a.u. The total measurement time varied between 5 and 24 h. For MINFLUX tracking, the sample was imaged using 2% 642-nm excitation laser power with a pinhole diameter of 0.8 a.u.

## MINFLUX data analysis for 3D structural imaging in fixed cells

For structural MINFLUX imaging, raw final valid localizations from the last targeting iteration of the 3D imaging sequence (Supplementary Table 2) were exported from Imspector as a .mat file. Custom MATLAB analysis software was used to identify and segregate clusters of three localizations essentially as previously described[16], with some modifications. All clustering and distance calculations were performed in 3D on $xyz$ coordinates and the reported interblade distances are 3D Euclidean distances. In brief, $xyz$ coordinates were imported, and a 0.7 correction factor was applied to $z$ coordinates to correct for refractive index mismatch. To remove poorly localized molecules, data were filtered so that each trace contained over 10–20 localizations, and we required a raw s.d. per dimension of less than 10–20 nm, much larger than the median localization precision. A $z$ threshold was applied manually based on the apparent plane of the plasma membrane isolate those molecules at or near the membrane. An effective photon frequency at offset (EFO) threshold was applied between 120,000 and 150,000 depending on the first peak of photon emission frequency to remove localizations that might come from multiple fluorescent emitters. The data were then processed using a density-based clustering algorithm that uses two-step DBSCAN clustering followed by an expectation maximization Gaussian mixture model (GMM) fit to assign the 3D position of fluorophores, as previously described[16,57]. All data were analysed using the same parameters so that fair comparisons can be drawn between conditions. The first DBSCAN step was used to preassign localizations, and to identify and remove noise. The DBSCAN parameters in this first step were set to an epsilon of 10 nm and required five neighbours for a core point. The second DBSCAN step was set to epsilon = 7 nm and a minimum of 5 points. The initial GMM fit sigma was set to 5 nm, approximately equal to the localization error. The fluorophore centre positions were estimated as the mean values of the GMM fit, and the error was determined as the s.d. of the localizations within a cluster. After this clustering, an error threshold of 10 nm removed any poorly localized positions. We identified PIEZO trimers as clusters of three fluorophore positions that were isolated from all other detected fluorophore positions by more than 60 nm by subjecting identified molecular positions to DBSCAN clustering with epsilon = 60 nm and 3 minimum points. Next, a nearest-neighbour analysis required each point to have 2 neighbours between 6 and 60 nm, a window that spans the maximum expected range of interblade separations from available PIEZO cryo-EM structures. Clusters of three localizations passing each step was segmented and the average 3D interblade distance was calculated directly from the assigned molecular positions. As interblade distances are computed in 3D, the random orientation of individual channels relative to the imaging axes only rotates the trimer

in space and does not bias the distribution of interblade distances. To verify the accuracy of our clustering algorithm in assigning molecular positions, we calculated the mean position of each trace after EFO thresholding and reanalysed the data using the same density-based clustering algorithm, with minpts set to 1 in both steps. Example data are shown in Extended Data Fig. 2c. Variability between replicates was assessed using a Kruskal–Wallis test with Dunn's post hoc test (Supplementary Table 1).

## MINFLUX data analysis for 3D tracking in live cells

For MINFLUX tracking experiments, raw valid localizations from the final targeting iteration of the 3D tracking sequence (Supplementary Table 3) were exported from Imspector as .mat files, imported into MATLAB and analysed using custom MATLAB analysis code to analyse tracks and obtain diffusion coefficients. For each localization, we used the $x$, $y$ and $z$ coordinates, the trace identifier, the time stamps and the EFO. A 0.7 refractive index correction factor was applied to $z$ values. Localizations were grouped by trace ID to isolate individual trajectories. To ensure robust MSD estimation from well-localized single emitters, tracks were prefiltered using empirical thresholds that were kept fixed across all conditions. A maximum EFO cut-off of 130,000 was applied. The EFO cut-off ensured that only single dyes were imaged, as the photon emission frequency peak of a single dye was determined to be around 75 kHz, and we observed a second emission peak at approximately twice the single-dye emission frequency (-150 kHz) corresponding to two dyes, as described in a previous report that similarly used 3D MINFLUX tracking and an under-labelling strategy[58] (Extended Data Fig. 5a,b). A maximum allowed time gap of 18 ms between successive localizations was imposed, and any trajectory was truncated at the first gap exceeding this threshold, preventing artificial linking of positions across long dark periods. Trajectories were required to have greater than 200 localizations per trajectory so that the MSD could be computed over a sufficiently long timescale.

For each filtered trajectory, we calculated the MSD of the PIEZO molecules undergoing diffusion in three dimensions as a function of lag time $\tau$ with a weighted linear model $MSD(\tau) = 6D\tau$. All MSDs and diffusion coefficients were calculated from 3D displacements. Although the confinement of PIEZO channels to the plasma membrane means that $z$ excursions are small, all motion is explicitly included. Microscopic diffusion coefficients were obtained for each trajectory by fitting $MSD(\tau)$ over 5–50 ms. This window was chosen because, at this timescale, the channel will tend not to interact with diffusional barriers created by membrane–cytoskeletal interactions[36]. Macroscopic diffusion coefficients were obtained using the same model from 50–350 ms. To obtain ensemble diffusion coefficients, all accepted trajectories were pooled, binned into 50 linearly spaced lag-time bins between 0 and 350 ms, and a weighted mean and s.e.m. were computed for each bin using the number of displacement pairs as weights. The ensemble MSD curve was then fit with a weighted linear MSD model. For visualization only (Fig. 2h and Extended Data Fig. 7), accepted 3D trajectories were centred on their centre of mass and the first 1 s of motion was overlaid.

## Electrophysiology analysis of heterologous cells

Cells were transfected with 1 µg DNA 48 h before measurements and plated onto 12-mm poly-D-lysine-coated glass coverslips (Corning, 354086) before recordings. Transfected cells were identified by fluorescence. Whole-cell currents were recorded using a Multiclamp 700A amplifier, Digidata 1550B digitizer and pClamp10.7 software (all from Molecular Devices). To record indentation-evoked currents, data were sampled at 20 kHz and low-pass filtered at 10 kHz. Patch pipette electrodes were pulled using borosilicate glass (34BF150-86-10, Sutter Instruments) and had a resistance of 3–5 MΩ when filled with the pipette solution (see below). The standard extracellular recording solution contained 135 mM NaCl, 3 mM KCl, 1 mM MgCl$_2$, 2.5 mM CaCl$_2$, 10 mM D-glucose, 10 mM HEPES (pH 7.3 with NaOH; 300 ± 5 mOsm was

adjusted with D-mannitol). The pipette solution contained 133 mM CsCl, 5 mM EGTA, 1 mM MgCl$_2$, 1 mM CaCl$_2$, 10 mM HEPES, 4 mM Mg-ATP, 0.4 mM Na$_2$-GTP (pH 7.3 with CsOH; 294 ± 2 mOsm). Cells were mechanically stimulated for 145 ms at a holding potential of −80 mV, using a glass probe heat-polished to a 3–4-µm diameter (34B150-86-10, Sutter Instruments) and driven by a piezoelectric controller and actuator (E625 LVPZT Controller/Amplifier; Physik Instrumente) attached to the micromanipulator with a custom dye-anodized aluminium adapter. The probe was positioned at an 80° angle. The probe was initially positioned at about 2–4 mm from the cell and advanced at 0.5 µm ms$^{-1}$ in 0.5 µm increments. The interstimulus intervals were 20 s. The maximum current ($I_{max}$) was identified from the family of peak current responses to increased membrane indentation. Cells with high access resistance (>20 MΩ) or low seal resistance (<1 GΩ) were excluded from data analysis. Cells that changed their morphology during repetitive poke stimulation were also excluded from data analysis. No series resistance compensation was applied.

To record cell swelling- and shrinking-induced currents, cells were perfused at a rate of 3–4 ml min$^{-1}$ with an iso-osmotic solution containing 45 mM NaCl, 2.4 mM KCl, 1 mM MgCl$_2$, 2 mM CaCl$_2$, 10 mM D-glucose, 10 mM HEPES (pH 7.3 with NaOH; 300 ± 5 mOsm was adjusted with D-mannitol) using the VC-6 valve control system (Warner Instruments). The hypo-osmotic and hyper-osmotic solutions had the same composition, but mannitol was omitted in the hypo-osmotic solution (125 ± 5 mOsm), and the osmolarity of the hyper-osmotic solution was adjusted to 400 mOsm with mannitol. The pipette solution was identical to that described in the previous paragraph. The whole-cell currents were elicited by voltage ramps from −80 mV to 80 mV from the holding potential of −40 mV; the voltage ramps were applied at 0.1 Hz and with a 1 s duration. The interstimulus interval was 20 s. The cell membrane capacitance was estimated using the membrane test of pClamp 10.7, and the amplitude values of recorded current were normalized to the membrane capacitance to obtain the current density. The current–voltage relationship was reconstructed by plotting the current density versus the test voltage. Whole-cell currents were sampled at 10 kHz and low-pass filtered at 2 kHz. All patch-clamp experiments were conducted at the room temperature (20–23 °C). Swelling- or shrinking-induced currents were obtained by subtracting the currents recorded before application of either hypo- or hyper-osmotic solution, respectively, from the maximum current recorded during perfusion with the corresponding solution. Cells that developed membrane blebs during hypo-osmotic challenge were excluded from data analysis.

### Cross-linking MS

Expi293 cells (Thermo Fisher Scientific) were maintained at 37 °C with 8% CO$_2$ in Expi293 medium, shaking at 125 rpm on a rotator with a 19-mm orbit diameter, and were verified to be free of mycoplasma using the using the MycoAlert Mycoplasma Detection Kit (Lonza). Cells were authenticated by the supplier for post-thaw viability, mycoplasma testing and sterility. The cells were cultured to a density of 4 × 10$^6$ cells per ml, exchanged into fresh medium containing 250–500 µM of the click amino acid *trans*-cyclooct-2-en-L-lysine (axial isomer) (SiChem), and three separate flasks were transfected with a 1:1 ratio of (1) PIEZO2 plasmid and tRNA/synthetase expression vector; (2) PIEZO2(ΔIDR5) and tRNA/synthetase expression vector, each using EndoFectin Expi293 transfection reagent (GeneCopeia); or (3) were not transfected. After 48 h of expression, the cells were washed three times at 15-min intervals with medium to wash out excess click amino acid. On the final wash, the cells were resuspended in 10 ml of medium supplemented with 1% BSA and blocked for 5 min. Tetrazine-PEG4-Biotin (Thermo Fisher Scientific) was added to a final concentration of 4 µM and allowed to react for 15 min with occasional mixing. After washing twice in BSA-supplemented medium, the cells were washed with HBSS with 20 mM HEPES pH 8.0. On the final wash, the cells were resuspended

in HBSS + 20 mM HEPES pH 8.0 + 1 mM DSP dithiobis(succinimidylpropionate) cross-linker and incubated for 30 min at room temperature with occasional mixing. The reaction was quenched by adding 200 µl of 1 M Tris pH 8.0 and incubating for 15 min with occasional mixing. The cells were next pelleted by centrifugation, solubilized in ice-cold 25 mM HEPES (pH 7.4), 0.15 M NaCl, 1% C12E9, 0.1% GDN, 1× HALT protease inhibitor and rotated at 4 °C for 1 h. Insoluble cell debris was pelleted at 40,000g for 10 min, and the supernatant was kept on ice. The solubilized proteins were next processed for tandem affinity purification. First, biotinylated proteins were isolated using Pierce Monomeric Avidin Agarose on a polyprep column according to the manufacturer's instructions using a wash buffer (25 mM HEPES (pH 7.4), 0.15 M NaCl, 0.1% C12E9) and eluted with wash buffer containing 2 mM D-biotin. Next, Halo-Tagged proteins were isolated using Magne HaloTag Beads (Promega). After immobilizing for 90 min at room temperature, the beads were washed with RIPA buffer to remove non-specifically bound protein. Cross-linked proteins were released from covalently immobilized Halo-tagged PIEZO molecules using RIPA buffer + 30 mM dithiothreitol at 50 °C for 15 min. The eluate was collected and stored on ice. Eluted proteins were run on a 4–20% polyacrylamide gels with Tris-Glycine buffer for 1 h and silver stained for visualization. For proteomics MS, the proteins were run on the same gel for 20 min and stained with SimplyBlue Safe Stain (Coomassie G-250) (Thermo Fisher Scientific). The lane was cut out from just below the well to ~10 kDa and submitted for nano-flow liquid chromatography coupled with tandem MS (nano-LC−MS/MS) analysis at the Scripps Research Center for Metabolomics and Mass Spectrometry. In brief, the gel was destained, and the proteins were denatured, reduced and alkylated before digestion with trypsin overnight. The peptides were analysed by nano-LC−MS/MS, and the data were searched against the predicted fragment ions from the trypsin digestion of human proteins using the proteomics search engine Mascot (Matrix Science Limited). The analysed results contained proteins identified at the 95% confidence interval. Protein identifiers were cross-referenced against the UniProt database, and each protein ID was appended with an exponentially modified protein abundance index (emPAI). Proteins that were detected in untransfected cells were determined to be background and removed from the PIEZO2 and PIEZO2(ΔIDR5) datasets.

### siRNA-mediated knockdown of candidate scaffolding proteins

*SWELL1*-KO HEK293 cells[16,59] were maintained in Freestyle 293 medium (Thermo Fisher Scientific) at 37 °C with 8% CO$_2$, shaking at 125 RPM on a rotator with a 19-mm orbit diameter, and were verified to be free of mycoplasma using the using the MycoAlert Mycoplasma Detection Kit (Lonza). Before transfection, 3 ml of cells were grown in a 30 mm diameter uncoated Petri dish to a density of 1 × 10$^6$ cells per ml. Then, 1.5 µg of mPiezo2-IRES-mNG was co-transfected with 2 µl of a 40 µM stock (80 pmol total) of ON-TARGETplus SMARTpool siRNAs (Dharmacon) targeted against the mRNAs of candidate scaffolding proteins using EndoFectin Expi293 transfection reagent (GeneCopeia). After 48 h, cells were plated onto 10 mm poly-D-lysine-coated coverslips, allowed to settle for 1 h, and assessed using patch-clamp electrophysiology.

### Clonal *FLNB*-KO cells

Clonal *FLNB*-KO cells were created using *SWELL1*-KO HEK293 cells[16,59] using the EditCo Bio Gene Knockout Kit. *SWELL1*-KO cells were authenticated as previously described: successful KO of *SWELL1* genes was determined by PCR genotyping and Sanger sequencing targeted regions for frameshift mutations, and verified by MS analysis[16,59]. Three sgRNAs were designed against exon 8 of the human *FLNB* gene (NCBI: NM_001164317.2). The sgRNAs were precomplexed with spCas9-2NLS (Synthego) and 1.5 × 10$^6$ cells were nucleofected using a Lonza 4D Nucleofector System, the P3 Primary Cell Kit S and a nucleocuvette strip (Lonza) using the default HEK293 electroporation program. Cells were recovered for 10 min in Freestyle 293 medium (Thermo Fisher

Scientific) and then grown for 3 days with shaking at 125 rpm. Single cells were isolated using a Propel Bigfoot flow cytometer (Thermo Fisher Scientific) at the Scripps Research Flow Cytometry Core in the wells of three 96-well plates containing DMEM + 10% FBS + 1× penicillin–streptomycin and grown for 3 weeks at 37 °C, 5% $CO_2$ to form clonal colonies. After visible colonies were formed, cells were dissociated using Tryple Express (Thermo Fisher Scientific), quenched with DMEM + 10% FBS, and 50% of the dissociated mixture was plated into a single well of a 12-well plate to expand further. Meanwhile, the remaining mixture (containing around 300,000 cells) was centrifuged and genomic DNA was isolated using the QuickExtract DNA Extraction Solution (BioSearch Technologies). A 1.245 kb fragment of genomic DNA from each colony was amplified using two PCR primers using Q5 DNA Polymerase (NEB), isolated with a PCR purification column (Zymo Research) and Sanger sequenced using a nested sequencing primer (Genewiz). Sequencing traces were analysed using the EditCo ICE analysis tool. A single *SWELL1*-KO/*FLNB*-KO clone was chosen that contained a homozygous deletion of 52 bp in exon 8 of *FLNB* (chromosome 3: 58098750–58098801 of the GRCh38/hg38 reference assembly), resulting in a frameshift. The clone was expanded and maintained in adherent culture in DMEM + 10% FBS + 1× penicillin–streptomycin in an incubator at 37 °C, 5% $CO_2$.

### *Flnb* knockdown in PtK2 cells

Four arrayed Dicer-substrate interfering RNAs (DsiRNAs) (IDT) were designed against the *P. tridactylus Flnb* sequence (Supplementary Table 1) using the published PtK2 cell transcriptome as a guide[42]. A commercial non-targeting dsiRNA was included as a control (IDT 51-01-19-08). PtK2 cells were plated into a six-well plate containing 3 ml of medium 1–3 h before transfection. Each well was transfected with 20 pmol of each of the 4 dsiRNAs in 300 µl Optimem with 7.5 µl TransfeX transfection reagent (ATCC). After 48 h, the cells were split and plated onto Matrigel-coated coverslips with embedded gold fiducials, exchanged into a medium containing 250 µM of the click amino acid *trans*-cyclooct-2-en-L-lysine (axial isomer) (SiChem) and transfected a second time with identical amounts of dsiRNA, with the addition of 1.5 µg of an equimolar ratio of the mPiezo2 and the tRNA/tRNA synthetase expression plasmids. The cells were allowed to express for an additional 48 h and then prepared for MINFLUX imaging as described above. Knockdown efficiency was verified using qPCR.

### Mice

All experiments were performed under the policies and recommendations of the International Association for the Study of Pain and approved by the Scripps Research Animal Care and Use Committee. Mice were kept in standard housing under a 12 h–12 h light–dark cycle at 22 °C with humidity between 30% and 80% (not controlled). Mice were kept on pelleted paper bedding and provided with paper square nestlets and polyvinyl chloride pipe enrichment with ad libitum access to food and water. PCR genotyping was performed from tail snip DNA samples using Transnetyx. All mice received metal identification tags on their ears at 18–30 days old. After weaning (21–30 days old), mice were co-housed in groups of 2–5 littermates of the same sex. Animal sample sizes were based on similar studies in the literature[6,60].

### Immunohistochemical co-localization of PIEZO2–smFlag and FLNB in mouse skin

Two male *Piezo2^smFP-Flag/smFP-Flag* mice[60] (*Piezo2em1.1Ddg/J*, CD-1 genetic background, Jackson Laboratories, 039935, a gift from D. Ginty, one at 2.5 weeks of age and one at 4 weeks of age) and two male CD-1 mice (one at 6 weeks of age and one at 4 weeks of age) were used for hairy skin immunohistochemistry[60]. One 4-week-old male *Piezo2^smFP-Flag/smFP-Flag* mouse and one 4-week-old CD-1 control mouse were used for glabrous skin immunohistochemistry. In brief, mice were euthanized with isoflurane and killed by cervical dislocation. The dorsal surface

of the mouse was dehaired with depilatory cream (Nair Cocoa Butter) for 3 min, the skin was thoroughly rinsed with industrial water and gentle manual massaging for 30 s and patted dry with paper towels. The dorsal back skin was rapidly collected, the epidermis was pinned down onto Styrofoam and scraped with a surgical scalpel to remove subcutaneous fat, taking care to note the rostrocaudal axis. The plantar (glabrous) surface of the hind paw, not including the digit tips, was collected using spring scissors, with the underlying tissues removed. White cardstock was pressed onto the dermis to flatten the samples, and the dorsal back skin was trimmed to approximately 1 × 1 cm. Skin samples were drop-fixed in freshly prepared and prechilled 1% paraformaldehyde in PBS pH 7.4 for 2 h on ice. Skin samples were rinsed twice in cold PBS, then transferred into ice-cold 30% sucrose in PBS and incubated for 18 h at 4 °C until the tissues sank. The samples were briefly washed in ice-cold OCT medium (Sakura Finetech, 4583) and embedded in OCT medium in cryomolds (with the cardstock) on crushed dry ice. All tissues were sectioned dermis-first at 25 µm at −20 °C onto gelatin-coated slides (FD NeuroTechnolgies, PO101). Hairy skin was sectioned normal to the rostrocaudal axis, and glabrous skin was sectioned normal to the proximodistal axis, beginning with the distal portion. Slides were air-dried at room temperature for 1 h and a hydrophobic barrier was drawn (ImmEdge Vector Laboratories H-4000) around the tissue sections. The slides were rehydrated in 200 ml PBS to remove OCT and cardstock and, from then on, all washes were performed with a volume of 40 ml. All washes were performed for 10 min. The samples were washed in PBS and then blocked for 2 h at room temperature in 5% normal goat serum (Life Technologies, PCN5000) in 0.1% PBST. Block was aspirated and primary antibody was applied as follows in blocking buffer for 48 h at 4 °C: 1:500 polyclonal guinea pig anti-Flag[60] (a gift from D. Ginty), 1:500 polyclonal rabbit anti-FLNB (Thermo Fisher Scientific, PA5-52098) and 1:1,000 polyclonal chicken anti-NFH (Abcam, ab4680). The slides were washed three times in PBST followed by PBS and incubated in the following highly cross-adsorbed secondary antibodies for 24 h at 4 °C: for confocal imaging: 1:2,000 goat anti-guinea pig Alexa Fluor 594 (Life Technologies, A11076), 1:2,000 goat anti-rabbit Alexa Fluor 647 (Life Technologies, A21245), 1:2,000 goat anti-chicken Alexa Fluor 488 (Life Technologies, A32931); for STED imaging: 1:2,000 goat anti-guinea pig STAR RED (Abberior, STRED-1006), 1:750 goat anti-rabbit STAR ORANGE (Abberior, STORANGE-1002). A no-primary control was always performed. The slides were washed three times in 0.1% PBST, then PBS, and then mounted in SlowFade Diamond (Life Technologies, S36967). The slides were sealed with nail polish, dried for 2 h, stored at 4 °C, and imaged on the Nikon AX confocal microscope or Abberior Instruments Facility Line 3D STED microscope.

### smFISH

Mouse dorsal root ganglia were dissected fresh from two adult C57BL6/J male mice, embedded in optimal cutting temperature compound (OCT, Sakura), and flash-frozen in liquid nitrogen. The protocols for the RNAscope Multiplex Fluorescent Reagent Kit V2 (ACDBio, 323100) and tyramide signal amplification dyes (Perkin Elmer) were followed exactly according to the manufacturers' instructions. Protease IV was applied for 22 min. Probes (all from ACDBio) for mouse *Flnb* (572481), *Ntrk2* (423611-C2), *Ntrk3* (423621-C2) and *Piezo2* (400191-C3) were applied to detect transcript. Slides were imaged on a Nikon AX confocal microscope using a ×16 water-immersion objective with Nyquist zoom. Cell borders were drawn around highly expressed marker transcript (*Ntrk2* and *Ntrk3*) signals to define individual cells.

### Confocal imaging and data analysis

Images were acquired on the Nikon AX confocal microscope with NIS Elements software and the image settings (laser power, gain, resolution, pixel dwell time, objective and pixel dimension settings) were kept the same for all conditions. Meissner corpuscles were imaged with a Nikon

×60/1.4 NA oil-immersion objective and hair follicle lanceolates were imaged with a Nikon ×100/1.42 NA oil-immersion objective. Images were analysed in Fiji. For all images, the brightness and contrast adjustments were applied uniformly to the entire image.

## STED imaging and data analysis

STED imaging was performed on an Abberior Instruments Facility Line 3D STED microscope on an Olympus IX83 microscope body. The excitation lasers and STED depletion lasers were autoaligned with a fluorescent bead fiducial sample. The samples were imaged in 2D mode with a ×60/1.42 NA oil-immersion Olympus objective with 561 and 640 nm excitation lasers with a 775 nm depletion laser with fluorescent lifetime imaging (TIMEBOW) enabled for all of the experiments. All of the images were acquired with identical excitation power, STED depletion power, pinhole diameter and line accumulations so that the fluorescence intensities were comparable across samples. STED images were acquired at 20 nm per pixel. Co-registered confocal and STED images were acquired in two channels (STAR RED for PIEZO2-smFlag, STAR ORANGE for FLNB) for each field of view. Single-label controls were used to confirm detection bandwidths and verify negligible bleed-through between channels. STED data were analysed and deconvolved in Lightbox 2025 software (v.2024.48.21878-gc86bbd647c) using lifetime-based PHASOR deconvolution. Deconvolution was used to improve lateral resolution and signal-to-noise. The deconvolution model incorporated the refractive index of the mounting medium (1.420), the measured axial distance from the coverslip to correct for depth-dependent point spread function changes and the lifetime information from the dyes, which helps to distinguish fluorophore signal from the background compared with intensity-only deconvolution. All images were processed with the same deconvolution and background subtraction parameters. In the Lightbox program, deconvolution parameters were set to 50 iterations and a sharpness value of 10, and PHASOR background weights of 1.50 were applied to the STAR RED and STAR ORANGE channels. The background was removed using a rolling-ball subtraction (kernel size = 20, weight = 1.0). These values were empirically chosen to yield stable FWHM values of approximately 60–80 nm without oversharpening or ringing artifacts.

The deconvolved images were exported to Fiji (v.2.16.0/1.54p) for segmentation and co-localization analysis. To restrict measurements to individual lanceolate endings, we generated a binary mask from the confocal STAR RED channel, which provides a continuous representation of PIEZO2-positive lanceolate endings (Extended Data Fig. 9e). The confocal STAR RED (PIEZO2-smFlag) image was Gaussian blurred with a radius of 10 pixels to smooth local intensity fluctuations and then thresholded using a fixed intensity threshold applied identically to all images. This value was chosen from the intensity histograms to include lanceolate signal and exclude background. The FLNB channel was hidden during ROI generation to avoid bias. These ROIs were applied to the deconvolved STED images of both channels. Within each lanceolate ROI, co-localization between PIEZO2-smFlag and FLNB was quantified on the deconvolved STED images by pixel intensity correlation using the Coloc2 plugin in Fiji with a point-spread-function diameter of 4.0 pixels, chosen based on the measured STED FWHM. No additional intensity thresholds were applied within the ROIs. We report Pearson's correlation coefficient $r$ and Spearman's rank correlation coefficient $\rho$ per lanceolate ending. FWHM was measured from the deconvolved STED images in Fiji using the 'fwhm_on_spots' jython-fiji macro (https://github.com/sommerc/spots_fwhm).

## DRG neurons

Isolation and culture of mouse DRG neurons from wild type C57BL/6J mice aged 3–5 months were performed as previously described[1,6]. In brief, DRGs were dissected and incubated for 1 h at 37 °C in serum-free medium containing 1.25% collagenase IV (Life Technologies), followed by incubation with 1 U ml⁻¹ papain (Thermo Fisher Scientific) for 30 min at 37 °C. Cells were then triturated and transferred into complete growth medium (Ham's F12/DMEM + 10% FBS supplemented with the following growth factors (from Gibco): 50 ng ml⁻¹ GDNF, 100 ng ml⁻¹ NGF, 50 ng ml⁻¹ NT-4, 50 ng ml⁻¹ NT-3, 50 ng ml⁻¹ BDNF and 10 μM cytosine arabinoside (AraC)) and plated onto laminin-coated poly-D-lysine coverslips (Corning). Cells were allowed to adhere for 1–2 h before the addition of extra complete medium. For DsiRNA knockdown experiments, cells were nucleofected using the Amaxa P3 Primary Cell 4D-Nucleofector X Kit S (Lonza) as previously described[1,8]. Four arrayed DsiRNAs (IDT) designed against mouse *Flnb* mRNA were used to knockdown *Flnb* and a non-targeting dsiRNA was included as a control (IDT 51-01-19-08). In total, 120 pmol of DsiRNA and 400 ng of pmaxGFP vector (Lonza) were nucleofected per reaction. After nucleofection, cells were allowed to recover in serum-free medium for 10 min at 37 °C and then plated with complete medium containing growth factors, except without the addition of AraC. Cells were allowed to adhere for 1–2 h before the addition of extra complete medium. Measurement of swelling-induced current was performed as described for heterologous cells, except the extracellular solution contained 30 μM DCPIB to block SWELL1-mediated chloride currents. Patch-clamp recordings were performed on the fourth and fifth day after nucleofection. Only large-diameter cells (>70 μm) with neuronal morphology were patched. The extracellular iso-osmotic and hypo-osmotic solutions, intracellular recording solution and other conditions were identical to those used for heterologous expression experiments (see above), with the exception that cells were patched in the standard extracellular recording solution followed by perfusion with DCPIB (30 μM)-containing iso-osmotic solution for at least 5 min before 2 min perfusion with the hypo-osmotic solution (also containing 30 μM DCPIB). The holding potential between voltage ramps was −40 mV. Cells that exhibited a sudden increase in current of several hundred pA or several nA after the application of hypotonic solution, without recovering after reintroduction to the iso-osmotic solution, were excluded from data analysis due to the potential compromise of the gigaseal stability or plasma membrane integrity during cell swelling.

## Structural models

Structural models from single-particle cryo-EM for PIEZO1 (6B3R) and PIEZO2 (6KG7) were obtained from the Protein Data Bank (PDB). The PIEZO2 cryo-EM structure lacks the extracellular loop containing the tagging location at amino acid 105, so an AlphaFold III model was generated for a monomer of mouse PIEZO2, and the last PIEZO repeat domain was superposed onto the equivalent domain of the 6KG7 cryo-EM structure in UCSF Chimera software. Interblade distances were measured using amino acid Gln105 of this model.

## Data visualization and statistical tests

Data were visualized and statistical tests performed in MATLAB (MathWorks) and Prism (GraphPad) software. Molecular structures were visualized in MolStar Viewer (https://molstar.org/viewer/) and Chimera (UCSF) software. DNA and mRNA sequences were designed and analysed in SnapGene analysis software (Dotmatics).

## Reporting summary

Further information on research design is available in the Nature Portfolio Reporting Summary linked to this article.

## Data availability

Published protein structures were obtained from the RSCB Protein Data Bank (6B3R (PIEZO1) and 6KG7 (PIEZO2)). Protein sequences were obtained from UniprotKB. AlphaFold III models were generated with the Google DeepMind AlphaFold Server. Raw data are available at Zenodo[61] (https://doi.org/10.5281/zenodo.17644763). All reagents that are not

commercially available are available from the corresponding authors on reasonable request. Source data are provided with this paper.

## Code availability

Custom MATLAB code for analysis of MINFLUX structural and tracking data are available at GitHub (https://github.com/PatapoutianLab/MINFLUX_Localization_and_Tracking_Analysis) and Zenodo[62] (https://doi.org/10.5281/zenodo.17625937).

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

**Acknowledgements** We thank J. Matthias for help with MINFLUX instrument and software troubleshooting; D. Ginty for sharing *Piezo2*[smFP-Flag/smFP-Flag] mice and anti-Flag antibodies; K. Spencer and S. Henderson for assistance with imaging; the staff at the Scripps Research Department of Animal Resources for resources and support services; and the members of the Patapoutian laboratory for feedback and discussions. MINFLUX and STED imaging was performed at and supported by the Scripps Research Core Microscopy Facility. MS was performed at the Scripps Center for Metabolomics and Mass Spectrometry. This work is supported by NIH grants K99 GM155547 (to E.M.M.) and R35 NS105067 (to A.P.). E.M.M. was supported by a George E. Hewitt Foundation for Medical Research postdoctoral fellowship and The Warren Alpert Distinguished Scholar Award in Neuroscience. A.P. is a Howard Hughes Medical Institute Investigator.

**Author contributions** E.M.M. conceived and designed the project, performed MINFLUX, STED and confocal imaging, molecular biology experiments, cross-linking MS, wrote the MINFLUX analysis and tracking code, analysed and interpreted all data, prepared figures and wrote the manuscript. O.Y. performed electrophysiology experiments. R.Z.H. performed skin immunostaining, smFISH and confocal imaging. A.K.K. assisted with cross-linking MS. A.P. contributed to project design and supervised the project. All of authors discussed results and contributed to manuscript editing.

**Competing interests** The authors declare no competing interests.

**Additional information**
**Correspondence and requests for materials** should be addressed to Eric M. Mulhall or Ardem Patapoutian.

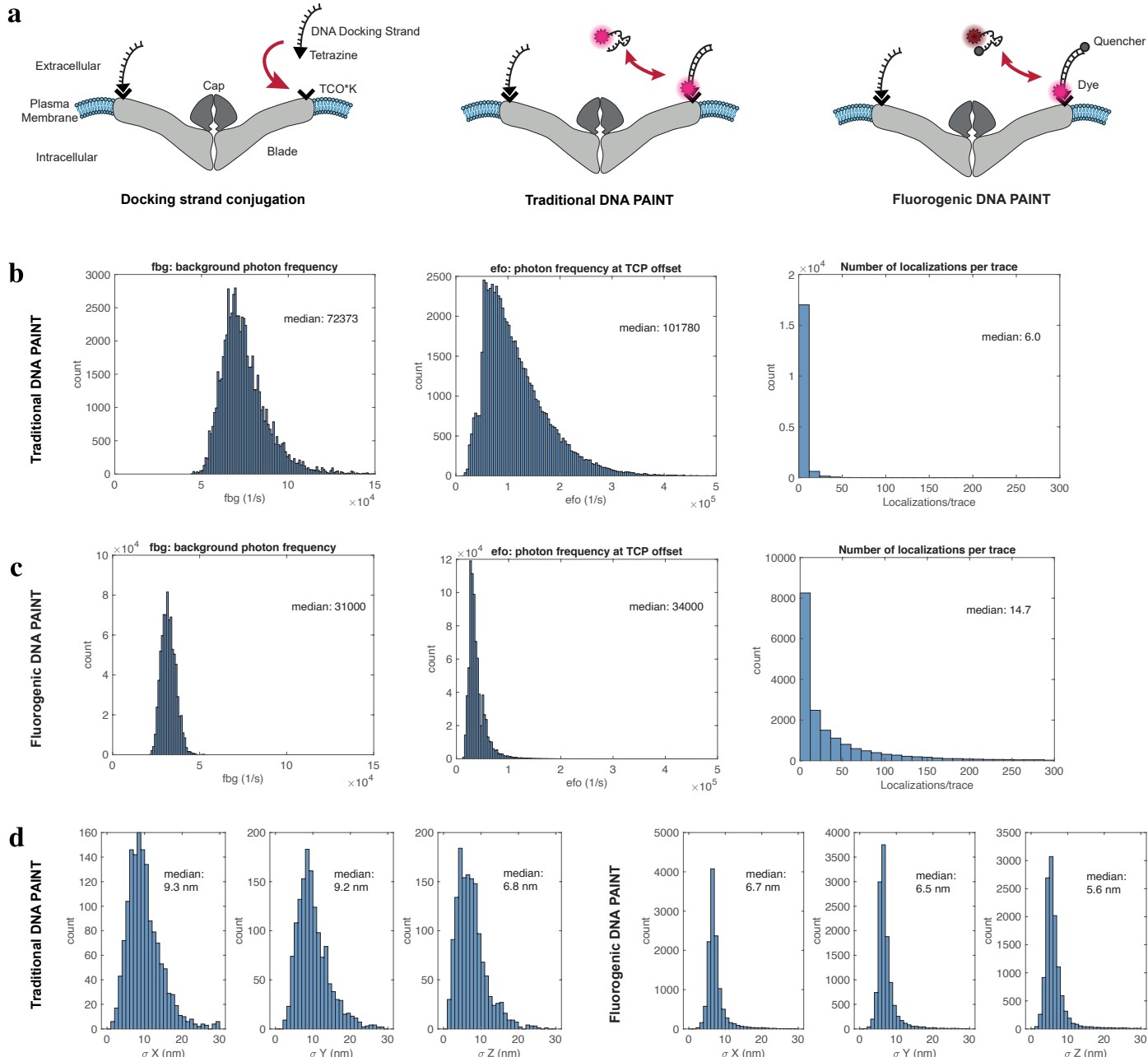

**Extended Data Fig. 1 | Fluorogenic DNA PAINT with MINFLUX nanoscopy.**
**a**, Schematic of DNA PAINT with traditional and fluorogenic DNA PAINT.
A DNA docking strand is conjugated to the distal extracellular blade of PIEZO1
or PIEZO2. Traditional DNA PAINT imaging strands remain fluorescent in
solution when not bound to the docking strand, but fluorogenic imaging strands
are quenched, reducing background. **b-c**, PtK2 cells expressing mPIEZO1 TCO*K
103 imaged with 5 nM imaging strand, with and without a fluorescence quencher
on the imaging strand. Imaging parameters were exactly the same (5% excitation
power, 0.60 AU pinhole). Left, determined fluorescence background photon
emission frequency in the sample. Middle, effective photon emission frequency
at offset TCP position for the final iteration of a localization. Right, Number of
localizations collected per trace. n = 2 cells per condition. **d**, Standard deviation
per trace calculated in each dimension for unfiltered traces containing >10
localizations.

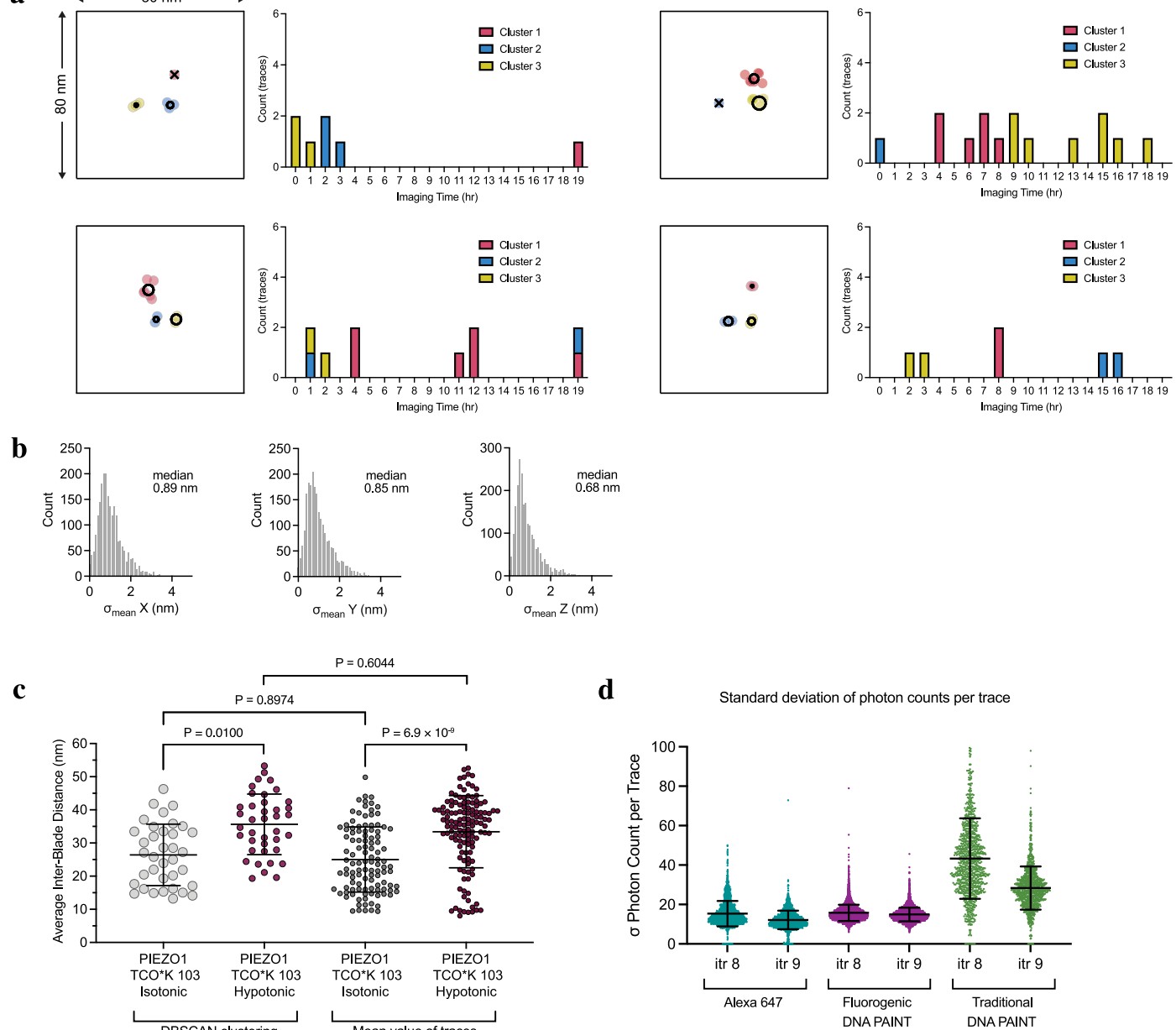

**Extended Data Fig. 2 | MINFLUX imaging stability with fluorogenic DNA PAINT. a**, Representative examples of triple-labelled PIEZO1 molecules imaged with 3D MINFLUX over a total acquisition period of 19 h. The left panels show the mean positions of individual traces (coloured circles) within identified trimeric clusters. The right panels show histograms of the time elapsed since the start of the experiment for each trace in the corresponding cluster. **b**, Localization precision of independent detection events within any cluster for clusters with >1 trace. **c**, Comparison of measured inter-blade distances for triple-labelled PIEZO1 molecules when clustering localizations versus the mean position of each trace. In both cases, data were subjected to the same density-based clustering

algorithm. Statistical test: two-sided Kolmogorov–Smirnov (KS) test. PIEZO1 Isotonic DBSCAN: mean = 26.4 nm, n = 36 molecules, n = 4 cells; PIEZO1 Hypotonic DBSCAN: mean = 35.6 nm, n = 39 molecules, n = 3 cells; PIEZO1 Isotonic mean values: mean = 25.0 nm, n = 110 molecules, n = 4 cells; PIEZO1 Hypotonic mean values: mean = 33.4 nm, n = 129 molecules, n = 3 cells. **d**, Representative standard deviation of photon counts per trace for PIEZO1 molecules imaged with direct Alexa 647 dye conjugation (itr 8: mean = 15.3, itr 9: mean = 12.1; n = 2014 traces), fluorogenic DNA PAINT (itr 8: mean = 15.8, itr 9: mean = 14.9; n = 7254 traces), and traditional DNA PAINT (itr 8: mean = 43.2, itr 9: mean = 28.4; n = 1013 traces). Error bars: mean ± SD (**c**,**d**).

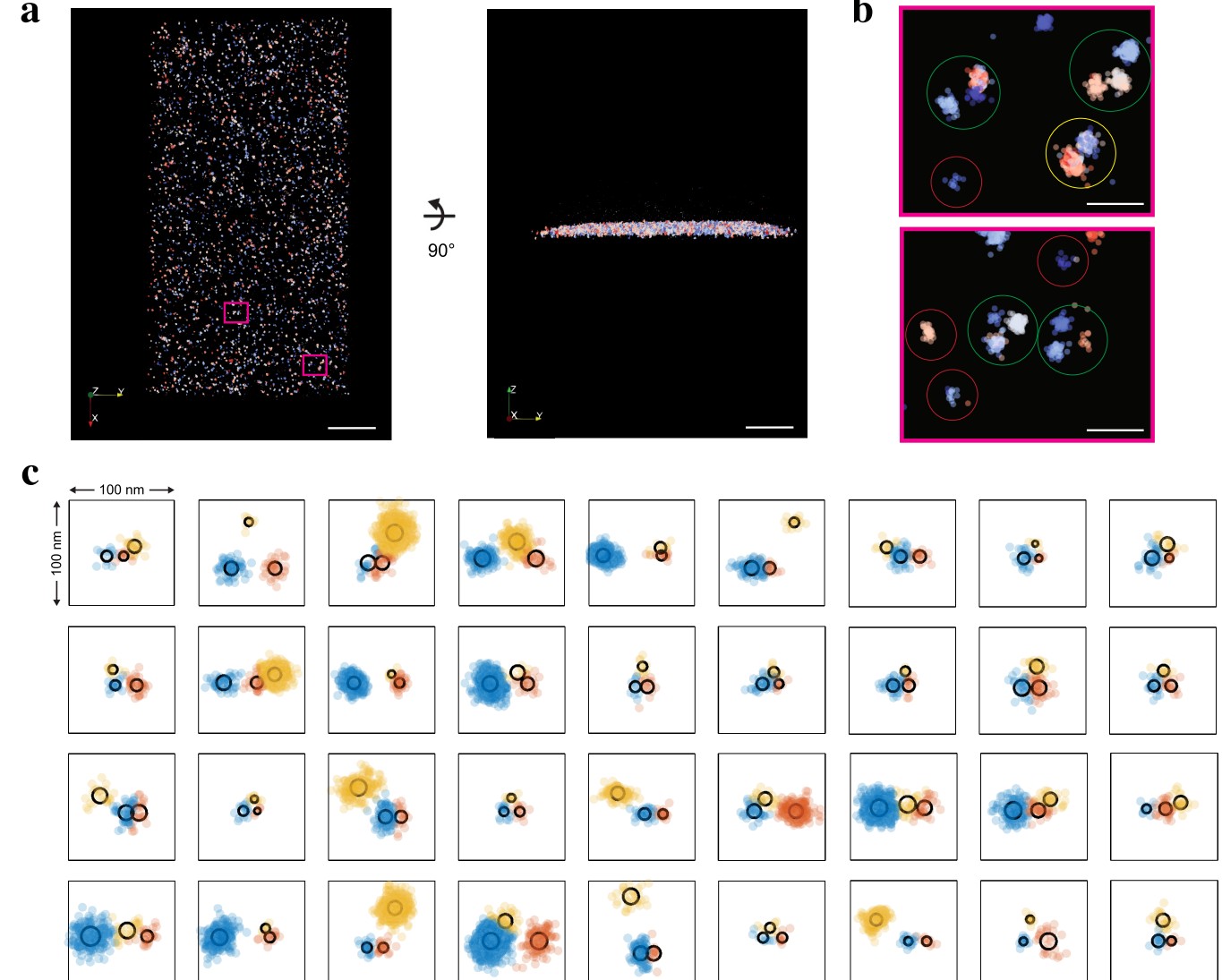

**Extended Data Fig. 3 | Representative MINFLUX data and particle segmentation. a**, A representative rendered MINFLUX image of valid final localizations (5-nm spheres coloured by trace ID) from a PtK2 cell membrane expressing mPIEZO1 TCO*K 103 (from 4 cells) and imaged with 5 nM fluorogenic imager strand. Scale bar = 1 μm. **b**, Zoomed-in renderings of localizations (5-nm spheres) of the magenta boxes in **a** showing single-labelled (red), double-labelled (yellow) and triple-labelled (green) PIEZO1 molecules (scale bar, 50 nm). **c**, PIEZO1 trimers identified by the clustering algorithm in a 100×100 nm bounding box for the PIEZO1 TCO*K 103 Isotonic condition in Fig. 1d. Raw localizations (6-nm spheres) coloured by DBSCAN cluster ID. The centre positions were determined by a 3D GMM fit (black circles). Segmented particles are viewed top-down from a flat plane fit between each of the three centre positions.

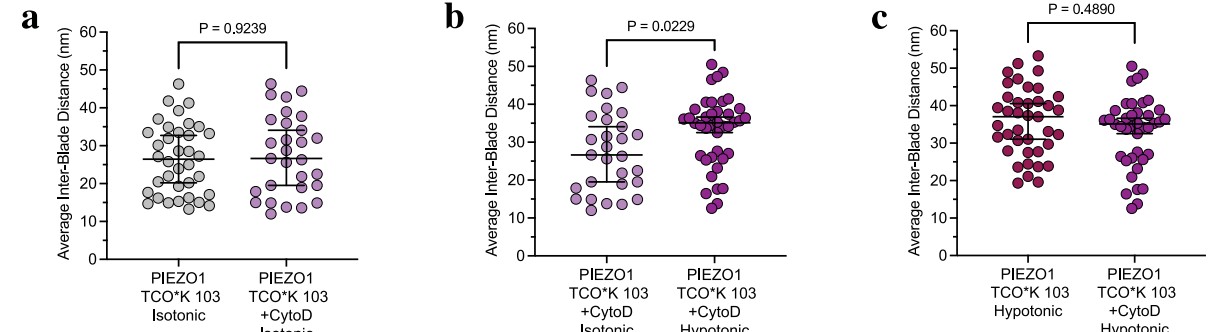

**Extended Data Fig. 4 | Actin disruption with Cytochalasin D does not significantly alter PIEZO1 structural mechanics. a**, Disruption of the actin cytoskeleton with 10 μM cytochalasin D (magenta circles; median = 26.6 nm; n = 29 molecules, n = 3 cells) does not alter the resting conformation of PIEZO1 relative to untreated cells (grey circles; data from Fig. 1b). **b**, Expansive membrane stretch from hypo-osmotic expansion in cytochalasin D treated cells results in blade expansion in PIEZO1 (dark magenta circles; median = 35.1 nm; n = 40 molecules, n = 3 cells). **c**, Comparison of blade expansion from hypo-osmotic expansion (dark red circles; data from Fig. 1d). All statistical tests performed with a two-sided KS test, error bars shown as median and 95% CI.

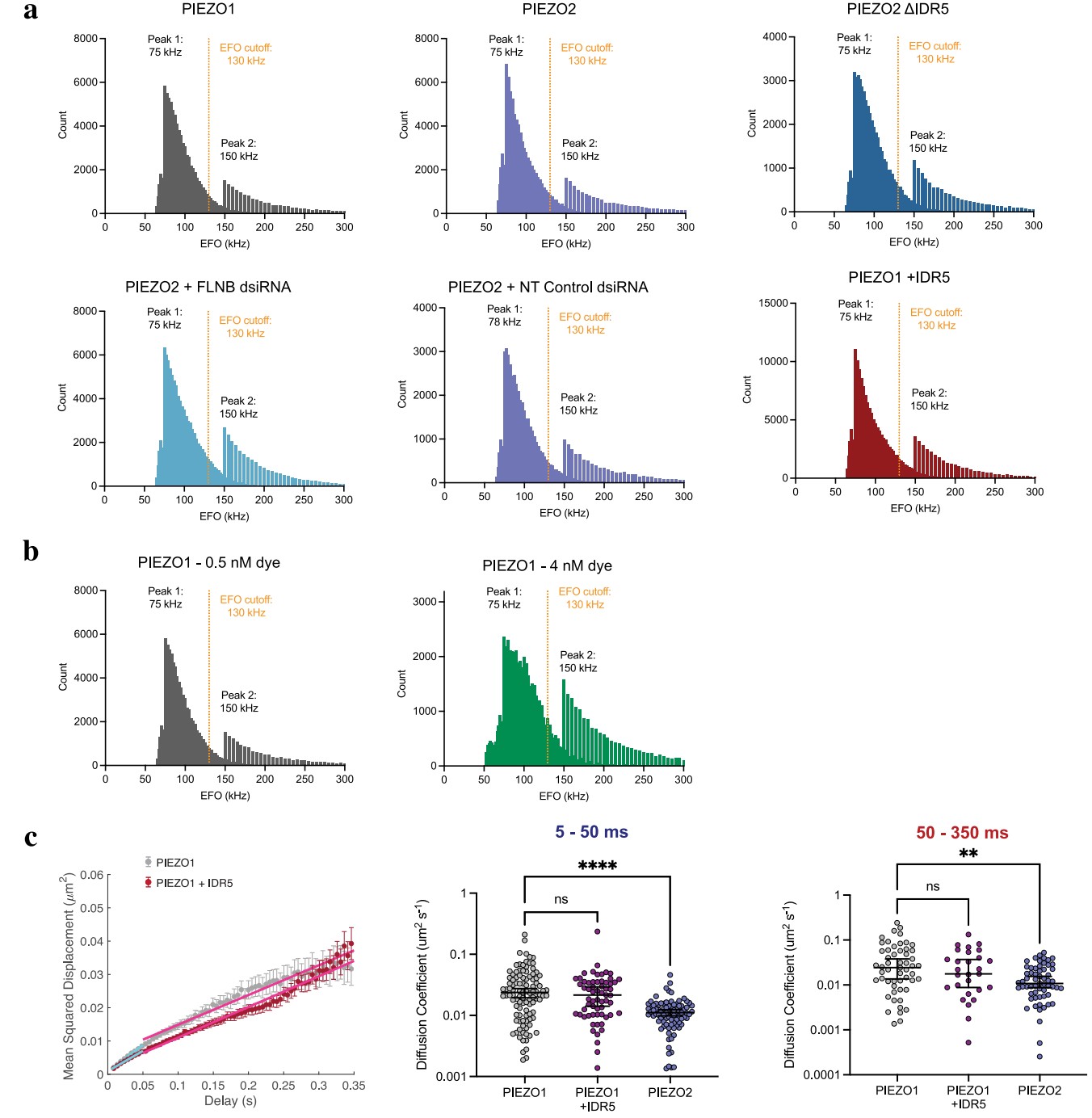

**Extended Data Fig. 5 | MINFLUX single-molecule tracking: data filtering and PIEZO1-IDR5 chimera. a**, Histograms of EFO (effective photon frequency measured at TCP offset) for each MINFLUX tracking experiment. The photon emission peaks corresponding to one dye (Peak 1) or two dyes (Peak 2) are noted. An upper EFO threshold of 150 kHz was applied to each dataset to eliminate signal arising from multiple dyes. **b**, Histograms of EFO for PIEZO1 measured with 0.5 nM dye (from Extended Data Fig. 5a) and 4 nM dye. Increased labelling concentration led to a disproportionate increase in the higher-frequency EFO peak (~150 kHz) relative to the single-molecule peak (~75 kHz), consistent with a higher fraction of multi-molecule events at higher labelling densities. **c**, Results from single-molecule tracking of the PIEZO1 + IDR5 chimera in which the fifth IDR of PIEZO1 (amino acids 551–575) was replaced with the corresponding IDR5 region from PIEZO2 (amino acids 620–672). Left, Ensemble Mean Squared

Displacement (MSD) against time for trajectories containing at least 200 localizations fit between 5–50 ms (PIEZO1: D = 0.0218 $\mu m^2 s^{-1}$, PIEZO1 + IDR5: D = 0.0204 $\mu m^2 s^{-1}$) and between 50–350 ms (PIEZO1: D = 0.0197 $\mu m^2 s^{-1}$, PIEZO1 + IDR5: D = 0.0153 $\mu m^2 s^{-1}$). (n, trajectories: PIEZO1: n = 55, PIEZO1 + IDR5: n = 29). Error bars: mean ± SEM. Middle, microscale diffusion coefficients for PIEZO1 (grey circles; median = 0.024 $\mu m^2 s^{-1}$, n = 104), PIEZO1 + IDR5 (magenta circles; median = 0.022 $\mu m^2 s^{-1}$, n = 62), and PIEZO2 (blue circles; median = 0.011 $\mu m^2 s^{-1}$, n = 87). KW test: ****$P$ = 2.0×10$^{-9}$. Error bars: median ± 95% CI. Right, macroscale diffusion coefficients for PIEZO1 (grey circles; median = 0.024 $\mu m^2 s^{-1}$, n = 55), PIEZO1 + IDR5 (magenta circles; median = 0.018 $\mu m^2 s^{-1}$, n = 29), and PIEZO2 (blue circles; median = 0.011 $\mu m^2 s^{-1}$, n = 61). KW test: **$P$ = 0.0027. Error bars: median ± 95% CI. All statistical tests are two-sided.

**a**

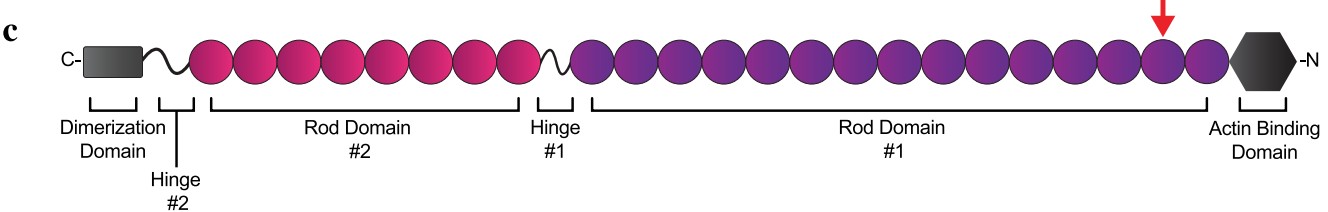

```
GTAGGAGCTGGTGTGGGTGACATTGGTGTGGAGGTGGAAGATCCCCAGGGGAAGAACACCGTGGAGTTGCTCGTGGAAGACAAAGGAAACCAGGTGTATCGATGTGTGTACAAACCCATG
CATCCTCGACCACACCCACTGTAACCACACCTCCACCTTCTAGGGGTCCCCTTCTTGTGGCACCTCAACGAGCACCTTCTGTTTCCTTTGGTCCACATAGCTACACACATGTTTGGGTAC
```

```
   1        5         10        15        20        25        30        35
   A G V G D I G V E V E D P Q G K N T V E L L V E D K G N Q V Y R C V Y K P M
```

Exon 8

FLNB KO Indel

sgRNA 1                                                                sgRNA 2

```
CAGCCTGGCCCTCACGTGGTCAAGATCTTCTTTGCTGGGGACACTATTCCTAAGAGTCCCTTCGTTGTGCAGGTTGGGGAAGGTGAGTGCTGGGCTGCTGGCCACATGTGCTTCTCATAG
GTCGGACCGGGAGTGCACCAGTTCTAGAAGAAACGACCCCTGTGATAAGGATTCTCAGGGAAGCAACACGTCCAACCCCTTCCACTCACGACCCGACGACCGGTGTACACGAAGAGTATC
```

```
  40        45        50        55        60        65
  Q P G P H V V K I F F A G D T I P K S P F V V Q V G E
```

Exon 8

sgRNA3

**b**

WT FLNB: -382-GAGVGDIGVEVEDPQGKNTVELLVEDKGNQVYRCVYKPMQPGPHVVKIFFAGDTIPKSPFVVQVGEACNPNACRASGRGLQPKGVRIRETTDFKVD-479-

Truncated FLNB: -382-GAGVGDIGVEVED^RIDVCTNPCSLALTWSRSSLLGTLFLRVPSLCRLGKPAIQMPAGPVAEAYNPKASVSGRPQISRLTPKLQEVGSSV*P*RV...

**c**

C-  Dimerization Domain  Hinge #2  Rod Domain #2  Hinge #1  Rod Domain #1  Actin Binding Domain  -N

**Extended Data Fig. 6 | *FLNB* and *SWELL1* knockout HEK293 cells. a**, Genomic DNA sequence surrounding and including exon 8 of human FLNB. The position of the indel in the *SWELL1*-KO + *FLNB* KO HEK293 cells (magenta) is shown within the exon (blue). Arrayed CRISPR sgRNA hybridization locations are shown in grey. **b**, Human FLNB protein sequences encoded by Refseq cDNA NM_001457.4 surrounding the indel, starting after amino acid 382. Indel location is shown with a red caret and introduced stop codons are shown as red asterisks. **c**, A schematic of the domains in FLNB with the red arrowhead denoting the location of truncation in the *FLNB* KO cells.

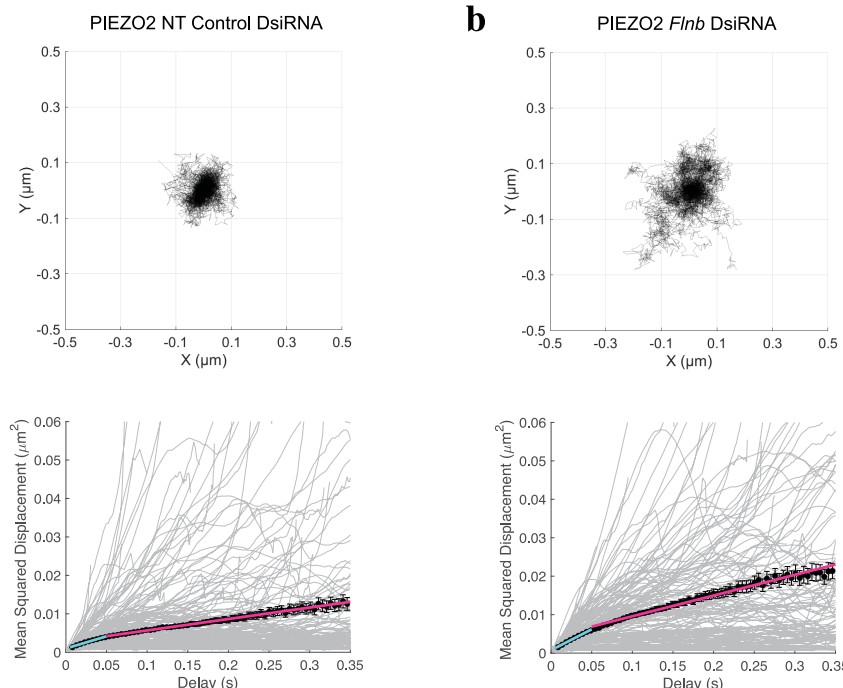

**Extended Data Fig. 7 | Single-molecule tracks and individual trajectories for FLNB DsiRNA knockdown. a**, Top, overlaid trajectories of PIEZO molecules from single molecule tracking with MINFLUX aligned by centre of mass for trajectories lasting at least 1 s for PtK2 cells expressing PIEZO2 with non-targeting DsiRNA. Bottom, Mean Squared Displacement against time for trajectories containing at least 200 localizations for the trajectories shown in Fig. 3g. n = 50 trajectories. **b**, Top, overlaid single-molecule trajectories for cells expressing PIEZO2 with *Flnb* DsiRNA. Bottom, Mean Squared Displacement against time for trajectories shown in Fig. 3g. n = 81 trajectories. Error bars: mean ± SEM (**a**,**b**).

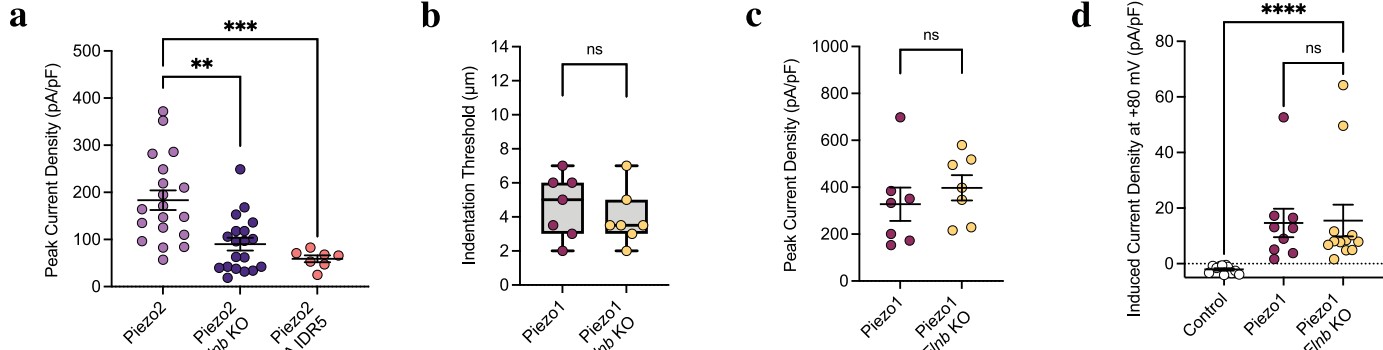

**Extended Data Fig. 8 | Mechanically Evoked Currents from PIEZO2 and PIEZO1 in FLNB Knockout Cells. a**, Quantification of the peak indentation-evoked current for the data in Fig. 3j. Piezo2: 183.5 ± 42.1 pA/pF, n = 19 cells; Piezo2 *Flnb* KO: 90.2 ± 20.7 pA/pF, n = 19 cells; Piezo2 ΔIDR5: 59.0 ± 22.3 pA/pF, n = 7 cells. KW test: **\*\*P* = 0.0015, \*\*\*P* = 0.0010. **b**, Indentation threshold to elicit a macroscopic current from Piezo1 in Swell1 KO and Swell1 KO + *Flnb* KO cells (n = 7 cells each). Mann-Whitney test: P = 0.5897. Box plot shows median (centre line) and interquartile range (box, 25th–75th percentiles); whiskers indicate

minimum and maximum. **c**, Peak indentation-evoked current from Piezo1 in Swell1 KO and Swell1 KO + *Flnb* KO cells. Piezo1: 327.4 ± 123.8 pA/pF, n = 7 cells; Piezo2 *Flnb* KO: 397.4 ± 150.2 pA/pF, n = 7 cells. Mann-Whitney test: *P* = 0.2593. **d**, Amplitude of stretch-induced currents from osmotic swelling for untransfected Swell1 KO cells (−1.9 ± 0.6 pA/pF, n = 11 cells), Swell1 KO cells transfected with Piezo1 (14.6 ± 4.9 pA/pF, n = 9 cells), and Swell1 + *Flnb* KO cells transfected with Piezo1 (15.5 ± 4.5 pA/pF, n = 12 cells). KW test: \*\*\*\*P = 0.0002. Error bars in **a**, **c**, and **d** are mean ± SEM. All statistical tests are two-sided.

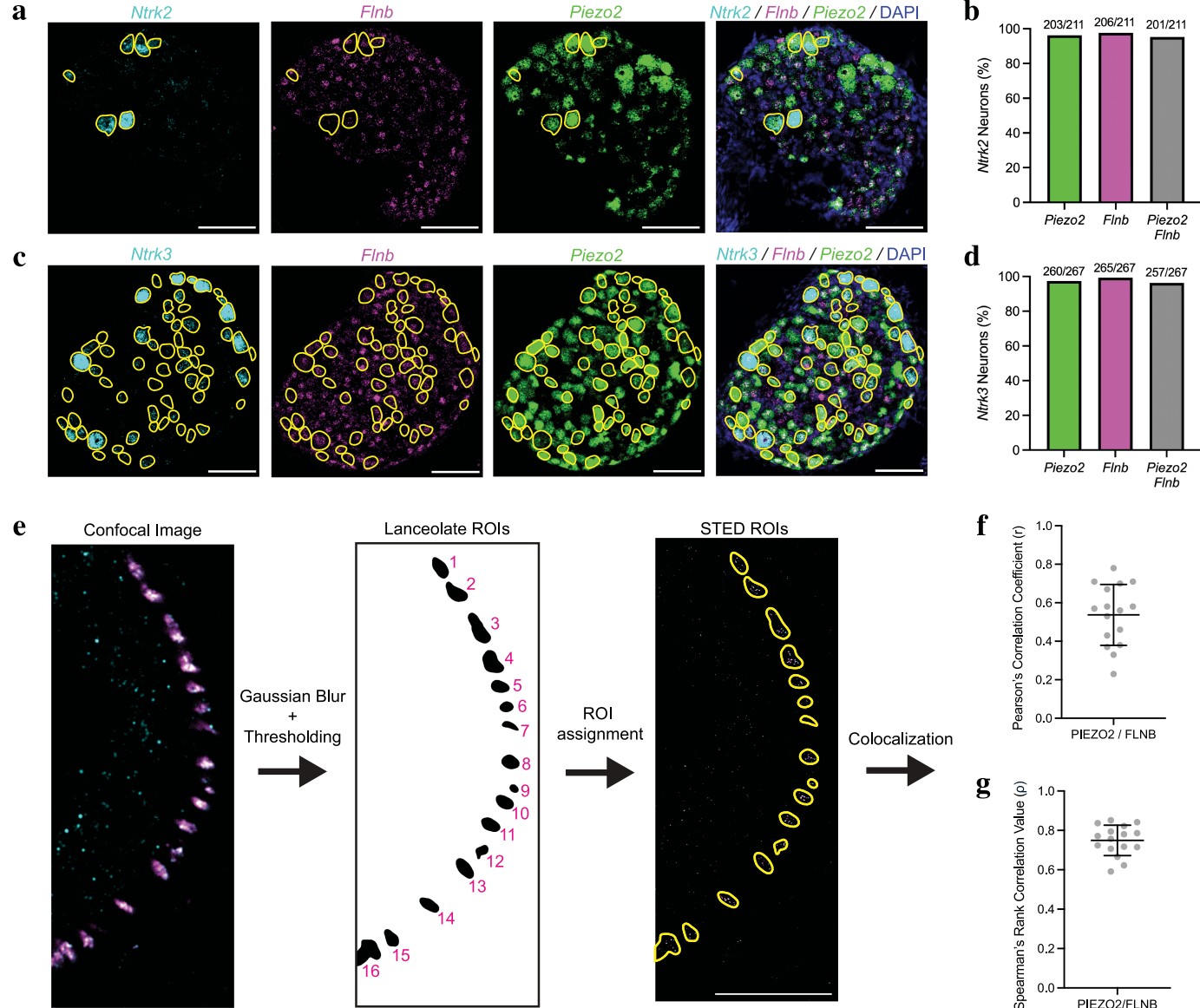

**Extended Data Fig. 9 | Analysis of FLNB and PIEZO2 co-localization.**
**a**, Representative images (11 images from 2 mice) of smFISH on sectioned mouse DRG for *Ntrk2*, *Flnb*, *Piezo2* and merged with DAPI. Yellow outlines indicate *Ntrk2*+ cells. Scale bar = 100 μm. **b**, Quantification of DRG smFISH images with the percentage of *Ntrk2*+ cells expressing *Piezo2*, *Flnb*, or both. The number of co-expressing neurons is above the bar. **c**, Representative images (11 images from 2 mice) of sectioned mouse DRG smFISH for *Ntrk3*, *Flnb*, *Piezo2* and merged with DAPI. Scale bar = 100 μm. **d**, Quantification of DRG smFISH images with the percentage of *Ntrk3*+ cells expressing *Piezo2*, *Flnb*, or both. **e**, Overview of lanceolate ROI assignment for the hair follicle shown in Fig. 4b. **f**, Pearson's correlation coefficient (*r* = 0.54 ± 0.16) for the α-PIEZO2-FLAG and α-FLNB channels of the representative segmented lanceolate ROIs for the hair follicle in **e**. n = 16 lanceolates from 1 representative image. **g**, Spearman's rank correlation coefficient for the same ROIs (ρ = 0.75 ± 0.08). Error bars are mean ± SD. n = 16 lanceolates from 1 representative image.

Eric Mulhall

# Reporting Summary

## Statistics

For all statistical analyses, confirm that the following items are present in the figure legend, table legend, main text, or Methods section.

| n/a | Confirmed | |
|---|---|---|
| ☐ | ☒ | The exact sample size (*n*) for each experimental group/condition, given as a discrete number and unit of measurement |
| ☐ | ☒ | A statement on whether measurements were taken from distinct samples or whether the same sample was measured repeatedly |
| ☐ | ☒ | The statistical test(s) used AND whether they are one- or two-sided<br>*Only common tests should be described solely by name; describe more complex techniques in the Methods section.* |
| ☒ | ☐ | A description of all covariates tested |
| ☐ | ☒ | A description of any assumptions or corrections, such as tests of normality and adjustment for multiple comparisons |
| ☐ | ☒ | A full description of the statistical parameters including central tendency (e.g. means) or other basic estimates (e.g. regression coefficient) AND variation (e.g. standard deviation) or associated estimates of uncertainty (e.g. confidence intervals) |
| ☐ | ☒ | For null hypothesis testing, the test statistic (e.g. *F*, *t*, *r*) with confidence intervals, effect sizes, degrees of freedom and *P* value noted<br>*Give P values as exact values whenever suitable.* |
| ☒ | ☐ | For Bayesian analysis, information on the choice of priors and Markov chain Monte Carlo settings |
| ☒ | ☐ | For hierarchical and complex designs, identification of the appropriate level for tests and full reporting of outcomes |
| ☐ | ☒ | Estimates of effect sizes (e.g. Cohen's *d*, Pearson's *r*), indicating how they were calculated |

*Our web collection on statistics for biologists contains articles on many of the points above.*

## Software and code

Policy information about availability of computer code

| Data collection | MINFLUX data were acquired on a commercial MINFLUX 3D microscope an Olympus IX83 microscope body (Abberior Instruments) using Imspector software (v16.3.15645-m2205) with MINFLUX drivers. Confocal images were collected with NIS-Elements software (version 5.40.01, Nikon). STED imaging was performed on an Abberior Instruments Facility Line 3D STED microscope on an Olympus IX83 microscope body using Lightbox 2025 software (version: 2024.48.21878-gc86bbd647c). Electrophysiology data was collected with pClamp software (version 10.2 and 10.7, Molecular Devices). |
|---|---|
| Data analysis | MINFLUX data analysis was performed using custom code written in MATLAB (version R2021b, MathWorks). Code for structure data and tracking analysis is available at https://github.com/PatapoutianLab/MINFLUX_Localization_and_Tracking_Analysis (https://doi.org/10.5281/zenodo.17625937). Confocal images were analyzed using Fiji (version 2.16.0/1.54p, https://fiji.sc/). STED data was analyzed and deconvolved using Lightbox 2025 software (version: 2024.48.21878-gc86bbd647c). STED colocalization was performed in Fiji using the Coloc 2 plugin (version 3.1.0). FWHM calculations were performed with fwhm_on_spots (https://github.com/sommerc/spots_fwhm). DNA sequences were created and analyzed in SnapGene (Version 8.0.3, Dotmatics). Data visualization and statistical tests were performed with MATLAB 2023 (version R2023b, MathWorks) and Prism (version 10.4.2, GraphPad). Visualization of localizations in Extended Data Fig. 2a were performed using ParaView (version 5.10, kitware). Molecular structures were visualized using MolStar viewer (https://molstar.org/viewer/, version 5.4.1) and Chimera software (version 1.15, UCSF). Graphics were created using Adobe Illustrator (version 2025, Adobe) and BioRender (biorender.com). Sanger sequencing traces of FLNB KO cells were analyzed with the Editco ICE analysis tool (https://ice.editco.bio/#/). Peptides identified from mass spectometry were analyzed using Mascot (Matrix Science, version 2.8.0). |

For manuscripts utilizing custom algorithms or software that are central to the research but not yet described in published literature, software must be made available to editors and reviewers. We strongly encourage code deposition in a community repository (e.g. GitHub). See the Nature Portfolio guidelines for submitting code & software for further information.

# Data

Policy information about availability of data

All manuscripts must include a data availability statement. This statement should provide the following information, where applicable:

- Accession codes, unique identifiers, or web links for publicly available datasets
- A description of any restrictions on data availability
- For clinical datasets or third party data, please ensure that the statement adheres to our policy

Data supporting the article, including the raw MINFLUX analysis output for each experimental condition, are provided as source data. Published protein structures were obtained from the RSCB Protein Data Bank (PIEZO1 6B3R and PIEZO2 6KG7). AlphaFold III models were generated with the Google DeepMind AlphaFold Server. Raw data is available at: https://doi.org/10.5281/zenodo.17644763. All reagents that are not commercially available are available from the corresponding authors upon reasonable request. Source data are provided with this paper. For mass spectometry proteomics, all samples were analyzed using Mascot software (Matrix Science) with human proteins contained in the NCBInr protein database, assuming the digestion enzyme trypsin. Hits were cross referenced against human proteins in the UniprotKB human proteome (UP000005640).

# Research involving human participants, their data, or biological material

Policy information about studies with human participants or human data. See also policy information about sex, gender (identity/presentation), and sexual orientation and race, ethnicity and racism.

| | |
|---|---|
| Reporting on sex and gender | N/A |
| Reporting on race, ethnicity, or other socially relevant groupings | N/A |
| Population characteristics | N/A |
| Recruitment | N/A |
| Ethics oversight | N/A |

Note that full information on the approval of the study protocol must also be provided in the manuscript.

# Field-specific reporting

Please select the one below that is the best fit for your research. If you are not sure, read the appropriate sections before making your selection.

☒ Life sciences   ☐ Behavioural & social sciences   ☐ Ecological, evolutionary & environmental sciences

For a reference copy of the document with all sections, see nature.com/documents/nr-reporting-summary-flat.pdf

# Life sciences study design

All studies must disclose on these points even when the disclosure is negative.

| | |
|---|---|
| Sample size | No analyses were performed in advance to predetermine sample size. Samples sizes were selected based upon prior studies in the literature using comparable assays and effect sizes (e.g. Mulhall, et al., 2023, Ranade, et al., 2014) and on feasibility constraints of MINFLUX imaging and patch clamp recording. For each experiment type, we aimed to include at least three independent biological replicates (independent cultures/ animals prepared on different days) and to repeat experiments on multiple days, with additional technical replicates within each biological replicate as appropriate. The experimental unit is indicated in each figure legend (e.g., molecules, trajectories, cells, or mice), and exact sample sizes (n) are reported there. For MINFLUX structural imaging, each condition includes measurements from ≥3 cells across ≥3 independent experiments/replicates. Because several primary readouts are distribution-based (e.g., inter-blade distance distributions), data acquisition was continued in some conditions to increase the precision of the estimated distribution (i.e., to better capture the full range of observed conformations), without performing interim significance testing and without excluding data based on outcomes. For electrophysiology, recordings were collected across independent transfections and days, with untransfected controls measured on each experimental day and manipulations paired with matched controls. All key findings were reproduced in independent experiments as noted in the figure legends. |
| Data exclusions | For electrophysiology, cells with high access resistance (> 20 MΩ) or low seal resistance (< 1 GΩ) were excluded from data analysis. Cells that changed their morphology during repetitive poke stimulation were also excluded from data analysis. Cells that developed membrane blebs during hypoosmotic challenge were excluded from data analysis. DRG neurons that exhibited a sudden increase in current of several hundred pA or several nA after the application of hypotonic solution, without recovering upon reintroduction to the iso-osmotic solution, were excluded from data analysis due to the potential compromise of the gigaseal stability or plasma membrane integrity during cell swelling. |
| Replication | At least three biological and experimental replicates were performed for each experiment, and experiments were performed over at least two separate days. All attempts at replication were successful. |

| Randomization | This study did not allocate experimental units to groups, and so no randomization was required for any experiment reported. |
| --- | --- |
| Blinding | For the siRNA knockdown experiments in Fig. 3c, the experimenter was blinded to the condition during the recordings. All other electrophysiology experiments were conducted as previously published without blinding. For all MINFLUX experiments, data were analyzed using automated analysis algorithms with the same settings, so blinding was not necessary. For all other experiments, blinding was not necessary since there were no comparisons. |

# Reporting for specific materials, systems and methods

We require information from authors about some types of materials, experimental systems and methods used in many studies. Here, indicate whether each material, system or method listed is relevant to your study. If you are not sure if a list item applies to your research, read the appropriate section before selecting a response.

## Materials & experimental systems

| n/a | Involved in the study |
| --- | --- |
| ☐ | ☒ Antibodies |
| ☐ | ☒ Eukaryotic cell lines |
| ☒ | ☐ Palaeontology and archaeology |
| ☐ | ☒ Animals and other organisms |
| ☒ | ☐ Clinical data |
| ☒ | ☐ Dual use research of concern |
| ☒ | ☐ Plants |

## Methods

| n/a | Involved in the study |
| --- | --- |
| ☒ | ☐ ChIP-seq |
| ☒ | ☐ Flow cytometry |
| ☒ | ☐ MRI-based neuroimaging |

## Antibodies

| Antibodies used | Primary antibodies: guinea pig anti-FLAG (a gift of David Ginty, described in doi: 10.1016/j.neuron.2023.08.023), rabbit anti-FLNB (Thermo Fisher #PA5-52098), and chicken anti-NFH (Abcam ab4680).<br>Secondary antibodies: goat anti-guinea pig Alexa Fluor 594 (Life Technologies A11076), goat anti-rabbit Alexa Fluor 647 (Life Technologies A21245), goat anti-chicken Alexa Fluor 488 (Life Technologies A32931), goat anti-guinea pig STAR RED (Abberior #STRED-1006), goat anti-rabbit STAR ORANGE (Abberior #STORANGE-1002). |
| --- | --- |
| Validation | All antibodies are previously published for use in mouse tissue. For commercially available antibodies, citations are available on the manufacturer's website. For the guinea pig anti-FLAG, use on mouse tissue is described in doi: 10.1016/j.neuron.2023.08.023 and https://doi.org/10.7554/eLife.10874. We always performed no-primary control experiments to validate lack of staining with secondary antibodies.<br>The primary antibodies were validated as follows:<br>Guinea pig anti-FLAG (from https://doi.org/10.7554/eLife.10874): " In CbfbFlag mice, the Flag antibody allows for specific detection of endogenous Flag-CBFβ, which appears to be expressed in nearly all DRG neurons, at varying levels (Figure 4A, B)." The authors performed staining of CbfbFlag on CbfbFlag mice and WT controls and observed specific staining.<br>Rabbit anti-FLNB (from manufacturer's website): "Relative expression in western blot: Antibody specificity was demonstrated by detection of known differential basal expression of the target across tissue/cell models. Expression of Filamin B was observed specifically in A-549 cells and in HEK293 cells using anti-Filamin B Polyclonal Antibody (Product # PA5-52098) in western blot. The relative expression levels of Filamin B within each cell line is shown using RNA-Seq.""Relative expression in different tissues in IHC: Detection of differential expression levels of Filamin B demonstrates antibody specificity. Immunohistochemical analysis of Filamin B using anti-Filamin B Polyclonal Antibody (Product # PA5-52098), shows significant staining of Filamin B in human prostate and shows minimal or weak staining in human skeletal muscle tissues. The relative expression levels of Filamin B within each tissue is shown using RNA-Seq."<br>Chicken anti-NFH (from manufacturer's website): "Anti-Neurofilament heavy polypeptide antibody (ab4680) is a Chicken Polyclonal antibody and is validated for use in ICC, IHC-FrFl, WB. Anti-Neurofilament heavy polypeptide antibody (ab4680) has been cited over 135 times in peer reviewed journals and is trusted by the scientific community. Abcams high quality validation processes ensure Anti-Neurofilament heavy polypeptide antibody (ab4680) has high sensitivity and specificity. Anti-Neurofilament heavy polypeptide antibody (ab4680) has 12 independent reviews from customers." Example citation using this antibody in our laboratory: https://doi.org/10.1038/s41586-022-04860-5. |

## Eukaryotic cell lines

Policy information about cell lines and Sex and Gender in Research

| Cell line source(s) | Swell1-knockout cells were of the Freestyle HEK293-F cell line, originally obtained from ThermoFisher Scientific, and modified as described in Kefauver, et al. 2018. PtK2 (NBL-5) cells were obtained from ATCC (ATCC CCL-56). Clonal Flnb knockout cells were custom made using CRISPR-Cas9 on the Swell1-knockout HEK293-F cell line background as described in the methods. Expi293 cells were obtained from ThermoFisher Scientific. |
| --- | --- |
| Authentication | Commercially available cell lines were authenticated by the supplier. Expi293 cells (from ThermoFisher Scientific) were authenticated for post-thaw viability, mycoplasma testing by PCR, and sterility testing. PtK2 cells (from ATCC) were autheticated by morphological analysis, species verification by isoenzymology, STR profiling, and mycoplasma testing. |

Knockout of the genes encoding Swell1 (LRRC8A, LRRC8B, LRRC8D, and LRRC8E) in the Swell1-KO cells were verified previously in Kefauver, et al. 2018.  Successful knock-out of Swell1 genes was determined by PCR genotyping and Sanger sequencing targeted regions for frameshift mutations and verified by mass spectrometry analysis. Knockout of Flnb in clonal Flnb + Swell1-KO knockout cells was determined by PCR genotyping and Sanger sequencing targeted regions for frameshift mutations.

| | |
|---|---|
| Mycoplasma contamination | All cell lines tested negative for mycoplasma contamination using the MycoAlert® Mycoplasma Detection Kit (Lonza). |
| Commonly misidentified lines (See ICLAC register) | None |

# Animals and other research organisms

Policy information about studies involving animals; ARRIVE guidelines recommended for reporting animal research, and Sex and Gender in Research

| | |
|---|---|
| Laboratory animals | Mice were kept in standard housing with a 12 hour light/dark cycle, with lights on from 6 AM to 6 PM. Room temperature was kept around 22C, and humidity between 30-80% (not controlled). Mice were kept on pelleted paper bedding and provided with paper square nestlets and PVC pipe enrichment. Mice were given ad libitum access to food and water. Mouse ages ranged from 2.5 weeks to 4 months of age for IHC and smFISH experiments. WT mice used for electrophysiology and smFISH experiments were C57BL6/J background (Jackson Laboratories Strain #:000664). Piezo2-smFP-FLAG mice used for IHC experiments were a gift of David Ginty (Piezo2em1.1Ddg/J, CD-1 genetic background, Jackson Laboratories Strain #039935). PCR genotyping from tail snip DNA samples was performed in house using guidelines from Jackson Laboratory and/or through Transnetyx. All mice except for C57BL/6J mice were given metal identification tags (National Band & Tag, 1005-1) on the right ear at an age of 18-30 days old. Upon weaning between 21-30 days of age, mice were co-housed in groups of 2-5 littermates of the same sex. |
| Wild animals | This study did not involve wild animals. |
| Reporting on sex | Sex was not considered in study design. |
| Field-collected samples | This study did not involve field-collected samples. |
| Ethics oversight | All experiments were approved by the Scripps Research Animal Care and Use Committee under protocol # 08-0136. All experiments were performed under the policies and recommendations of the International Association for the Study of Pain and approved by the Scripps Research Animal Care and Use Committee. |

Note that full information on the approval of the study protocol must also be provided in the manuscript.

# Plants

| | |
|---|---|
| Seed stocks | Report on the source of all seed stocks or other plant material used. If applicable, state the seed stock centre and catalogue number. If plant specimens were collected from the field, describe the collection location, date and sampling procedures. |
| Novel plant genotypes | Describe the methods by which all novel plant genotypes were produced. This includes those generated by transgenic approaches, gene editing, chemical/radiation-based mutagenesis and hybridization. For transgenic lines, describe the transformation method, the number of independent lines analyzed and the generation upon which experiments were performed. For gene-edited lines, describe the editor used, the endogenous sequence targeted for editing, the targeting guide RNA sequence (if applicable) and how the editor was applied. |
| Authentication | Describe any authentication procedures for each seed stock used or novel genotype generated. Describe any experiments used to assess the effect of a mutation and, where applicable, how potential secondary effects (e.g. second site T-DNA insertions, mosiacism, off-target gene editing) were examined. |

