## [Peer Review file · Nature]

The Molecular Basis of Force Selectivity by PIEZO2

Corresponding Author: Professor Ardem Patapoutian

Version 0:

Reviewer comments:

Referee #1

(Remarks to the Author)

This manuscript by Mulhall et al., identifies a new mechanism through which Piezo2's distinct mechanosensitivity is conferred. Using a combination of single-molecule MINFLUX fluorescence nanoscopy and electrophysiology, the authors demonstrate that Piezo2 has by smaller and less variable interblade distances compared to Piezo1, causing Piezo2 to possess a more rigid structure. These differences in rigidity are proposed to underlie different activation mechanisms. For example, Piezo1 can be activated by hypertonic cell swelling, which results in blade expansion, supporting a force-from-lipids Piezo1 gating model. On the other hand, this same stimulus led to a compacting of Piezo2 blades and no channel activation, suggesting Piezo2 force sensing is not mediated by lipids. Instead, the authors used protein crosslinking and mass spectrometry-based proteomics to identify molecular candidates that could serve as tethers to link Piezo2 to the underlying actin cytoskeleton, which would support a tethering mechanism of force transduction. A siRNA knockdown screen identified Filamin-B (FLNB) as a potential candidate. Knockdown of FLNB allowed for stretch-activated Piezo2 responses, which are not normally observed, suggesting FLNB is a molecule that can confer response selectivity to Piezo2. The authors also showed that Piezo2 and FLNB co-express in DRG neurons at both the mRNA and protein levels.

Overall, the study is well designed and includes appropriate statistical analyses. The paper is well written, the data are of high quality and clearly presented, and the conclusions drawn by the authors are appropriate and supported. The finding that Piezo2 force transduction is governed by a tether-coupled model is not new, but the identification of the (or perhaps "a") tether is new and exciting.

My only real critique of the study is that the authors should include functional data in DRG neurons showing that knockout of FLNB alters the force selectivity of Piezo2. How closely does the cytoskeleton of the cell lines used mirror that of what is found in DRG neurons? While the authors do show strong co-expression data, a knockout mouse line is commercially available and inclusion of analysis of mechanosensitive currents, in my view, would leave without question that FLNB is a critical molecular tether of native Piezo2 channels. The authors noted that a full-on behavioral analysis is out of the scope of this paper, and I do agree; however, a biophysical analysis of mechanosensitive currents in actual DRG neurons lacking FLNB would very much strengthen the manuscript's main conclusion.

(Remarks on code availability)

Referee #2

(Remarks to the Author)

PIEZO channels are the principal mechanotransduction ion channels in vertebrates, broadly expressed across tissues and involved in a wide range of physiological processes that require mechanical sensing. Among them, PIEZO1 is the most extensively studied and is widely recognized to be gated by the "force-from-lipid" mechanism. It responds to membrane stretch and can be activated by increases in membrane tension. In contrast, the mechanogating properties of PIEZO2 are

less well understood. PIEZO2 is expressed in distinct cell types and is typically activated by cellular indentation rather than membrane stretch. Indeed, previous studies have shown that membrane stretch alone has minimal effects on PIEZO2 activity, raising questions about the molecular mechanism underlying its activation.

In this study, the authors investigate the molecular basis of PIEZO2 gating using MINFLUX microscopy to visualize force-induced conformational changes. This technique, previously validated by the authors for PIEZO1, enables high-resolution tracking of interblade distance changes within the PIEZO triskelion structure in response to mechanical stimuli. While hypotonic conditions, which increase membrane tension, drive conformational expansion of PIEZO1, PIEZO2 shows little to no structural change under the same conditions. Strikingly, when the actin cytoskeleton is disrupted using cytochalasin D, PIEZO2 becomes functionally responsive to membrane stretch. Under these conditions, PIEZO2 also undergoes measurable conformational expansion in response to hypotonic challenge. To identify the molecular link between PIEZO2 and the actin cytoskeleton, the authors combine crosslinking with proteomics and identify filamin B (FLNB) as a key intermediary. Knockdown and colocalization studies further support the role of FLNB in anchoring PIEZO2 to the cytoskeleton and modulating its gating behavior.

In our view, this study represents a foundational contribution to the field. It establishes a molecular framework for understanding how PIEZO2 detects shape changes and highlights the critical role of cytoskeletal tethering in conferring gating specificity. Below, we outline some technical concerns, most of which we believe can be addressed by modifying the text.

Concerns:

1. For 3D tracking experiments, HaloTag constructs were used, and the authors emphasize that they took great care to ensure they were tracking single emitters, addressing a limitation of the earlier study. They state that a “roughly quantal emission intensity pattern was used to isolate single emitters (see Methods).” First, there is no information about the quantal pattern intensity in the Methods. Second, it is not clear why there would not be three labels per functional channel rather than only one emitter per channel.
2. It is unclear whether disrupting the actin cytoskeleton or knocking down FLNB affects PIEZO1 activity or whether PIEZO1 maintains its sensitivity to membrane stretch under these conditions.
3. In Fig. 3C, the FLNB siRNA knockdown shows much less increase in stretch-induced current density compared to the FLNB KO condition. In fact, aside from a few cells, the effects of siRNA knockdown are minimal.
4. In Fig. 3J, the effect of Δ I_{DR5} on indentation-induced amplitude is much greater than in FLNBKO. Why do the effects differ? Does this suggest that other intermediaries besides FLNB are involved in linking PIEZO2 to cytoskeletal elements?
5. The findings of this study convincingly demonstrate that the PIEZO2 blades and IDR5 domain are necessary for the “force-by-tether” response, but they do not establish whether these domains are also sufficient. For example, would a PIEZO1–PIEZO2 chimera, where both the first three blade repeats and the IDR5 domain of PIEZO1 are replaced with those from PIEZO2, exhibit sensitivity primarily to cellular indentation? A functional test of such a construct could be highly informative if successful. While there are certainly plausible reasons why this experiment might fail, it seems like a relatively straightforward and worthwhile test to pursue.

(Remarks on code availability)

Referee #3

(Remarks to the Author)

In the paper “The Molecular Basis of Force Selectivity by PIEZO2” Mulhall et al perform MINFLUX experiments to monitor conformations of PIEZO2 in different conditions and in comparison to PIEZO1. I will not evaluate the biological aspects of this paper, but will comment exclusively the MINFLUX measurements and analysis with respect to technical soundness. This manuscript uses MINFLUX in the imaging mode using DNA-PAINT, however, to my knowledge, for the first time with quenched imaging strands, which reduces the background. This straight forward and timely extension should improve the precision; however, it could also come with own challenges. In addition, the authors show MINFLUX tracking data on Halo-tag labeled PIEZO.

Compared to any other imaging modality, MINFLUX imaging, even using the commercial Abberior MINFLUX, is very complex with numerous parameters to set, and a high risk of artifacts. In this light, I am not entirely convinced that the findings are supported by the data, and I would recommend additional control measurements and complementary data analysis as outlined below.

1. Quality of MINFLUX with fluorogenic DNA-PAINT

a. Whereas the absence of bright diffusing imaging strands and the lower background should improve precision, the quencher will interact transiently with the fluorophore even when the imaging strand is bound to the docking strand. This will lead to intensity fluctuations and a potential bias or loss in accuracy, as the MINFLUX microscope assumes constant intensities during the measurements in all pattern positions. As this is the first report on using fluorogenic DNA-PAINT with MINFLUX, I would recommend showing on simple test samples that the experimental precision is not strongly affected. This could for example be DNA origamis, or direct evaluation of the flickering with Fluorescence correlation spectroscopy or similar.

b. The median localization precisions in ED Fig. 1d and e are quite large, and much larger than the 1-2 nm precisions usually reported in MINFLUX imaging. This holds true for even for fluorogenic probes. This could be a consequence of the flickering, or hint to additional problems, like vibrations, which can be investigated by tracking a bright fluorescent bead. How many photons were used in the last iteration? Is there really an improvement in the localization precision compared to wide-field DNA-PAINT, where routinely localization precisions of 2 nm are achieved, but at much higher throughput? Please analyze the standard deviation of the mean position of independent detection events (re-binding of the imaging strand to the same

docking strand), as an additional measure for the precision. How many re-binding events you measure on average per docking strand?

a. Evaluating structures on the nanometer scale requires nanometer stability over the measurement times (please state how long these are). Such instabilities would not be visible in the standard deviation of the localizations in one track but would lead to random displacements of the tracks themselves.

I would recommend evaluating the displacements over time from repeated binding events (separate tracks) on the same docking strand for each experiment.

2. Data analysis

For data analysis the raw localizations are clustered using a DBSCAN algorithm followed by a GMM. The cluster positions were clustered again to find clusters of 3 within a 60 nm radius. There might be the danger that this approach produces results that are partially analysis artifacts and do not directly report on the biological question (here: extension of PIEZO blades):

a. The reason for the first DBSCAN/GMM step is not entirely clear. We know that all localizations in each track come from the same fluorophore. After cleaning up individual tracks (e.g., remove the first localizations that often show a bias) the mean position of the localizations in each track should give a precise estimate (~1 nm precision) on the docking strand position. Re-binding events should be easy to link because of their close proximity (<2 nm) compared to the distance of distinct docking strands (>20 nm).

b. The number of clusters of three is very low compared to the number of tracks. This of course could be a result of poor labeling efficiencies. Still, I am a bit worried that these clusters of 3 are random associations because of the clustering algorithm with a defined search radius. Also, the quite different shapes and sizes in EDFig3b (which might already be cherry picked examples) point to this. As the plasma membrane is mostly flat, we should not have such strong projection artifacts and obtain mostly triangular shapes. One control would be to add a random displacement in x and y (drawn e.g. from the range -1 to 1 μm for each direction) to each track (same 'tid'), and to redo the analysis.

c. A more robust, direct and statistically powerful analysis approach I would suggest here is to use pairwise distance histograms (or pair correlation functions). These should be calculated on the averages of each track (after cleaning up) to suppress the otherwise too strong self-correlation peak and to improve accuracy. These histograms should have clear and pronounced maxima at the interblade distance. Apart from the simple analysis without tunable parameters, this analysis would have the advantage that also PIEZO complexes with only 2 labels would contribute to the signal, whereas random background is averaged out.

d. Repeatability and statistical tests. In MINFLUX even minor day-to-day misalignments or small changes in the sample preparation can bias the results. This is why I find it crucial that the authors use technical replicates, prepared and imaged independently on separate days, similar to the different conditions they test. In addition to performing statistical tests on the different conditions, they should also perform the same tests on technical replicates to show that the significance is not due to technical biases, especially for those results with a small effect size but strong significance (e.g., Figure 2b,c, Fig. 3e).

3. Additional comments:

a. Please add the Abberior sequence file or a list with the sequence parameters.

b. I would show a zoom of EDFig. 2a.

c. Why was 3D MINFLUX necessary, given the 2D geometry of the problem and given that 3D MINFLUX has a much poorer lateral precision than 2D MINFLUX? I don't see where the z position information is required for the analysis.

(Remarks on code availability)

Referee #4

(Remarks to the Author)

I co-reviewed this manuscript with one of the reviewers who provided the listed reports.

(Remarks on code availability)

Version 1:

Reviewer comments:

Referee #1

(Remarks to the Author)

The authors have adequately addressed all my concerns.

(Remarks on code availability)

Referee #2

(Remarks to the Author)

My concerns were fully addressed by the authors in the revised version.

(Remarks on code availability)

Referee #3

(Remarks to the Author)

While the revised manuscript and the rebuttal could clarify some of my comments, my main concerns about the data analysis have not been addressed sufficiently.

1. Response 1 to Reviewer 2 seems to contain some misinterpretation of the data, I do not think the analysis shows two peaks. Instead, the 'Peak 2' that appears only with a higher EFO cutoff seems to be the tail of the overall distribution. Likely it is the background that shifts the cutoff to above the EFO cutoff.

2. Reply to 1b:

a. the localization precision in MINFLUX is usually not reported as the standard error of the mean, but as the standard deviation of the position in each trace. Please plot the figures in this way to be comparable to previous MINFLUX experiments. I would also recommend comparing to the CRB: $STD = L/(2\sqrt{2N})$.

b. When evaluating drifts, the beads themselves are not so informative. Instead, please choose data sets with more than 1 gold bead used for beam monitoring and correct the drift for the entire data set including all gold beads. Then report the position over time for the gold beads. This gives an estimate of the residual drift. I would recommend plotting the position over time, but also a histogram of the position. From the raw curves it seems that individual data points are easily displaced by 10 nm.

c. DBSCAN vs trace average: it seems that the trace average results in many more data points and higher statistical power. Why not report those in the main manuscript? This is in contrast to Reply 2a that DBSCAN increases precision.

3. Reply to 1a (should have been c):

a. I would suggest investigating drift like this more in detail (and looking at more clusters). It is a bit curious that the three corners seemed to have been imaged mostly sequentially (especially panel 2: first blue, then all red, then all yellow) and not random. An overall drift could explain that.

4. Reply 2c: I would strongly recommend to do the suggested pairwise distance analysis, at least as a control. If there is a preferred blade distance, it will show up very prominently in the distance histograms, if not, then there is something wrong with the data/analysis. It is easy to implement and would greatly improve confidence in the manuscript.

5. Reply 2d: Please do the suggested statistical test across technical/different day replicates to show that significance is not caused by biases.

(Remarks on code availability)

Referee #4

(Remarks to the Author)

I co-reviewed this manuscript with one of the reviewers who provided the listed reports.

(Remarks on code availability)

Version 2:

Reviewer comments:

Referee #3

(Remarks to the Author)

The authors tried to address my comments. Unfortunately, some of the results still could not overcome my concerns.

1. The pairwise distance analysis does not show the expected behaviour. There is no clear peak or shoulder at the inter-blade distance of ~25 nm (PIEZO1) or ~20 nm (PIEZO2). The peaks at ~12 nm (PIEZO1) and ~17 nm (PIEZO2) show the opposite behaviour to the previous analysis (where PIEZO2 had smaller inter-blade distances). These peaks can be explained by re-binding of imaging strands that cause a re-blinking peak. But this analysis does not show any clear organization of the PIEZO proteins. Any flexibility would broaden a peak, and would be visible also in the previous analysis.

2. Significance test. This is not what I meant. I was asking for statistical significance test between the same technical replicates (e.g. Fig.2b between Dataset 1, 2 and 3), and comparing individual technical replicates across the different conditions to see if the control (comparing e.g. Dataset 1 Isotonic, to Dataset 1 Isotonic + CytD, and the other replicates) to see that the reported statistical significance can be attributed to the condition, and not to systematic errors.

(Remarks on code availability)

Referee #5

(Remarks to the Author)

Mulhall et al present a detailed work outlining novel insights into the molecular mechanisms of the force stimuli sensitive ion channel PIEZO2. The study especially relies on detailed superresolution fluorescence microscopy experiments using Minflux and STED microscopy as well as custom-tailored analysis of imaging data. I was specifically asked to judge those superresolution microscopy experiments and data.

First of all, the whole study is well conducted and structured with lots of controls introduced. The outcome is outstandingly interesting and highlights novel important insights that will have an impact in the field. The microscopy experiments are cutting-edge, involving both imaging on fixed cells as well as single-molecule tracking on live cells. This all required detailed and customized data analysis procedures. In principle, I would like to see this excellent study published, but it needs some clarification regarding the microscope data analysis. The data analysis has not been justified and described very well, which I however find important to fully judge the respective results.

First of all, I suggest to already in the main text briefly mention that the Minflux as well as STED microscopy images were recorded on fixed cells as well as what labels have been used and in short how the data has been analyzed.

Also, I find the wording rigid very confusing. I first interpreted rigidity as membrane fluidity or ordering. However, the authors mean the conformational compactness of the proteins – correct? The authors might want to adapt the wording, or, when used the first time, clearly state what they mean by rigidity.

Now to the image/data analysis, which is very important and has not been outlined very well. I suggest detailing it better in this work:

- The distance analysis of the Minflux imaging data could be described in more detail in this manuscript. I know it has been introduced before and is described briefly in the methods, but it would be great to give more details, especially with respect to why the different analysis steps and why and how the thresholds have been chosen and employed. How has the possible random orientation of the receptor proteins in the images been included – or can this be neglected, or has a 3D information been included (nothing is said about the latter)?
- The tracking analysis has been done in 3D – correct? How has the information on the axial dimension been used in the tracking analysis? In general, hardly anything has been written about the tracking analysis – or did I miss it?
- The outline of the EFO threshold is nice – to prove even more that the larger frequency peaks come from multi-molecule events, the authors could, for one experimental protein condition, show EFO diagrams taken from experiments at higher and lower dye-label concentration.
- I could not follow at all the STED microscopy analysis. Why has deconvolution been used and why have certain thresholds been applied? The reason and details of the steps of the final co-localization algorithm remain a mystery to me.

(Remarks on code availability)

The code seems good. It is well documented and did its job. As mentioned, the details and reasons for doing the analysis as is needs to be outlined better in the main text.

Version 3:

Reviewer comments:

Referee #5

(Remarks to the Author)

The authors have very well replied to all of the comments and revised the manuscript accordingly. I suggest publication as is.

(Remarks on code availability)

Appropriate code for the analysis.

Response to Referees' comments:

We would like to thank the reviewers for their thoughtful suggestions and constructive feedback. We provide our specific responses below.

Referee #1 (Remarks to the Author):

This manuscript by Mulhall et al., identifies a new mechanism through which Piezo2's distinct mechanosensitivity is conferred. Using a combination of single-molecule MINFLUX fluorescence nanoscopy and electrophysiology, the authors demonstrate that Piezo2 has by smaller and less variable interblade distances compared to Piezo1, causing Piezo2 to possess a more rigid structure. These differences in rigidity are proposed to underlie different activation mechanisms. For example, Piezo1 can be activated by hypertonic cell swelling, which results in blade expansion, supporting a force-from-lipids Piezo1 gating model. On the other hand, this same stimulus led to a compacting of Piezo2 blades and no channel activation, suggesting Piezo2 force sensing is not mediated by lipids. Instead, the authors used protein crosslinking and mass spectrometry-based proteomics to identify molecular candidates that could serve as tethers to link Piezo2 to the underlying actin cytoskeleton, which would support a tethering mechanism of force transduction. A siRNA knockdown screen identified Filamin-B (FLNB) as a potential candidate. Knockdown of FLNB allowed for stretch-activated Piezo2 responses, which are not normally observed, suggesting FLNB is a molecule that can confer response selectivity to Piezo2. The authors also showed that Piezo2 and FLNB co-express in DRG neurons at both the mRNA and protein levels.

Overall, the study is well designed and includes appropriate statistical analyses. The paper is well written, the data are of high quality and clearly presented, and the conclusions drawn by the authors are appropriate and supported. The finding that Piezo2 force transduction is governed by a tether-coupled model is not new, but the identification of the (or perhaps "a") tether is new and exciting.

My only real critique of the study is that the authors should include functional data in DRG neurons showing that knockout of FLNB alters the force selectivity of Piezo2. How closely does the cytoskeleton of the cell lines used mirror that of what is found in DRG neurons? While the authors do show strong co-expression data, a knockout mouse line is commercially available and inclusion of analysis of mechanosensitive currents, in my view, would leave without question that FLNB is a critical molecular tether of native Piezo2 channels. The authors noted that a full-on behavioral analysis is out of the scope of this paper, and I do agree; however, a biophysical analysis of mechanosensitive currents in actual DRG neurons lacking FLNB would very much strengthen the manuscript's main conclusion.

To test whether FLNB is a critical molecular tether of native PIEZO2 channels, we measured mechanically evoked currents from dissociated mouse DRG neurons. We elected to use acute DsiRNA knockdown of *Flnb* rather than a germline *Flnb* knockout since we estimate that re-deriving and breeding these mice would take ~8 months. We elicited expansive membrane stretch with osmotic swelling in the presence of the SWELL1 inhibitor DCPIB and observed no significant currents above baseline (now in Fig. 4f and shown below). This indicates both that the SWELL1-dependent chloride current evoked by hypotonic stimuli is sufficiently blocked by DCPIB, and that PIEZO2 does

not respond to expansive membrane stretch in DRG neurons. Next, we nucleofected DRG neurons with either DsiRNAs targeting *Flnb* or a non-targeting DsiRNA. We found that DsiRNA-mediated *Flnb* knockdown elicited significant responses to expansive membrane stretch compared with WT cells and the non-targeting DsiRNA control (now in Fig. 4f). These results indicate that FLNB is required for conferring force selectivity to PIEZO2 in somatosensory neurons.

DRG neurons have an axon-dominated actin-spectrin/neurofilament cortex that is fundamentally different from the focal-adhesion/stress-fiber cytoskeleton in HEK293 and PtK2 cells. Notably, we observe the same FLNB-dependent tuning of PIEZO2 force selectivity across HEK293, PtK2, and DRG neurons despite their divergent cytoskeletal architectures, supporting that the effect is intrinsic to the PIEZO2–FLNB linkage rather than a peculiarity of any one type of cell cytoskeleton.

Referee #2 (Remarks to the Author):

PIEZO channels are the principal mechanotransduction ion channels in vertebrates, broadly expressed across tissues and involved in a wide range of physiological processes that require mechanical sensing. Among them, PIEZO1 is the most extensively studied and is widely recognized to be gated by the "force-from-lipid" mechanism. It responds to membrane stretch and can be activated by increases in membrane tension. In contrast, the mechanogating properties of PIEZO2 are less well understood. PIEZO2 is expressed in distinct cell types and is typically activated by cellular indentation rather than membrane stretch. Indeed, previous studies have shown that membrane stretch alone has minimal effects on PIEZO2 activity, raising questions about the molecular mechanism underlying its activation.

In this study, the authors investigate the molecular basis of PIEZO2 gating using MINFLUX microscopy to visualize force-induced conformational changes. This technique, previously validated by the authors for PIEZO1, enables high-resolution tracking of interblade distance changes within the PIEZO triskelion structure in response to mechanical stimuli. While hypotonic conditions, which increase membrane tension, drive conformational expansion of PIEZO1, PIEZO2 shows little to no structural change under the same conditions. Strikingly, when the actin cytoskeleton is disrupted using cytochalasin D, PIEZO2 becomes functionally responsive to membrane stretch. Under these conditions, PIEZO2 also undergoes measurable conformational expansion in response to hypotonic challenge. To identify the molecular link between PIEZO2 and the actin cytoskeleton, the authors combine crosslinking with proteomics and identify filamin B (FLNB) as a key intermediary. Knockdown

and colocalization studies further support the role of FLNB in anchoring PIEZO2 to the cytoskeleton and modulating its gating behavior.

In our view, this study represents a foundational contribution to the field. It establishes a molecular framework for understanding how PIEZO2 detects shape changes and highlights the critical role of cytoskeletal tethering in conferring gating specificity. Below, we outline some technical concerns, most of which we believe can be addressed by modifying the text.

Concerns:

1. For 3D tracking experiments, HaloTag constructs were used, and the authors emphasize that they took great care to ensure they were tracking single emitters, addressing a limitation of the earlier study. They state that a “roughly quantal emission intensity pattern was used to isolate single emitters (see Methods).” First, there is no information about the quantal pattern intensity in the Methods. Second, it is not clear why there would not be three labels per functional channel rather than only one emitter per channel.

We now include a description of how we obtained single emitters in the methods subsection ‘MINFLUX data analysis’:

Tracks were pre-filtered to have greater than 200 localizations per trajectory, and a maximum EFO (effective photon frequency measured at TCP offset) of 130,000. The EFO cutoff ensured that only single dyes were imaged, since the photon emission frequency peak of a single dye was determined to be ~75 kHz, and we observed a second emission peak at approximately twice the single-dye emission frequency (~150 kHz) corresponding to two dyes (Extended Data Fig. 5a).

We also include plots in Extended Data Fig. 5a for photon emission frequency of each final targeting iteration from each tracking dataset that illustrate these emission peaks (also shown below).

We are able to measure only one label per channel rather than three because we sparsely labeled the cells with very low concentrations of substrate (~1 nM). This labeling concentration results in significantly less than 1 label per channel on average. This is important not only for ensuring we are only measuring one channel, and but also for separating fluorescent puncta by distances significantly

larger than the diameter of their point spread function. We now clarify this in the 'Cell preparation for MINFLUX tracking' methods subsection.

2. It is unclear whether disrupting the actin cytoskeleton or knocking down FLNB affects PIEZO1 activity or whether PIEZO1 maintains its sensitivity to membrane stretch under these conditions.

To determine whether FLNB affects PIEZO1 activity, we compared PIEZO1 responses to membrane stretch and indentation in the presence and absence of FLNB. In *Flnb* + Swell1 knockout cells, osmotic swelling-induced current densities were not significantly different from those in *Flnb* wild-type Swell1 knockout cells (now in Extended Data Fig. 8d and shown below). This indicates that FLNB does not affect the force selectivity of PIEZO1. Similarly, indentation-evoked responses did not differ significantly between the two cell types, with no change in either activation threshold or peak current density (now in Extended Data Fig. 8b,c). Since filamins have been proposed to modulate cortical stiffness through actin crosslinking¹, and cortical stiffness can alter PIEZO1 gating², these results suggest that FLNB does not measurably influence PIEZO1 gating via modulation of cortical stiffness. We now include this interpretation in the results and discussion sections.

3. In Fig. 3C, the FLNB siRNA knockdown shows much less increase in stretch-induced current density compared to the FLNB KO condition. In fact, aside from a few cells, the effects of siRNA knockdown are minimal.

We suspect that the low percentage of responders in the FLNB siRNA knockout experiment may be due to a relatively slower rate of FLNB protein turnover. Slow protein turnover has been observed for other filamins, such as FLNA in smooth muscle cells following inducible knockout³. These observations informed our decision to use *Flnb* knockout cells for detailed electrophysiological characterization and more potent, extended dsRNA treatment in Ptk2 cells for the MINFLUX experiments and in DRG neurons for electrophysiology.

4. In Fig. 3J, the effect of ΔIDR5 on indentation-induced amplitude is much greater than in FLNBKO. Why do the effects differ? Does this suggest that other intermediaries besides FLNB are involved in linking PIEZO2 to cytoskeletal elements?

We speculate that more pronounced effect of ΔIDR5 compared to FLNB knockout might reflect two non-mutually exclusive possibilities: (1) deletion of IDR5 may directly alter the intrinsic mechanical properties of PIEZO2, and (2) FLNB functions as part of a larger protein complex that interacts with

IDR5, such that removal of the domain disrupts multiple interactions beyond FLNB itself. Since FLNB is such a large protein and other filamins are known to simultaneously bind multiple partners at once in addition to actin, the tethering interaction is likely to be complex.

We briefly address and rephrase this point in the main text:

“PIEZO2 Δ IDR5 also exhibited significantly reduced indentation responses and increased indentation threshold⁵, although to a much larger degree than those observed for Flnb deletion (Fig. 3j,k). We suspect that deletion of IDR5 might disrupt the intrinsic structural mechanics of the channel or that FLNB functions as part of a larger protein complex that interacts with this domain.”

5. The findings of this study convincingly demonstrate that the PIEZO2 blades and IDR5 domain are necessary for the “force-by-tether” response, but they do not establish whether these domains are also sufficient. For example, would a PIEZO1–PIEZO2 chimera, where both the first three blade repeats and the IDR5 domain of PIEZO1 are replaced with those from PIEZO2, exhibit sensitivity primarily to cellular indentation? A functional test of such a construct could be highly informative if successful. While there are certainly plausible reasons why this experiment might fail, it seems like a relatively straightforward and worthwhile test to pursue.

We agree that the proposed PIEZO1–PIEZO2 chimera would be conceptually informative, but in our experience, chimeras involving partial blade-repeat swaps are rarely functional, likely due to incompatibility between adjoining transmembrane domains⁴. Since chimeras that don’t involve blade repeats are generally more successful, we instead generated a PIEZO1 + IDR5 chimera in which the fifth IDR of PIEZO1 (aa 551–575) was replaced with the corresponding IDR5 region from PIEZO2 (aa 620–672). To test whether IDR5 alone is sufficient for tethering, we measured the diffusion of this chimera by single-molecule MINFLUX tracking, which is our most sensitive assay for detecting physical coupling. We found that microscale and macroscale diffusion coefficients were indistinguishable from wild-type PIEZO1 (now in Extended Data Fig. 5b and shown below), indicating that IDR5 itself is necessary but not sufficient for immobilization, and that tethering to actin via FLNB likely requires additional domains beyond IDR5.

Referee #3 (Remarks to the Author):

In the paper “The Molecular Basis of Force Selectivity by PIEZO2” Mulhall et al perform MINFLUX experiments to monitor conformations of PIEZO2 in different conditions and in comparison to PIEZO1. I will not evaluate the biological aspects of this paper, but will comment exclusively the MINFLUX measurements and analysis with respect to technical soundness.

This manuscript uses MINFLUX in the imaging mode using DNA-PAINT, however, to my knowledge, for the first time with quenched imaging strands, which reduces the background. This straight forward and timely extension should improve the precision; however, it could also come with own challenges. In addition, the authors show MINFLUX tracking data on Halo-tag labeled PIEZO.

Compared to any other imaging modality, MINFLUX imaging, even using the commercial Abberior MINFLUX, is very complex with numerous parameters to set, and a high risk of artifacts. In this light, I am not entirely convinced that the findings are supported by the data, and I would recommend additional control measurements and complementary data analysis as outlined below.

1. Quality of MINFLUX with fluorogenic DNA-PAINT

a. Whereas the absence of bright diffusing imaging strands and the lower background should improve precision, the quencher will interact transiently with the fluorophore even when the imaging strand is bound to the docking strand. This will lead to intensity fluctuations and a potential bias or loss in accuracy, as the MINFLUX microscope assumes constant intensities during the measurements in all pattern positions. As this is the first report on using fluorogenic DNA-PAINT with MINFLUX, I would recommend showing on simple test samples that the experimental precision is not strongly affected. This could for example be DNA origamis, or direct evaluation of the flickering with Fluorescence correlation spectroscopy or similar.

We agree that if residual interactions between the quencher and dye occurred, they would in principle perturb localizations if they occur within a single MINFLUX trace while the docking strand is bound to the imaging strand. Such interactions might transiently lower the photon count during the targeting iteration, potentially shifting the maximum-likelihood estimate of the fluorophore's position. However, in practice, the Abberior imaging sequence performs TCP pattern repeats for each iteration (generally five repeated TCPs in 1 millisecond) to counteract short-lived dark states that might occur – for example – from triplet blinking. If the flickering occurred at the microsecond timescale, it would be strongly mitigated by the repetitive scanning⁵.

However, if the flickering occurred at the millisecond timescale – longer than the time it takes for one iteration to complete – we would expect to see increased variability in photon counts between successive iterations. To test this empirically, we compared the photon counts recorded for the final XY and Z iterations (iterations 8 and 9 of our targeting sequence) for PIEZO1 molecules imaged in a cell with (i) a directly conjugated Alexa 647 dye (imaged in 10 mM MEA), (ii) fluorogenic DNA-PAINT imager (5 nM), and (iii) traditional DNA-PAINT imager (5 nM). We observe no apparent difference in photon count per iteration for the final xy and z iterations used to produce localizations (corresponding to iterations 8 and 9 in our targeting sequence) for PIEZO1 molecules imaged with fluorogenic DNA PAINT (5 nM imager) compared with a directly conjugated Alexa 647 dye imaged in standard imaging buffer with 10 mM MEA:

In contrast, we observed more variability, and higher photon counts for samples imaged with traditional DNA PAINT (5 nM imager), which likely arises from unquenched dyes diffusing into the imaging field. Similarly, if we calculate the standard deviation of photon counts per trace (shown below), we see that the variability between an Alexa Fluor 647 molecule and fluorogenic DNA PAINT is similar, but much higher for traditional DNA PAINT. These data are now in Extended Data Fig. 2d.

Potential flickering at any timescale might lead to an increase in localization error and a decrease in precision. However, we show in our response to the point below that both error and precision are improved when imaging with fluorogenic DNA PAINT in cells compared with traditional DNA PAINT.

b. The median localization precisions in EDFig. 1d and e are quite large, and much larger than the 1-2 nm precisions usually reported in MINIFLUX imaging. This holds true for even for fluorogenic probes. This could be a consequence of the flickering, or hint to additional problems, like vibrations, which can be investigated by tracking a bright fluorescent bead. How many photons were used in the last iteration? Is there really an improvement in the localization precision compared to wide-field DNA-PAINT, where routinely localization precisions of 2 nm are achieved, but at much higher throughput? Please analyze the standard deviation of the mean position of independent detection events (re-

binding of the imaging strand to the same docking strand), as an additional measure for the precision. How many re-binding events you measure on average per docking strand?

This discrepancy arises from the way that the data is analyzed and presented rather than issues with the instrument or samples. We use density-based clustering of MINFLUX localizations to assign molecular positions, as previously described and validated^{6,7}. This approach has several advantages, including the ability to discard outlier localizations in an unbiased way. The error values we reported in Extended Data Fig. 1 were the calculated standard deviation of the clustered localizations that were fit with the 3D Gaussian mixture model, which we denoted as σ . This is a metric of the spatial scatter of the localizations, not localization precision in the classical sense. Our goal in reporting error rather than precision was to show that we can easily determine distal PIEZO blade positions from the full distributions of single-molecule localizations.

Localization precision is commonly calculated for MINFLUX localizations as the standard error of a localizations associated with one trace ID. This is usually done by taking the weighted average of $\sigma_c = \sigma / \sqrt{n}$ for each trace, where σ is the standard deviation and n is the number of localizations. When we apply this per-trace precision estimation to the data in Extended Data Fig. 1, we find that the precision of traditional DNA PAINT is on par with that commonly reported in the literature, while our fluorogenic DNA PAINT approach is indeed more precise (shown below). We now include these per-trace precision analyses in Extended Data Fig. 1 d-e and replace our previous error estimates to align with those values commonly used for MINFLUX in the literature.

We also calculated the precision of clusters identified and assigned via the density based-clustering algorithm. Since this method is highly effective at removing outlier localizations, we found that the precision is indeed enhanced for both methods versus simply taking the mean value of each trace (shown below). We now also include these density-based precision data in Extended Data Fig. 1 d-e.

In terms of instrument stability, we note in the methods that we used gold nanoparticle fiducials that are physically embedded into the glass coverslip. This innovation largely prevents the beads from

'wiggling' from Brownian motion. At least 3 of these fiducials are used for active sample stabilization via back-scattering from a 980 nm laser source, typically resulting in stage displacement or vibration of less than 1 nm in all axes. Additionally, we used active beamline monitoring, which corrects for drift, using at least 3 of the embedded gold fiducial nanoparticles. This system localizes the embedded fiducials with the same 640 nm laser as is used for imaging. We include below a representative example of localization error of four gold fiducials used for beamline monitoring over an entire imaging run below.

This allows the instrument to measure and correct for beam drift. We include below an example below of the drift measured from these same gold fiducials. This drift is corrected for in our analyzed data. Please note that drift is corrected as a rolling average of this data.

As suggested, we also analyzed the standard deviation of the mean position of independent detection events as an additional measure of precision. Using the data in Fig. 1d for the PIEZO1 WT condition as a representative dataset, we calculated the mean of each trace. We then ran the normal clustering algorithm (i.e. second dbscan epsilon = 5 nm) but required only one trace per cluster (minpts = 1). Since the precision of one trace in a cluster is zero, we calculated the precision of clusters containing more than 1 trace. We find that the median precision of independent detection events within a cluster are less than 1 nm in all dimensions. The histograms of precision for all independent detection events are plotted below and are now included in Extended Data Fig. 2b.

We next asked whether taking this alternative approach of taking the mean of each trace could faithfully reproduce our data. For Fig. 1d (the PIEZO1 isotonic and hypotonic conditions), we subjected mean traces values to the same algorithm used to identify triple-labeled PIEZO proteins and found no significant difference in the cumulative distributions of our data (shown below, now included in Extended Data Fig. 2c). This also addresses the concern in comment 2a.

As suggested, we also analyzed how many re-binding events we measure on average per docking strand by measuring the number of traces within a cluster assigned to a specific molecular position of a PIEZO trimer for the same PIEZO1 example data above. We find that each cluster, corresponding to one PIEZO monomer, has 2.8 traces on average:

a. Evaluating structures on the nanometer scale requires nanometer stability over the measurement times (please state how long these are). Such instabilities would not be visible in the standard deviation of the localizations in one track but would lead to random displacements of the tracks themselves.

I would recommend evaluating the displacements over time from repeated binding events (separate tracks) on the same docking strand for each experiment.

We agree that these experiments require nanometer-scale stability, and we hope our response above sufficiently supports this. Our total measurement times varied from 5 to 24 hours, and we now note this in the methods section in the '3D MINFLUX Imaging' subsection. In addition to measuring displacements of the traces (tracks) themselves, we also analyzed the time between traces for identified triple-labeled PIEZO molecules. Below we show an example of these molecules which included multiple traces per cluster and plot a histogram of time each trace was measured after the start of the experiment. These data show that the same docking site can localized several times over the course of at least 19 hours with sufficient precision and are now included in Extended Data Fig. 2a.

2. Data analysis

For data analysis the raw localizations are clustered using a DBSCAN algorithm followed by a GMM.

The cluster positions were clustered again to find clusters of 3 within a 60 nm radius. There might be the danger that this approach produces results that are partially analysis artifacts and do not directly report on the biological question (here: extension of PIEZO blades):

a. The reason for the first DBSCAN/GMM step is not entirely clear. We know that all localizations in each track come from the same fluorophore. After cleaning up individual tracks (e.g., remove the first localizations that often show a bias) the mean position of the localizations in each track should give a precise estimate (~1 nm precision) on the docking strand position. Re-binding events should be easy to link because of their close proximity (<2 nm) compared to the distance of distinct docking strands (>20 nm).

The reason for the first DBSCAN/GMM step is to remove outliers and noise and is equivalent to cleaning up the individual tracks. We address this topic in the comments above and show how DBSCAN/GMM filtering increases precision.

b. The number of clusters of three is very low compared to the number of tracks. This of course could be a result of poor labeling efficiencies. Still, I am a bit worried that these clusters of 3 are random associations because of the clustering algorithm with a defined search radius. Also, the quite different shapes and sizes in EDFig3b (which might already be cherry picked examples) point to this. As the plasma membrane is mostly flat, we should not have such strong projection artifacts and obtain mostly triangular shapes. One control would be to add a random displacement in x and y (drawn e.g. from the range -1 to 1 μm for each direction) to each track (same 'tid'), and to redo the analysis.

There are several probable reasons for the low number of identified clusters versus the total number of tracks. The first reason is, as is pointed out, incomplete labeling efficiency. Since the nanoscale environment of a cell surrounding the PIEZO molecule is quite complex, we also suspect that steric hinderance plays a role, which makes sense given the non-normal distribution of traces per docking site in our response above. We have found that membrane permeabilization increases the detection efficiency, which supports the steric hinderance argument, but this has the issue of exposing proteins which are in intracellular compartments that are not functionally active. Perhaps the primary reason for the low number of identified clusters is the stringency of our trimeric identification algorithm. First, the density of PIEZOs in the plasma membrane are on the order of 10–100 channels per square micrometer⁸, and we scan for isolated trimeric PIEZO molecules with a search radius of $\epsilon = 60$ nm. This means that we certainly reject at least some molecules which are closer together than this distance, however, we find this step important to make sure that we measure the properties of only isolated PIEZO molecules that can't be confused for anything else. Second, as a quality measure, our analysis parameters require a significant number of localizations within a certain radius per cluster, which also tends to reject a significant number of traces.

We believe that the best metric for specificity of our measurements are the biological manipulations we perform. All samples were prepared, imaged, and analyzed in the same way, with the primary variable being either the PIEZO molecule being measured, or the mechanical manipulation of the cell. So, we can conclude that the conformational shifts we see are from biological conditions, rather than random noise from our analysis algorithm. Additionally, we previously validated our clustering algorithm for specificity⁷. For example, we previously measured inter-blade distances on from different tagged locations on the blade domain and observed significantly different inter-blade

distances, even though the same analysis parameters were used. These distances were also consistent with the range expected from curved and flattened cryo-EM structures of these proteins:

At the micro-meter scale, the plasma membrane can be approximated as locally flat, but this does not hold true at the nanoscale. Extensive AFM, electron microscopy, SCIM, and fluorescence imaging work demonstrate that plasma membranes of cells are undulating and corrugated at the same length scales that we are measuring⁹. In addition, PIEZO proteins uniquely interact with the plasma membrane, forming a three dimensional “dome”, whose footprint can span up to 100 nm¹⁰⁻¹². Even modest out-of-plane tilting of the protein can result in projection artifacts if these measurements were performed in 2D. In addition to these cell-intrinsic properties, random thermal noise displaces the flexible blade domains of both PIEZO channels, which means that their z-positions can vary by at least several nanometers. This is supported not only by our own MINFLUX data, but by independent physical modeling^{11,12}. We would also like to mention that our previously published PIEZO1 MINFLUX data have been replicated by a different laboratory using a different analysis and labeling approach¹³.

c. A more robust, direct and statistically powerful analysis approach I would suggest here is to use pairwise distance histograms (or pair correlation functions). These should be calculated on the averages of each track (after cleaning up) to suppress the otherwise too strong self-correlation peak and to improve accuracy. These histograms should have clear and pronounced maxima at the interblade distance. Apart from the simple analysis without tunable parameters, this analysis would have the advantage that also PIEZO complexes with only 2 labels would contribute to the signal, whereas random background is averaged out.

We prefer to use the cluster-based algorithm previously validated in our previous publication and by other groups^{6,7}, especially since the field can more easily compare our previously published results with the results in this manuscript. Additionally, we think that requiring 3 labels per PIEZO molecule is significantly more robust than allowing those doubly labeled molecules into the dataset, even if it is at the expense of more data, since requiring 3 labels to be present within a certain radius acts as a type of coincidence detection that increases our confidence that we are measuring the correct structure.

d. Repeatability and statistical tests. In MINFLUX even minor day-to-day misalignments or small changes in the sample preparation can bias the results. This is why I find it crucial that the authors

use technical replicates, prepared and imaged independently on separate days, similar to the different conditions they test. In addition to performing statistical tests on the different conditions, they should also perform the same tests on technical replicates to show that the significance is not due to technical biases, especially for those results with a small effect size but strong significance (e.g., Figure 2b,c, Fig. 3e).

We agree that these practices are crucial. As noted in our methods section under 'study design' and in the Nature reporting summary, we used at least 3 technical and biological replicates for each experimental condition. Although we didn't mention imaging on separate days in the original manuscript, we did do this, and we now include it in the methods and reporting summary.

3. Additional comments:

a. Please add the Abberior sequence file or a list with the sequence parameters.

We now include the separate imaging sequences used for structural imaging and for live cell tracking as supplemental data files "3D_Imaging_Sequence.json" and "3D_Tracking_Sequence.json".

b. I would show a zoom of EDFig. 2a.

We now include a zoom of this image. This figure is now renumbered as Extended Data Fig. 3a,b.

c. Why was 3D MINFLUX necessary, given the 2D geometry of the problem and given that 3D MINFLUX has a much poorer lateral precision than 2D MINFLUX? I don't see where the z position information is required for the analysis.

As outlined in response to reviewer comment 2d, the intrinsic flexibility of the blade domains, the undulating and corrugated shape of the plasma membrane, and the shape of the PIEZO-membrane domain all necessitate 3D imaging.

Referee #4 (Remarks to the Author):

I co-reviewed this manuscript with one of the reviewers who provided the listed reports.

References

- 1 Nakamura, F., Stossel, T. P. & Hartwig, J. H. The filamins: organizers of cell structure and function. *Cell Adh Migr* **5**, 160-169 (2011).
- 2 Gottlieb, P. A., Bae, C. & Sachs, F. Gating the mechanical channel Piezo1: a comparison between whole-cell and patch recording. *Channels (Austin)* **6**, 282-289 (2012).
- 3 Retailleau, K. *et al.* Arterial Myogenic Activation through Smooth Muscle Filamin A. *Cell Rep* **14**, 2050-2058 (2016).
- 4 Mousavi, S. A. R. *et al.* PIEZO ion channel is required for root mechanotransduction in *Arabidopsis thaliana*. *Proc Natl Acad Sci U S A* **118** (2021).
- 5 Marin, Z. & Ries, J. Evaluating MINFLUX experimental performance *in silico*. *bioRxiv*, 2025.2004.2008.647786 (2025).

- 6 Pape, J. K. *et al.* Multicolor 3D MINFLUX nanoscopy of mitochondrial MICOS proteins. *Proc Natl Acad Sci U S A* **117**, 20607-20614 (2020).
- 7 Mulhall, E. M. *et al.* Direct observation of the conformational states of PIEZO1. *Nature* **620**, 1117-1125 (2023).
- 8 Lewis, A. H. & Grandl, J. Piezo1 ion channels inherently function as independent mechanotransducers. *Elife* **10** (2021).
- 9 Parmryd, I. & Onfelt, B. Consequences of membrane topography. *FEBS J* **280**, 2775-2784 (2013).
- 10 Guo, Y. R. & MacKinnon, R. Structure-based membrane dome mechanism for Piezo mechanosensitivity. *Elife* **6** (2017).
- 11 Haselwandter, C. A., Guo, Y. R., Fu, Z. & MacKinnon, R. Elastic properties and shape of the Piezo dome underlying its mechanosensory function. *Proc Natl Acad Sci U S A* **119**, e2208034119 (2022).
- 12 Haselwandter, C. A., Guo, Y. R., Fu, Z. & MacKinnon, R. Quantitative prediction and measurement of Piezo's membrane footprint. *Proc Natl Acad Sci U S A* **119**, e2208027119 (2022).
- 13 Verkest, C., Roettger, L., Zeitzschel, N. & Lechner, S. G. Cluster nanoarchitecture and structural diversity of PIEZO1 at rest and during activation in intact cells. *bioRxiv*, 2024.2011.2026.625366 (2025).

Referees' comments:

Referee #3 (Remarks to the Author):

The authors tried to address my comments. Unfortunately, some of the results still could not overcome my concerns.

1. The pairwise distance analysis does not show the expected behaviour. There is no clear peak or shoulder at the inter-blade distance of ~25 nm (PIEZO1) or ~20 nm (PIEZO2). The peaks at ~12 nm (PIEZO1) and ~17 nm (PIEZO2) show the opposite behaviour to the previous analysis (where PIEZO2 had smaller inter-blade distances). These peaks can be explained by re-binding of imaging strands that cause a re-blinking peak. But this analysis does not show any clear organization of the PIEZO proteins. Any flexibility would broaden a peak, and would be visible also in the previous analysis.

The apparent discrepancy arises from comparing two different metrics. The pairwise distance histogram reports the *most frequent* distances between all pairs of molecular positions. In contrast, the metric we use in the manuscript is the average inter-blade distance per isolated homotrimeric PIEZO molecule (based on three sampled blade positions per molecule). Differences between the pairwise peak and the per-molecule inter-blade average are therefore expected if the blade domains are conformationally heterogeneous.

From our provided source data for Fig.1b, the average inter-blade distances per molecule are 26 nm for PIEZO1 and 21 nm for PIEZO2 (shown below, left). When we instead compute pairwise distances using every individual inter-blade measurement from the same source data (rather than per-molecule averages), the peak of the pairwise distribution (shown below, right) is different than the average inter-blade distance per molecule. This difference is expected for heterogeneous conformations and does not indicate an inconsistency between analyses.

For PIEZO1, the pairwise distribution indicates that many trimeric molecules adopt conformations with a short inter-blade separation. This is not a new observation and is consistent with our previously reported analysis of PIEZO1 blade cooperativity (<https://doi.org/10.1038/s41586-023-06427-4>). Mechanistically, this phenomenon may be partially explained by an intracellular “handshake” interaction between neighboring blade domains, described in more detail here: <https://doi.org/10.1126/sciadv.adt7046>. Related analyses of distinct PIEZO1 conformational states are also described in <https://doi.org/10.1126/sciadv.ady8052> and <https://doi.org/10.1126/sciadv.adw4402>. A detailed dissection of these substates is beyond the scope of this manuscript.

Importantly, when we overlay the pairwise-distance histograms of isolated trimeric PIEZO molecules with the distributions obtained from the raw mean-value datasets, the peaks align well (shown below). This demonstrates that the peaks in the raw data arise from the same population of identified trimeric PIEZO molecules, rather than spurious events.

We can also observe condition-dependent conformational shifts in the pairwise-distance histograms, even in cases where the average conformational shift is modest. For example, in the PIEZO2 datasets from Fig. 1e, isotonic and hypotonic conditions produce detectable shifts in the pairwise distributions (shown below). These stimulus-dependent shifts would not be expected if the peaks mainly reflected re-binding of imaging strands or sample drift. Additionally, as shown in our previous responses, we do not detect meaningful drift at these length scales.

Together, these analyses show that the pairwise distance distributions are fully consistent with our per-molecule inter-blade measurements. However, because each trimer contributes 3 distances and the distributions are influenced by conformational heterogeneity, the pairwise histograms are not the appropriate metric for quantifying the overall conformational state.

2. Significance test. This is not what I meant. I was asking for statistical significance test between the same technical replicates (e.g. Fig.2b between Dataset 1, 2 and 3), and comparing individual technical replicates across the different conditions to see if the control (comparing e.g. Dataset 1 Isotonic, to Dataset 1 Isotonic + CytD, and the other replicates) to see that the reported statistical significance can be attributed to the condition, and not to systematic errors.

We appreciate the reviewer's concern about potential systematic errors and agree that it is important to verify that apparent condition-dependent differences are not driven by a single outlying dataset. In our experiments, each technical replicate is also a biological replicate derived from different cells. Although we tried to always sample similar regions from each cell, these replicates inevitably contain biological variability and are not expected to be identical copies of the same underlying distribution.

To compare replicates in each condition, we used a Kruskal-Wallis (KW) test with Dunn's post-hoc. This is the standard non-parametric analog of one-way ANOVA for comparing three or more groups and directly assesses differences in distributions across conditions/replicates. Applying this KW test to the reviewer-specified datasets showed no significant differences between replicates within the same condition (shown below).

We next systematically performed this test to every condition imaged with MINFLUX for structural analysis and again found no significant difference between any replicates within a given condition, except for one replicate in the mPIEZO1 Hypotonic dataset in Fig. 1d which we attribute to under sampling. We now explicitly state this result in the Methods section “MINFLUX data analysis – 3D Structural Imaging in Fixed Cells” and provide a comprehensive statistics table in the Supplementary Information (Supplementary Table 1).

Despite the presence of inherent biological variability, these analyses support the conclusion that the condition-dependent differences we report are not driven by systematic errors in individual datasets but instead reflect robust effects of the experimental manipulation.

Referee #5 (Remarks to the Author):

Mulhall et al present a detailed work outlining novel insights into the molecular mechanisms of the force stimuli sensitive ion channel PIEZO2. The study especially relies on detailed superresolution fluorescence microscopy experiments using Minflux and STED microscopy as well as custom-tailored analysis of imaging data. I was specifically asked to judge those superresolution microscopy experiments and data.

First of all, the whole study is well conducted and structured with lots of controls introduced. The outcome is outstandingly interesting and highlights novel important insights that will have an impact in the field. The microscopy experiments are cutting-edge, involving both imaging on fixed cells as well as single-molecule tracking on live cells. This all required detailed and customized data analysis procedures. In principle, I would like to see this excellent study published, but it needs some

clarification regarding the microscope data analysis. The data analysis has not been justified and described very well, which I however find important to fully judge the respective results.

We thank the reviewer for their thoughtful suggestions and constructive feedback. We provide specific responses below. The manuscript is edited so that changes are tracked.

First of all, I suggest to already in the main text briefly mention that the Minflux as well as STED microscopy images were recorded on fixed cells as well as what labels have been used and in short how the data has been analyzed.

In the revised manuscript, we have added a sentence in the main text that states that the MINFLUX microscopy images were recorded on fixed cells (lines 79-80). In the same paragraph, we specify that ATTO 643 and Iowa Black FQ was used as the fluorophore and quencher on the fluorogenic imaging strands (lines 81-82), and provide a brief description of how the data was analyzed (lines 86-88).

For 3D live-cell tracking, we now state in the main text that we used Janelia Fluor 635 HaloTag ligand (line 151) and include a concise description of how the trajectories were analyzed (lines 153-155).

For STED and confocal imaging, we clarify that the images were acquired in fixed skin sections (Lines 235-237), and explicitly state which epitopes were targeted. The labeling steps are described in detail in the Methods section “Immunohistochemical co-localization of PIEZO2-smFLAG and FLNB in mouse skin”. We also rephrase the brief description of the analysis in the main text and refer the reader to a substantially expanded Methods section, as detailed in our response to the reviewer’s later point on STED analysis.

Also, I find the wording rigid very confusing. I first interpreted rigidity as membrane fluidity or ordering. However, the authors mean the conformational compactness of the proteins – correct? The authors might want to adapt the wording, or, when used the first time, clearly state what they mean by rigidity.

We agree that this term could be misinterpreted. Our intention was to describe an apparent increase in conformational compactness (i.e., reduced flexibility) of the PIEZO blade domains. We now clarify this at the first mention of “rigid” in the main text by explicitly stating that we are referring to the apparent conformational rigidity of the blade domains (lines 95-96).

Now to the image/data analysis, which is very important and has not been outlined very well. I suggest detailing it better in this work:

- The distance analysis of the Minflux imaging data could be described in more detail in this manuscript. I know it has been introduced before and is described briefly in the methods, but it would be great to give more details, especially with respect to why the different analysis steps and why and how the thresholds have been chosen and employed. How has the possible random orientation of the receptor proteins in the images been included – or can this be neglected, or has a 3D information been included (nothing is said about the latter)?

In the revised manuscript, we have separated the MINFLUX imaging and tracking analysis into two dedicated Methods sections and expanded both in detail. The analysis of MINFLUX structural imaging data is now described in “MINFLUX data analysis – 3D Structural Imaging in Fixed Cells”. In this section, we provide a step-by-step description of the analysis pipeline, including the rationale for each filtering step and the choice of thresholds and parameters.

We include 3D information in our analysis methods and so we account for possible random orientation of the receptor proteins. We now explicitly state this in the methods, and mention in the main text that inter-blade distances were calculated from 3D data.

- The tracking analysis has been done in 3D – correct? How has the information on the axial dimension been used in the tracking analysis? In general, hardly anything has been written about the tracking analysis – or did I miss it?

We confirm that the live-cell MINFLUX tracking data were acquired and analyzed in 3D, and that the axial (z) coordinate is fully included in the trajectory analysis.

Previously, the tracking analysis was described together with the structural imaging analysis. In the revised manuscript, we now provide a separate Methods section, “MINFLUX data analysis – 3D Tracking in Live Cells,” in which we greatly expand the description of our analysis. We detail the filtering criteria for localizations and trajectories, the choice of time windows for mean-squared displacement (MSD) fitting, and how diffusion coefficients are extracted from 3D trajectories. In the Results section of the main text, we now explicitly state that diffusion coefficients are derived from 3D trajectories and MSD analysis.

- The outline of the EFO threshold is nice – to prove even more that the larger frequency peaks come from multi-molecule events, the authors could, for one experimental protein condition, show EFO diagrams taken from experiments at higher and lower dye-label concentration.

To further prove that the larger frequency peaks come from multi-molecule events, we performed additional MINFLUX tracking experiments on PIEZO1 using a higher labeling concentration (4 nM JF635-HaloTag ligand) compared to the 0.5 nM used in the datasets included in the manuscript. Under otherwise identical imaging conditions, increasing the dye concentration led to a disproportionate increase in the higher-frequency EFO peak (~150 kHz) relative to the single-molecule peak (~75 kHz), consistent with a higher fraction of multi-molecule events at higher labeling densities. We now include the EFO histograms comparing 0.5 nM and 4 nM labeling concentrations in Extended Data Fig. 5b (also shown below).

- I could not follow at all the STED microscopy analysis. Why has deconvolution been used and why have certain thresholds been applied? The reason and details of the steps of the final co-localization algorithm remain a mystery to me.

We have substantially expanded and revised our “STED imaging and data analysis” methods section to clarify the rationale and implementation of each analysis step. We explain why and how deconvolution was used, and why the thresholds were chosen. We have also expanded our description of the co-localization algorithm.

Referee #5 (Remarks on code availability):

The code seems good. It is well documented and did its job. As mentioned, the details and reasons for doing the analysis as is needs to be outlined better in the main text.